# Microbial consortia at steady supply

**Thibaud Taillefumier[1,2,3], Anna Posfai[1], Yigal Meir[4], Ned S Wingreen[1,5]***

[1]Lewis-Sigler Institute for Integrative Genomics, Princeton University, Princeton, United States; [2]Department of Mathematics, The University of Texas at Austin, Austin, United States; [3]Department of Neuroscience, The University of Texas at Austin, Austin, United States; [4]Department of Physics, Ben-Gurion University of the Negev, Beer Sheva, Israel; [5]Department of Molecular Biology, Princeton University, Princeton, United States

**Abstract** Metagenomics has revealed hundreds of species in almost all microbiota. In a few well-studied cases, microbial communities have been observed to coordinate their metabolic fluxes. In principle, microbes can divide tasks to reap the benefits of specialization, as in human economies. However, the benefits and stability of an economy of microbial specialists are far from obvious. Here, we physically model the population dynamics of microbes that compete for steadily supplied resources. Importantly, we explicitly model the metabolic fluxes yielding cellular biomass production under the constraint of a limited enzyme budget. We find that population dynamics generally leads to the coexistence of different metabolic types. We establish that these microbial consortia act as cartels, whereby population dynamics pins down resource concentrations at values for which no other strategy can invade. Finally, we propose that at steady supply, cartels of competing strategies automatically yield maximum biomass, thereby achieving a collective optimum.

## Introduction

***For correspondence:** wingreen@princeton.edu

**Competing interests:** The authors declare that no competing interests exist.

Microbial diversity is ubiquitous. Every gram of soil or liter of seawater contains hundreds or more microbial species (*Daniel, 2005*). In humans, the gut microbiome comprises at least 500 microbial species (*Lozupone et al., 2012*). These diverse microbial communities are widely credited with division of labor, collectively reaping the benefits of specialization by dividing tasks among different organisms. In a few well-studied cases, microbial communities have been observed to coordinate their metabolic fluxes. For instance, depleting the pool of external resources available to a microbial community can lead to the establishment of mutualism via the exchange of metabolic by-products between species (*Hillesland and Stahl, 2010*). Shotgun sequencing has begun to unveil the biochemical networks at work in complex environmentally sampled communities (*Tyson et al., 2004*; *Gill et al., 2006*), and reconstructing the genomes of member species suggests that microbial communities exploit metabolic interdependencies to adapt to their environment (*Wrighton et al., 2014*; *Hug et al., 2015*). However, the lack of knowledge about gene functions and gene distributions in individual cells hinders the interpretation of this data (*Cordero and Polz, 2014*).

There are also serious conceptual challenges to understanding diversity in metabolically competing microbial communities. For instance, the emergence of diversity in 'consumer-resource' models is limited by the competitive exclusion principle: at stationary state, the number of coexisting species cannot exceed the number of available resources (*Hardin, 1960*; *Levin and Equilibria, 1970*). This principle severely limits diversity in models that consider a few resources as in the case of the 'paradox of the plankton' (*Petersen, 1975*). Another essential challenge is understanding the persistence of microbial diversity in the face of potentially more fit metabolic variants; these reinforce the challenge posed by the competitive exclusion principle: in consumer-resource models, a fitter strain

**eLife digest** Microbes are found in virtually every environment on Earth. Like other organisms, microbes grow by using enzymes to convert nutrients into proteins, DNA and other molecules that make up their cells. Together, these chemical transformations define the "metabolism" of a microbe.

In any given environment, there is almost always a diverse variety of microbes living together. Different microbes in these communities will use different combinations of enzymes to exploit the available nutrients, and members of well-studied communities have been found to work together to make the most of the nutrient source. This is remarkable because one might expect competition between microbes to select for a single "best" microbe, rather than diverse communities.

The economic concept of "division of labor" suggests that if microbes divide chemical tasks between each other, they will use the available resources more efficiently. The concept provides a possible explanation for metabolic diversity amongst microbes, yet it remains to be shown whether microbial communities actually benefit from a division of labor.

Here, Taillefumier et al. used mathematical models to reveal that even in a uniform environment, metabolic competition generally leads to the steady coexistence of distinct microbes, collectively called a "consortium". In a consortium, distinct microbes organize themselves to create a community-level metabolism that best exploits the nutrients present. The models showed that while growing, a consortium depletes the available pool of nutrients to such low levels that only members of the consortium can survive. The findings suggest that the benefit of metabolic diversity stems from the ability of a consortium to automatically deplete nutrients to levels at which no other microbes can invade.

Taillefumier et al. propose that consortia that arise naturally under conditions where there is a steady supply of nutrients produce the maximum mass of microbes. Future experiments that analyze the impact of fluctuating nutrient supply may help us to understand the benefit of metabolic diversity in real-world microbial communities.

colonizes a niche at the expense of those already present by depleting the pool of essential resources, generally leading to a collapse in diversity (*Shoresh et al., 2008*).

The above conceptual challenges call for a physically-based model for competing metabolic strategies. However, classical consumer-resource models generally prescribe the rate of biomass production via phenomenological functions of the abundances of essential resources without explicit conservation of fluxes (*Liebig, 1840*; *Monod, 1950*). Here, we introduce a flux-conserving physical model for microbial biomass production to address two intertwined questions: Can metabolic competition leads to microbial division of labor? And what efficiencies can microbes achieve via such a division of labor?

Considering that biomass (primarily protein [*Simon and Azam, 1989*; *Feijó Delgado et al., 2013*]) results from the assembly of building blocks (amino acids or amino acid precursors), we explicitly model the fluxes associated with the metabolic processing of these building blocks, including enzyme-mediated import and conversion (*Almaas et al., 2004*). Different metabolic strategies — or cell types — are defined by specific distributions of these enzymes, which collectively satisfy a budget constraint. The population dynamics of different cell types is governed by competition for external building blocks that are steadily supplied in a spatially homogeneous environment. We consider that the system is constantly subjected to colonization attempts by other cell types, possibly leading to invasion of the already present microbial population. For fixed external concentrations of building blocks we therefore seek uninvadable strategies, i.e. optimal cell types that achieve the fastest possible growth rate. Metabolic division of labor stably emerges if competitive population dynamics leads to the coexistence of jointly optimal cell types with distinct strategies, which we refer to as a microbial 'consortium'.

In human economies, consortia that avoid competition by controlling prices are called 'cartels'. At fixed building-block supply, we find that metabolic competition between microbes similarly leads to the emergence of a kind of cartel that controls resource availability via population dynamics.

Specifically, cartels avoid competition by pinning down resource concentrations at values for which no other strategy can outcompete the cartel's members. We employ optimization principles from transport-network theory to elucidate the structure of these cartels, relating the metabolic strategies of their constituent cell types to the hierarchy of external building-block availabilities. This analysis illustrates how division of labor among distinct metabolic types can be predicted from optimization principles. Finally, we propose that cartels also yield maximum biomass, constituting a microbial example of Adam Smith's 'invisible hand' leading to collective optimal usage of resources. In this regard, microbial cartels improve on human cartels insofar as price-fixing by the latter generally leads to non-optimal use of resources.

## Model

In this section, we present a model for the population dynamics of cell types metabolically competing for external resources (see **Figure 1**). Importantly, biomass production is governed by a physical model that respects flux conservation.

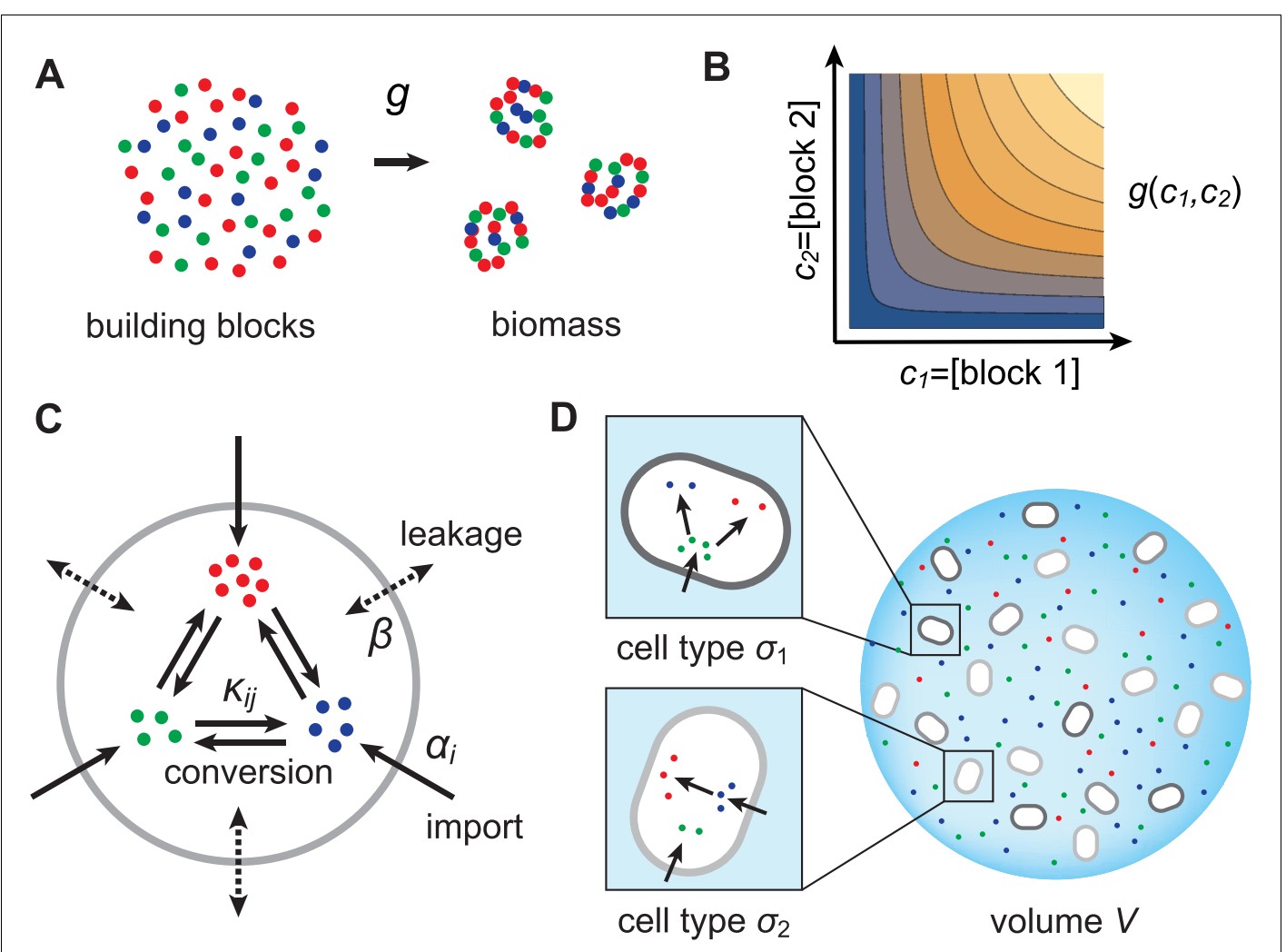

**Figure 1.** Model for metabolically competing cell types. (A) The rate of biomass production $g(c_1, \ldots, c_p)$ is a function of the internal building-block concentrations. (B) Biologically relevant growth-rate functions $g(c_1, \ldots, c_p)$ are increasing with respect to $c_i$ with diminishing returns. (C) Different cell types, i.e. metabolic strategies, are defined as specific distributions of enzymes for import $\alpha_i$ and conversion $\kappa_{ji}$, subject to a finite budget. (D) Cell types (e.g. $\sigma_1$ and $\sigma_2$) compete for external building blocks that are steadily and homogeneously supplied in volume $V$.

## Resource-limited growth model

As cellular growth is primarily due to protein biosynthesis, we consider biomass production to result from the incorporation of building blocks (amino acids or amino acid precursors) into biologically functional units (proteins). Specifically, we assume that biomass production requires $p$ types of building blocks and we denote by $b_i$, $1 \leq i \leq p$, the concentration of block $i$ in cellular biomass. To maintain the stoichiometry of building blocks in biomass, microbes that grow at rate $g$ incorporate block $i$ at rate $g b_i$. As the incorporation of building blocks is limited by the internal availability of free building blocks, we model the growth rate as a function $g(c_1, \ldots, c_p)$, where $c_i$ is the internal concentration of block $i$. To obtain a plausible functional form for $g(c_1, \ldots, c_p)$, we consider the rate of incorporation of a building block to be proportional to its concentration. We also consider that building blocks are sequentially incorporated into biomass (e.g. via protein elongation). Then the time to produce a unit of biomass (e.g. a protein) is the sum of the incorporation times for each type of block $i$, which we take to be proportional to $b_i/c_i$, the ratio of the building-block concentration in cellular biomass to the internal free building-block concentration. The growth rate, which is proportional to the inverse of this time, therefore has the form

$$g(c_1, \ldots, c_p) = \gamma \left( \frac{b_1}{c_1} + \ldots + \frac{b_p}{c_p} \right)^{-1}, \tag{1}$$

where $\gamma$ is a rate constant. For simplicity, we consider that all microbes use the same molecular machinery and building-block stoichiometry to produce biomass. Thus, we consider that the rate function $g(c_1, \ldots, c_p)$ is universal, independent of the metabolic strategy used by a microbe to accumulate building blocks. As defined by *Equation (1)*, $g(c_1, \ldots, c_p)$ is an example of a growth-rate function satisfying the biologically relevant requirements that more internal resources yield faster growth, i.e. $g(c_1, \ldots, c_p)$ is an increasing function of its arguments, and that the relatively scarcest resources are the most growth-rate limiting, i.e. $g(c_1, \ldots, c_p)$ has the property of diminishing returns. Importantly, our analysis and conclusions hold for all growth-rate functions that satisfy these natural requirements.

In order to accumulate block $i$ internally, a microbe can import block $i$ from the external medium or produce it internally via conversion of another building block $j$. Thus, to produce biomass, microbes can substitute resources for one another. We allow all possible imports and conversions. The quantitative ability of a microbe to import and convert building blocks constitutes its 'metabolic strategy', and corresponds to the cell's expression of the enzymes that mediate building-block import and conversion. For simplicity, we assume that metabolic fluxes are linear in both enzyme and substrate concentrations. This assumption corresponds to enzymes operating far from saturation, which is justified during resource-limited growth. Specifically, denoting the external concentration of block $i$ by $c_i^{\text{ext}}$, the enzyme-mediated fluxes associated with import and conversion of block $i$ have the form $\alpha_i c_i^{\text{ext}}$ and $\kappa_{ji} c_i$, respectively, where $\alpha_i$ and $\kappa_{ji}$ are enzymatic activities, which are proportional to the number of enzymes dedicated to each metabolic process. In addition to these active fluxes, we include passive transport of building blocks across cell membranes via facilitated diffusion (*Pi et al., 1991*; *Wehrmann et al., 1995*). For simplicity, we model passive transport via a single leakage rate $\beta$, yielding a net influx $\beta(c_i^{\text{ext}} - c_i)$ for building block $i$. As cells can only devote a certain fraction of their resources to the production of enzymes, we require the enzymatic activities of each microbe to satisfy a budget constraint,

$$\sum_i \alpha_i + \sum_{j,i} \kappa_{ji} \leq E, \tag{2}$$

where $E$ denotes the total enzyme budget. The metabolic strategy of a cell type $\sigma$ is specified by a set of enzyme activities $\{\alpha_{i,\sigma}, \kappa_{ij,\sigma}\}$ that satisfy the budget constraint *Equation (2)*.

Note that for given external building-block concentrations, a cell type is 'optimal', i.e. achieves the fastest growth rate, if no other cell type can achieve the same growth rate with a lower enzyme budget. If such a cell type existed, allocating the saved enzyme budget to importing more building blocks would yield a new cell type with higher internal building-block concentrations, and thereby faster growth.

Our model considers a very simplified coarse-grained description of metabolic pathways. In reality, the details of biochemistry play an important role in determining metabolic efficiency. While our modeling framework can be easily generalized to realistic metabolic networks, capturing the complexity of real metabolic pathways is not the purpose of the present analysis, which aims at general principles. Indeed, because our model is anchored in flux conservation, we expect our results concerning the emergence and benefit of division of labor in microbial communities to hold independent of specific pathway biochemistry.

## Conservation of building blocks

We consider various cell types $\sigma$ growing in a homogeneous environment of volume $V$. We denote the dimensionless population count of cell type $\sigma$ by $n_\sigma$ and the total population count by $N = \sum_\sigma n_\sigma$. In the volume $V$, we consider that the $p$ building blocks are steadily supplied at rates $s_i$ (concentration/time) and can be lost, e.g. via degradation and/or diffusion out of the volume, at a rate $\mu$. Each cell type processes building blocks according to its own metabolic strategy. Conservation of internal building block $i$ for cell type $\sigma$ prescribes the dynamics of the internal concentration $c_{i,\sigma}$ (see Appendix 1),

$$\dot{c}_{i,\sigma} = (\alpha_i + \beta)c_i^{\text{ext}} - \beta c_{i,\sigma} - \sum_{j \neq i} \kappa_{ji} c_{i,\sigma} + \sum_{j \neq i} \kappa_{ij} c_{j,\sigma} - g b_i, \tag{3}$$

where the only nonlinearity is due to the growth function $g$. Populations of the various cell types exchange building blocks with the external resource pool via import and passive transport, and also via biomass release upon cell death (*Simpson et al., 2007*; *Schütze et al., 2013*). Conservation of extracellular building block $i$ prescribes the dynamics of the external concentration $c_i^{\text{ext}}$ (see Appendix 2),

$$\dot{c}_i^{\text{ext}} = s_i - \mu c_i^{\text{ext}} - \frac{v}{V - Nv}\left(\sum_\sigma n_\sigma \phi_{i,\sigma}\right), \tag{4}$$

with cell-type-specific fluxes

$$\phi_{i,\sigma} = (\alpha_{i,\sigma} + \beta)c_i^{\text{ext}} - \beta c_{i,\sigma} - f \delta b_i, \tag{5}$$

where $\delta$ is the rate of cell death (assumed constant) and $f$ is the fraction of biomass released upon cell death. Per-cell fluxes $\phi_{i,\sigma}$ contribute to changing the external concentration $c_i^{\text{ext}}$ via a geometric factor $v/(V - Nv)$, the ratio of the average individual cellular volume $v$ to the total extracellular volume $V - Nv$. As intuition suggests, the smaller the number of cells of a particular type, the less that cell type impacts the shared external concentration via metabolic exchanges.

## Competitive population dynamics

The inverse of the cellular death rate $\delta$, i.e. the lifetime of a cell, is much larger than the timescales associated with metabolic processes such as building-block-diffusion, conversion, and passive/active transport. This separation of timescales justifies a steady-state approximation for the fast-variables: $\dot{c}_{i,\sigma} = 0$ and $\dot{c}_i^{\text{ext}} = 0$. With this approximation, *Equations (3) and (4)* become flux-balance equations for building blocks. Solving *Equation (3)* with $\dot{c}_{i,\sigma} = 0$ yields the internal concentrations $c_{i,\sigma}(c_1^{\text{ext}}, \ldots, c_p^{\text{ext}})$ as functions of the external concentrations. In turn, solving *Equation (4)* with $\dot{c}_{i,\sigma} = 0$ and using the functions $c_{i,\sigma}(c_1^{\text{ext}}, \ldots, c_p^{\text{ext}})$ yields the external concentrations $c_i^{\text{ext}}(\{n_\sigma\})$, as well as the growth rates of cell types $g_\sigma(\{n_\sigma\})$, as functions of the populations of cell types. Hence, the population dynamics of the cell types is described by a system of ordinary differential equations

$$\frac{\dot{n}_\sigma}{n_\sigma} = g_\sigma(\{n_\sigma\}) - \delta, \tag{6}$$

which are coupled via the external concentrations. Note that the population dynamics is driven and dissipative: building blocks are constantly both supplied to and lost from the external media, while cell death leads to loss of building blocks because only a fraction of biomass is recycled ($f < 1$). In particular, we expect the dissipative character of the dynamics to drive the microbial population toward

a stationary state, with at most $p$ coexisting cell types due to the competitive exclusion principle (*Hardin, 1960*; *Levin and Equilibria, 1970*).

The population dynamics prescribed above can be simulated by standard numerical methods. If the number of distinct strategies initially introduced exceeds the number of resources, then over time some cell types will become 'extinct', i.e. $n_\sigma<1$. We exploit this property to simulate competition between distinct cell types: whenever a cell type $\sigma$ is driven to extinction, we replace it with another randomly sampled strategy $\sigma'$ with $n_{\sigma'}=1$. If the already present cell types are not optimal, newly introduced cell types $\sigma'$ may increase in population at the expense of those present. In any case, we continue to introduce new cell types over the time course of the dynamics. The closer to optimality the already present cell types are, the smaller the probability that a new random strategy will successfully colonize. Eventually, at long times, the surviving population will consist entirely of optimal cell types and will no longer change. It is this final population that concerns us; we only simulate metabolic competition to gain insight into the final optimized population, which is independent of the specific dynamics of the simulation.

In the following, we characterize the enzyme distributions $\{\alpha_i, \kappa_{ij}\}$ of the cell types that are present in the final population established via competitive population dynamics for equal stoichiometry ($b_i = 1$). This characterization requires us to define the 'metabolic class' of each cell type in terms of its set of utilized enzymes. In particular, two strategies $\sigma$ and $\sigma'$ belong to the same metabolic class $\mathcal{M}$ if and only if $\alpha_i = 0 \Leftrightarrow \alpha'_i = 0$ and $\kappa_{ij} = 0 \Leftrightarrow \kappa'_{ij} = 0$, where $\alpha_i$, $\alpha'_i$, $\kappa_{ij}$, and $\kappa'_{ij}$ are enzyme activities. We will show that metabolic competition leads to the emergence of consortia of cell types belonging to specific metabolic classes. In our analysis, the term 'consortium' designates a community of distinct cell types that cannot be invaded, or equivalently, that cannot be outgrown by any other cell types at specific fixed supply rates. In particular, consortia are composed of co-optimal cell types. The term 'cartels' refer to special communities that, in addition of being consortia, can also pin down building-block concentrations at fixed values for a range of different supply rates. Such cartels comprise at least as many distinct cell types as there are resources.

## Results

### Numerical results

In this section, we demonstrate the possibility of stable coexistence at steady supply rates by simulating competitive population dynamics subject to continual invasion by new metabolic variants. We consider that coexistence is stable when a population of distinct cell types can resist invasion by any other metabolic variants. In our simulations, cell types have distinct metabolic strategies defined by randomly chosen enzyme distributions $\{\alpha_i, \kappa_{ji}\}$ satisfying the budget constraint *Equation (2)*, with the universal growth-rate function *Equation (1)* and uniform biomass stoichiometry ($b_i = 1$).

First, we show that competitive population dynamics with the continual introduction of new cell types leads to a stationary state with fixed building-block availability and with fixed populations of distinct cell types. Second, we show that these final cell types achieve optimal growth given the fixed building-block supply rates. Third, we show that final, optimal populations generally consist of consortia of distinct cell types and that a consortium of identical cell types can emerge for different building-block supplies.

#### Competitive dynamics leads to stationary states

We simulated the population dynamics *Equation (6)* subjected to continual invasion by metabolic variants and found that metabolic competition leads to stationary states. In our simulations, the volume of the colony $V$ is chosen so that the carrying capacity is $\approx V/v = 10^5$ cells and the dynamics is simulated for a duration of $10^5/\delta$, i.e. on the order of $10^5$ generations. *Figure 2* shows three independent simulations for $p = 3$ and supply rates $s_1 = 11, s_2 = 9, s_3 = 0$, starting with 24 different initial metabolic strategies for each simulation. While 3 types may coexist for extended periods according to the competitive exclusion principle, the 21 other types have populations $n_\sigma$ that decay exponentially until extinction, i.e. until $n_\sigma \leq 0.9999$. Upon extinction, a new type is introduced at $n_\sigma = 1$.

To begin each simulation, the 24 randomly chosen cell types $\sigma$ are introduced at low population ($n_\sigma = 1$) in the volume $V$ where the building blocks are abundantly supplied ($c_i^{\text{ext}} \simeq s_i/\mu$). Cellular growth quickly depletes the external concentrations of building blocks, until the overall population

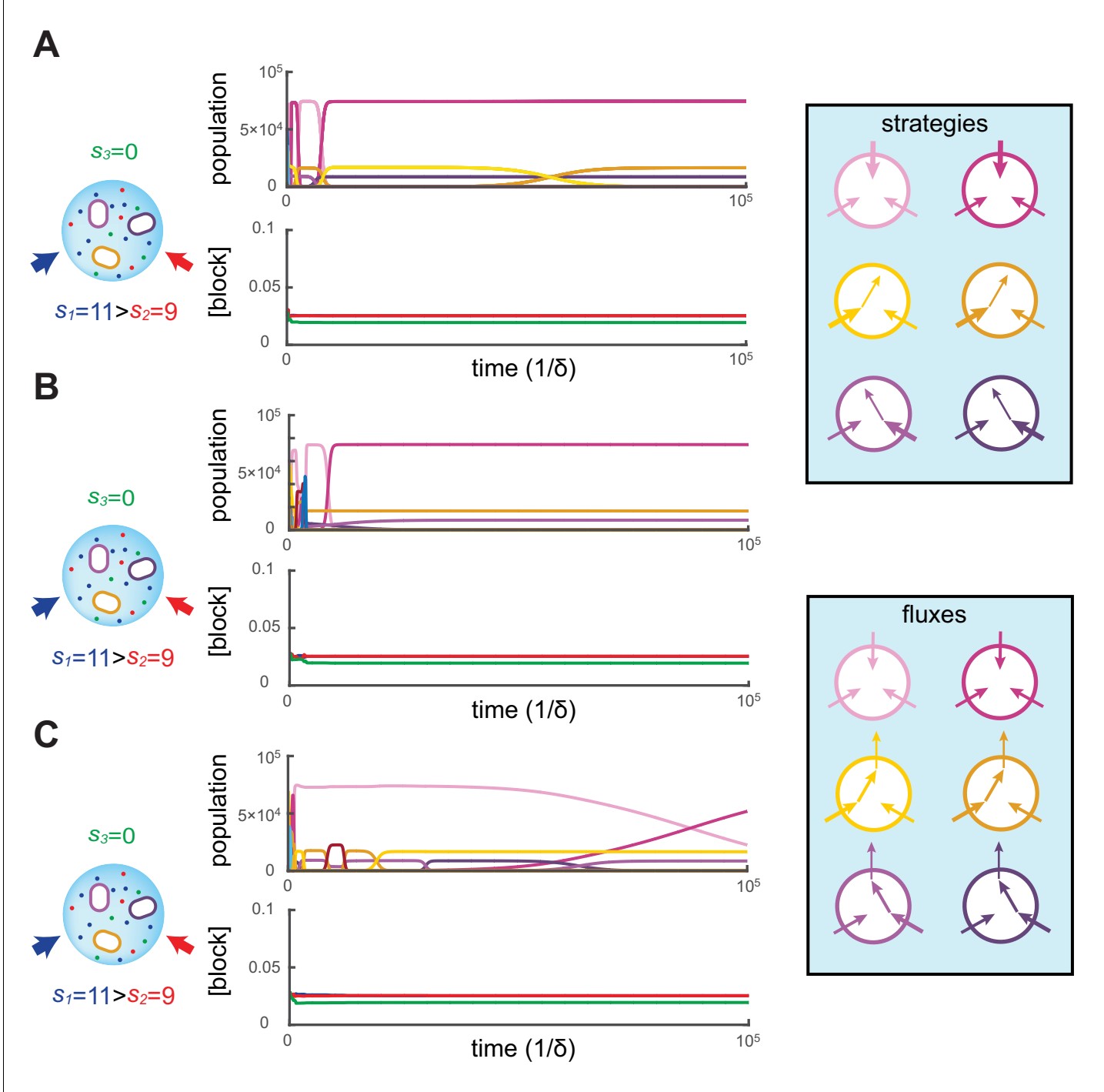

**Figure 2.** Simulated competitive dynamics. In all panels, the left schematic indicates supply rates, the central plot shows an example of competitive population dynamics, and the right diagram depicts the strategies and their internal building-block per-cell fluxes. The initial strategies and the newly introduced strategies were randomly generated with a different seed in **A**, **B**, and **C**, but for the same supply rates $s_1 = 11, s_2 = 9, s_3 = 0$. In each simulation, the external building-block concentrations quickly converge toward the same specific values, with $c_1^{ext} = c_2^{ext} > c_3^{ext}$. Also in each case, the simulation converges to the coexistence of the same three strategies. The dominant type with a population $\sim 75,000$ has a pure-importer strategy, the two other types convert one of the most abundant blocks (either block 1 or block 2) into block 3 and the strategy converting the most abundantly provided block has the larger population ($\sim 16,500$ vs. $\sim 8,500$). New cell types only manage to invade the already present bacterial population if they are 'fitter' versions of these three specific metabolic types. In particular, a 'fitter' strategy invades the overall population by replacing the already present strategy of the same metabolic type (curve crossings), with little effect on the other metabolic types or on the external building-block concentrations.

of cell types nears carrying capacity. At this point, the external building-block concentrations plummet to low values for which the growth rate of each cell type approximates the fixed death rate, i.e. $g_\sigma \approx \delta$. This is when metabolic competition begins in earnest. From this point on, in each simulation, the external building-block concentrations tend to the same stationary values, with the two externally provided building blocks 1 and 2 stabilizing at numerically identical values and building block 3 stabilizing at a lower concentration.

As external building-block concentrations approach their stationary values, virtually all newly introduced metabolic variants quickly become extinct because randomly chosen cell types are unable to compete with those already present. As a result, new cell types are continually introduced, for a total of more than $10^9$ different cell types during each simulation. Among these introduced metabolic variants, only a cell type that improves on an already present (nearly optimal) one can invade, and displace the existing cell type. In particular, for each displacement event, we find that the invading and invaded cell types employ almost identical distributions of enzymes and always belong to the same metabolic class. Moreover, during displacement events, the external building-block concentrations are only marginally perturbed, while the overall population of the invaded and invading strategy (e.g. the sum of crossing curves) is nearly constant. This behavior indicates convergence toward a stationary state with fixed building-block availability and with fixed populations of cell types. We confirmed the generality of this convergent behavior with additional simulations over a broad range of different supply rates.

## Cell types achieve optimal growth at stationary state

Our numerical simulations indicate that competitive population dynamics subjected to continual invasion leads to the emergence of stationary states. To show that these stationary states emerge independently of how metabolic variants are introduced, we developed an iterative optimization algorithm that yields stationary states without relying on random sampling of cell types. Specifically, the algorithm iterates two optimization steps: First, given distinct cell types $\sigma$, the algorithm implements a Newton-Raphson scheme to compute the steady-state external building-block concentrations $c_i^{\text{ext}}$ and the steady-state populations $n_\sigma$ within a relative precision $10^{-11}$. Second, given external building-block concentrations $c_i^{\text{ext}}$, the algorithm adapts a gradient-based constrained-optimization algorithm (**Wächter and Biegler, 2006**) to compute the strategies of the cell types $\sigma'$ which locally optimize cellular growth rate. Provided the growth rate of a locally optimal cell type $\sigma'$ exceeds the previous steady-state growth rate by a relative difference $g_{\sigma'}/\delta - 1 > \epsilon = 10^{-9}$, we initiate a new iteration with a set of starting cell types made of the union of the surviving cell types of the previous iteration $\{\sigma | n_\sigma \geq 1\}$ and the newly optimized cell type $\{\sigma' | g_{\sigma'} > \delta\}$ with $n_{\sigma'} = 1$. The algorithm halts when no cell type $\sigma'$ can grow at a rate that exceeds the steady-state growth rate $\delta$ by more than a relative difference $\epsilon$, thereby yielding theoretically optimal cell types with high accuracy. By design of the algorithm, the cell types present in the stationary state achieve the optimal growth rate allowed by the external building-block availabilities.

Crucially, for fixed supply rates, this optimization algorithm yields steady-state cell types that are virtually identical to the final cell types obtained via simulations. Moreover, additional simulations show that when present among the initial types, optimal steady-state cell types resist invasion by the more than $10^9$ random metabolic variants introduced over the course of $10^5$ generations. To demonstrate that simulations effectively converge toward optimal stationary states, we define the 'relative fitness' of a cell type $\sigma$ as the ratio $g_\sigma(c_1^{\text{ext}}, \ldots, c_p^{\text{ext}}) / \max_{\sigma'} g_{\sigma'}(c_1^{\text{ext}}, \ldots, c_p^{\text{ext}})$, where $\sigma'$ denotes a theoretically optimal cell type. **Figure 3** shows the evolution of cell-type relative fitnesses during simulations of competitive dynamics. When metabolic competition begins in earnest, all cell types have relative fitnesses smaller than one, indicating that optimal cell types would grow faster at identical external building-block concentrations. Ensuing displacement events by fitter cell types lead to an overall increase in the relative fitness of the surviving cell types. As a result, the relative fitness of all surviving cell types approaches one, demonstrating the convergence toward a stationary population of optimal cell types. Consequently, the final steady-state cell types resist invasion because no metabolic variant can outgrow them.

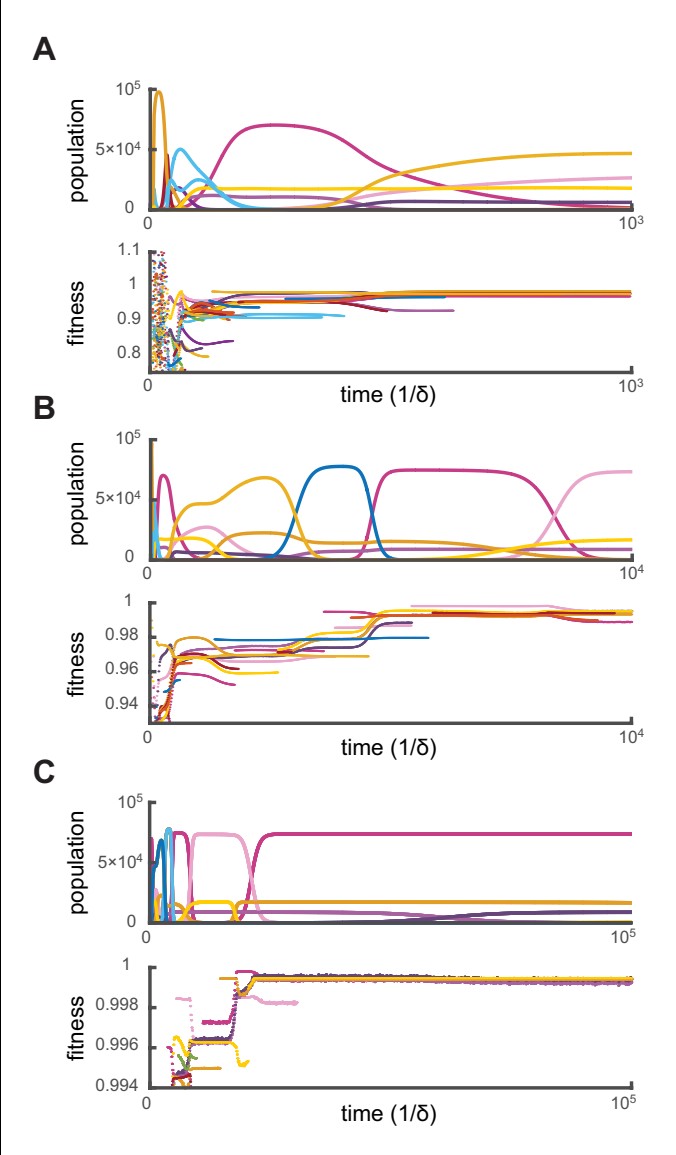

**Figure 3.** Relative fitness during competitive dynamics. In all panels, the top plot shows the same competitive population dynamics on different time scales for $s_1 = 11, s_2 = 9, s_3 = 0$, while the bottom plot shows the corresponding normalized fitness. The normalized fitness of a cell-type $\sigma$ is defined as the ratio $g_\sigma / \max g_{\sigma'}$, where $\sigma'$ are the theoretical steady-state cell types computed via iterative optimization. (A) Before the overall population reaches capacity, cell types can have a fitness larger than one as the external building-block concentrations are substantially higher than their steady-state values. Once competition begins in earnest, all cell types present have a fitness smaller than one, indicating that an optimal cell type would outcompete any present cell for identical external building-block concentrations. (B) On longer timescales, competition between increasingly fit cell types leads to the transient coexistence of 3 cell types with the same metabolic strategies as in *Figure 2*. Note that invasions by fitter cell types yield displacement events that increase the fitness of the surviving cell types. (C) On even longer timescales, the fitnesses of surviving cell types converge to one, showing that the final strategies obtained via competitive population dynamics achieve the optimal growth rate.

## Distinct optimal cell types emerge as consortia

At steady supply, our numerical simulations together with numerical optimization reveal that competitive population dynamics leads to the stable emergence of optimal cell types. The metabolic strategies of these optimal cell types exhibit network structures that are directly related to external

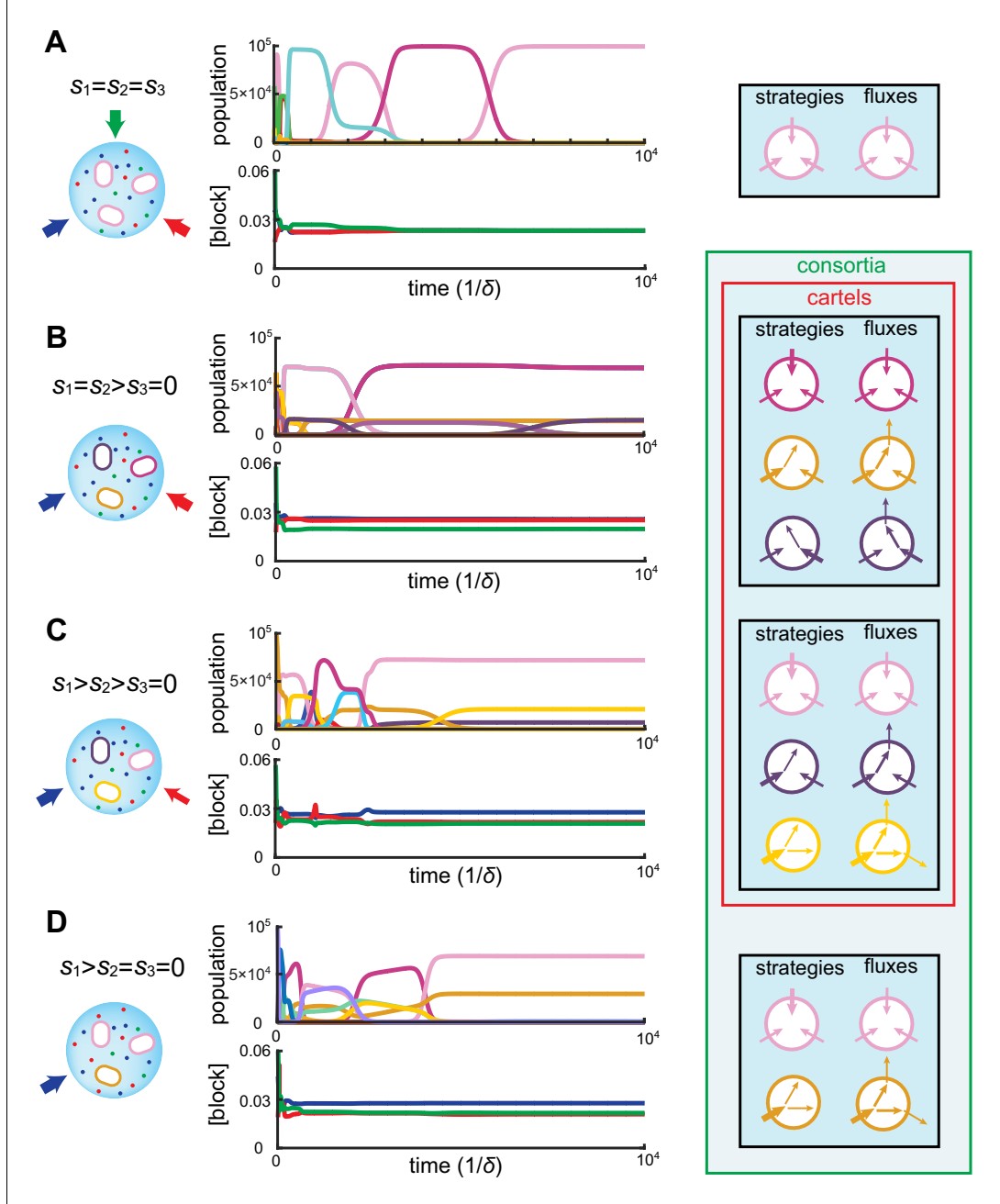

**Figure 4.** Simulated competitive dynamics. In all panels, the left schematic indicates supply rates, the central plot shows an example of competitive population dynamics, and the right diagram depicts the final strategies and their internal building-block per-cell fluxes. (A) If the building blocks are supplied with equal stoichiometry $s_1 = s_2 = s_3$, metabolic strategies that carry out conversions are wasteful and a single pure-importer cell type prevails. (B) If two building blocks are supplied with equal stoichiometry, e.g. $s_1 = s_2 > 0$ and $s_3 = 0$, three cell types can coexist: two 'symmetric' types using supplied blocks as a precursor for block 3, which accumulates externally due to passive leakage and release upon cell death, and, if $c_3^{\mathrm{ext}}$ is large enough, a third pure-importer type. (C) For large enough imbalance in the supply of building blocks 1 and 2, e.g. $s_1 > s_2 > s_3 = 0$, three distinct cell types can coexist: a pure-converter type imports block 1 and converts blocks 2 and 3; if $c_2^{\mathrm{ext}}$ is large enough, a mixed type emerges, importing blocks 1 and 2, and converting 1 to 3; and, if $c_3^{\mathrm{ext}}$ is large enough, a pure-importer type. (D) If only one building block is supplied, e.g. $s_1 > s_2 = s_3 = 0$, two strategies coexists: a pure-converter type releases blocks 2 and 3, which can lead to the emergence of a pure-importer type. The external building-block concentrations fluctuate, albeit only slightly later in the simulations, due to the invasion by and extinction of metabolic variants.

building-block availability. In fact, the building-block supply determines whether many distinct cell types can be jointly optimal, i.e. whether a consortium emerges. For instance, as shown in **Figure 4A**, if the building blocks are supplied with equal stoichiometry ($s_1 = s_2 = s_3$), the one survivor is a pure-importer strategy that imports each building block. Because this pure-importer strategy is the single most efficient cell type when building blocks are equally abundant, no other cell type can coexist with it and so no consortium emerges. However, when building blocks are supplied with a different stoichiometry, (e.g. with $s_3 = 0$ in **Figure 4B–D**), the pure-importer strategy coexists with other cell types that produce the non-supplied building block by conversion, thereby forming consortia of optimal cell types. The stability and optimality of these consortia is a collective property: No pure-importer strategy can survive without the converting cell types, and without a pure-importer strategy, there is a wasteful external accumulation of converted blocks.

Can the same metabolic consortia emerge for different building block supplies? To answer this question, observe in **Figure 2** that for supply rates $s_1 = 11, s_2 = 9, s_3 = 0$, competition leads to a stationary state where the concentrations of both supplied building blocks are equal. Correspondingly, the optimal cell types have symmetric strategies with respect to the usage of block 1 and block 2. However, the cell type that converts the most abundantly supplied block is more numerous than its symmetric counterpart, allowing for a symmetric steady state despite asymmetrical supply. For symmetric supply $s_1 = s_2 = 10, s_3 = 0$, **Figure 4B** shows that competition yields the same steady-state external concentrations, but with an equal population of each symmetric converter type. Thus, exactly the same consortium of cell types can emerge for different supply conditions, with different population counts but reaching the same external building-block concentrations. By definition, such consortia are cartels. **Figure 4** shows the different consortia that can emerge for representative supply conditions, highlighting which ones are cartels. Both numerical simulations and optimizations confirm that microbial cartels emerge for a large range of supply conditions.

## Analytical results

In this section, we mathematically elucidate the emergence of microbial consortia at steady state and characterize the benefit of the division of labor in these consortia. Our analysis exploits the demonstrated convergence of competitive population dynamics toward a stationary state, which allows us to analytically derive the metabolic strategies of optimal cell types. The benefit of division of labor among these optimal cell types follows geometrical considerations in the space of stationary states.

First, we exploit arguments from transport-network theory to systematically identify the metabolic classes of optimal cell types at steady state. Second, we elucidate the structure of microbial consortia by establishing which metabolic classes can be jointly optimal within a consortium. Third, we characterize the benefit of division of labor showing that consortia can act as cartels, whereby population dynamics pins down resource concentrations at values for which no other strategy can invade.

### Consortia cell types belong to optimal metabolic classes

A metabolic class is defined as the set of strategies that utilize the same enzymes, i.e. for which a particular subset of enzymes satisfies $\alpha_i > 0$ and $\kappa_{ji} > 0$. In total, there are $p$ import enzymes and $p(p-1)$ interconversion enzymes, for a total of $p^2$ enzymes. Thus, in principle, there are at most $2^{p^2}$ metabolic classes according to whether or not each type of enzyme is present ($\alpha_i > 0$ or $\kappa_{ij} > 0$). However, our simulations suggest that at steady state, the cell types that form consortia and achieve optimal growth belong to very specific metabolic classes: these 'optimal' classes utilize only a few, non-redundant metabolic processes (that is, many $\alpha_i$ and $\kappa_{ji}$ are zero).

Can we specify the network structures of optimal metabolic classes using rigorous optimization principles? Exploiting the linearity of metabolic fluxes, we adapt arguments from transport-network theory (**Bohn et al., 2007**) to achieve this goal for an arbitrary number of building blocks (see **Figure 5** and Appendix 3). Our approach consists in gradually reducing the number of candidate metabolic classes by showing that some classes $\mathcal{M}$ cannot contain an optimal strategy. Specifically, we consider a representative strategy $\sigma$ in $\mathcal{M}$ with enzyme budget $E = \sum_i \alpha_i + \sum_{ij} \kappa_{ij}$ at arbitrary external concentrations $c_i^{\text{ext}}$. For the same external concentrations $c_i^{\text{ext}}$, we show that one can always find a strategy $\sigma'$ from another metabolic class $\mathcal{M}'$ that achieves the same internal concentrations using a smaller enzyme budget $E' = \sum_i \alpha_i' + \sum_{ij} \kappa_{ij}' < E$. As the existence of a more 'economical' strategy $\sigma'$

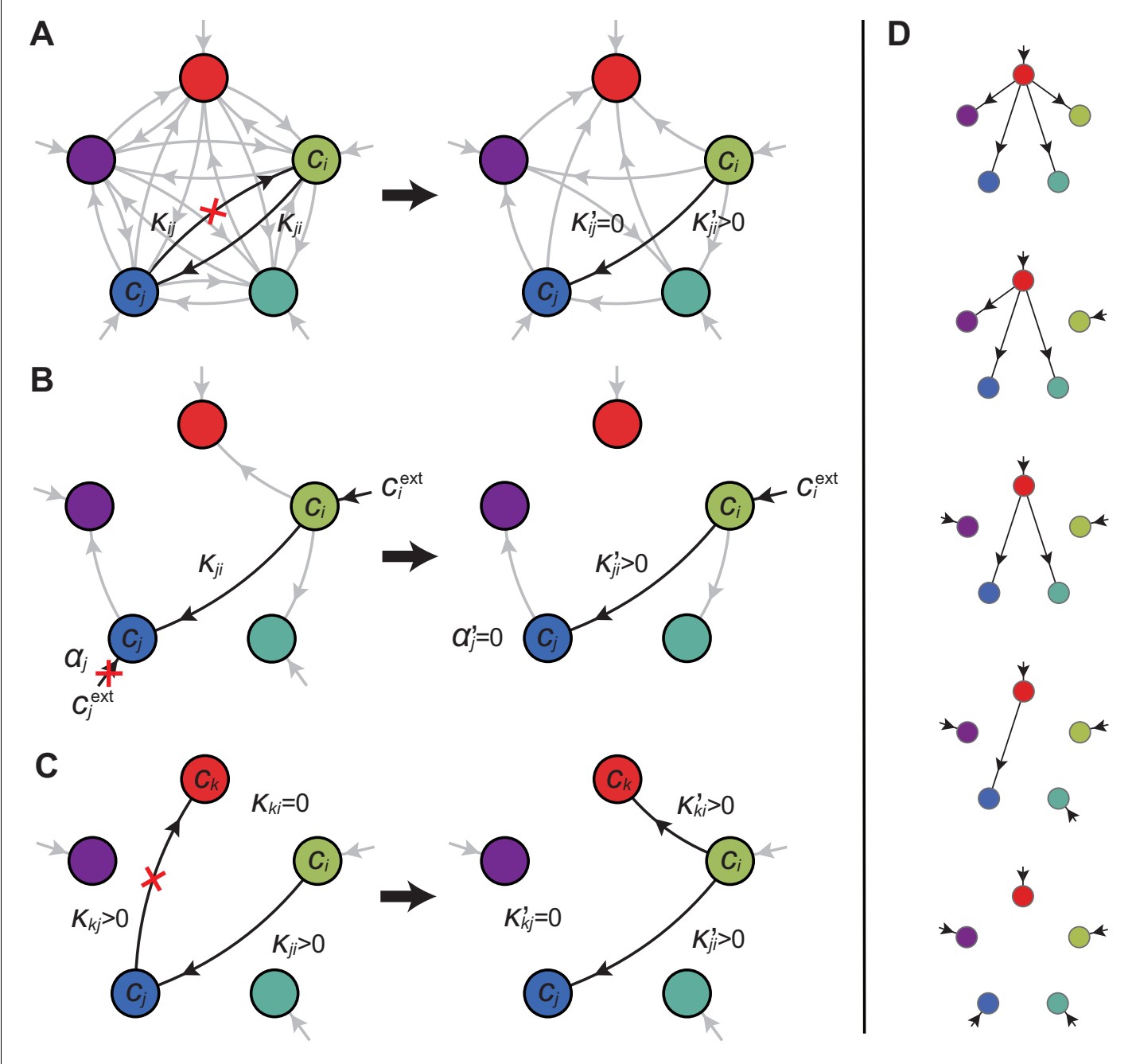

**Figure 5.** Optimal metabolic classes. A metabolic class is defined by the set of enzymes for which $\alpha_i > 0$ and $\kappa_{ji} > 0$. If a metabolic class is optimal, i.e. achieves the fastest growth rate, no other metabolic class can achieve the same growth rate with a lower enzyme budget. **(A)** Optimal metabolic classes cannot have topological 2-cycles. If cell type $\sigma$ (left) is such that the net conversion flux from block $i$ to block $j$ is positive, i.e. $\kappa_{ji}c_i > \kappa_{ij}c_j$, a cell type $\sigma'$ (right) that only differs from $\sigma$ by $\kappa'_{ij} = 0$ and $\kappa'_{ji} = \kappa_{ji} - \kappa_{ij}c_j/c_i$ achieves the same growth rate as $\sigma$ but more economically. More generally, optimal metabolic classes have no topological cycles, i.e. the graphs of their metabolic networks have a tree structure. **(B)** Optimal metabolic classes use a single precursor for each converted building block. If cell type $\sigma$ (left) accumulates block $j$ by import and by conversion from block $i$, there is always a more economical strategy $\sigma'$ (right) for which either $\alpha_j = 0$ or $\kappa_{ji} = 0$. **(C)** Optimal metabolic classes convert building blocks in the minimum number of steps. If cell type $\sigma$ (left) accumulates block $k$ via a 2-step conversion from block $i$, there is always a more economical strategy $\sigma'$ (right) that converts block $i$ directly into block $k$. **(D)** Optimal metabolic classes can only have a single tree of direct conversion(s).

contradicts the optimality of metabolic class $\mathcal{M}$, we can restrict our consideration to metabolic classes other than $\mathcal{M}$.

Using the above approach, we show that optimal metabolic networks process building blocks via non-overlapping trees of conversions (*Figure 5A*), each tree originating from an imported building block (*Figure 5B*), and each converted building block being obtained via the minimum number of conversions (*Figure 5C*). Intuitively, these properties ensure the minimization of waste (loss of building blocks via passive transport) during metabolic processing. Moreover, we show that optimal networks use a single building-block resource as precursor for conversions, i.e. there is at most one tree of conversions (*Figure 5D*). Thus, at steady state, requiring that a metabolic class is optimal, i.e. contains the fastest growing cell type, strongly constrains the graph of its metabolic network. These constrained graphs can be fully characterized and enumerated for $p$ building blocks: there are $p$ distinct graphs, each utilizing $p$ distinct enzymes, which defines a total of $1 + p(2^{p-1} - 1)$ metabolic classes after accounting for building-block permutations.

## Structure of metabolic classes in consortia

To find the composition of consortia, we must identify the enzyme distributions $\{\alpha_i, \kappa_{ji}\}$ within a metabolic class that yield the fastest growth for fixed external building-block concentrations. Knowing analytically the optimal enzyme distributions in each metabolic class allows us to characterize the structure of consortia at steady state via the maximum growth rate as a function of external building-block concentrations,

$$G(c_1^{\text{ext}}, \ldots, c_p^{\text{ext}}) = \max_{\sigma} g_{\sigma}(c_1^{\text{ext}}, \ldots, c_p^{\text{ext}}). \tag{7}$$

At competitive stationary state, the maximum growth rate must equal the death rate $\delta$ by *Equation (6)*. Otherwise, either there is a cell type such that $g_{\sigma} - \delta > 0$, yielding a diverging population $n_{\sigma}$, or $g_{\sigma} - \delta < 0$ for all cell types, yielding a vanishing microbial population. Thus, solving $G(c_1^{\text{ext}}, \ldots, c_p^{\text{ext}}) = \delta$ determines the set of steady-state external concentrations $c_1^{\star}, \ldots, c_p^{\star}$ for which an optimal strategy $\sigma^{\star}$ is present. By virtue of its optimality, the strategy $\sigma^{\star}$ achieves the fastest possible growth rate and is non-invadable at steady state. Consortia emerge for external building-block concentrations for which there is more than one optimal strategy, i.e. when there are distinct strategies $\sigma^{\star}$ for which the maximum-growth function is attained: $g_{\sigma^{\star}}(c_1^{\star}, \ldots, c_p^{\star}) = G(c_1^{\star}, \ldots, c_p^{\star}) = \delta$.

Obtaining analytical expressions for optimal enzyme distributions proves intractable for a nonlinear growth-rate function such as *Equation (1)*. However, optimal distributions can be obtained analytically for the minimum model $g(c_1, \ldots, c_p) = \gamma \min(c_1, \ldots, c_p)$, which is closely related to *Equation (1)* (see Appendix 4). In *Figure 6*, we represent the corresponding set of external building-block concentrations compatible with steady states, together with the associated optimal metabolic classes. For $p$ building blocks (see Appendix 5), we find that there exist $p!$ microbial cartels, each with $p$ distinct cell types for well-ordered external concentrations, e.g. $c_1^{\text{ext}} > c_2^{\text{ext}} > \ldots > c_p^{\text{ext}}$. In such cartels, cell type 1 converts building block 1 into the $p - 1$ other building blocks, cell type 2 converts building block 1 into the $p - 2$ least abundant building blocks and imports building block 2, and so forth, and cell type $p$ has a pure-importer strategy. We also find that for degenerate ordering with $q - 1$ equalities, e.g. $c_1^{\text{ext}} = \ldots = c_q^{\text{ext}} > c_{q-1}^{\text{ext}} > \ldots > c_p^{\text{ext}}$, there exist $(p - q)! C_p^q$ microbial cartels with $1 + q(p - q)$ distinct cell types. In such cartels, cell type $q'$, $1 \leq q' \leq q$ imports all blocks $1, \ldots, q$ but only uses block $q'$ as a precursor for blocks $j > q$, cell type $q''$, $q < q'' \leq 2q$ imports all blocks $1, \ldots, q + 1$ but only uses block $q'' - q + 1$ as a precursor for blocks $j > q + 1$, and so forth, and cell type $1 + q(p - q)$ has a pure-importer strategy. Moreover, we find that cartels that share $p - 1$ metabolic classes are joined by continuous paths in the space of external concentrations over which these $p - 1$ shared metabolic classes remain jointly optimal. Such paths define a graph which characterizes the topological structure of cartels in relation to changes in external building-block concentrations (see Appendix 6). Importantly, our analysis shows that the above cartels emerge with the same graph structure for all growth-rate functions satisfying $g(c_1, \ldots, c_p) \geq \gamma \min(c_1, \ldots, c_p)$ for some $\gamma > 0$ and having diminishing returns (quasi-concave property), which includes *Equation (1)*.

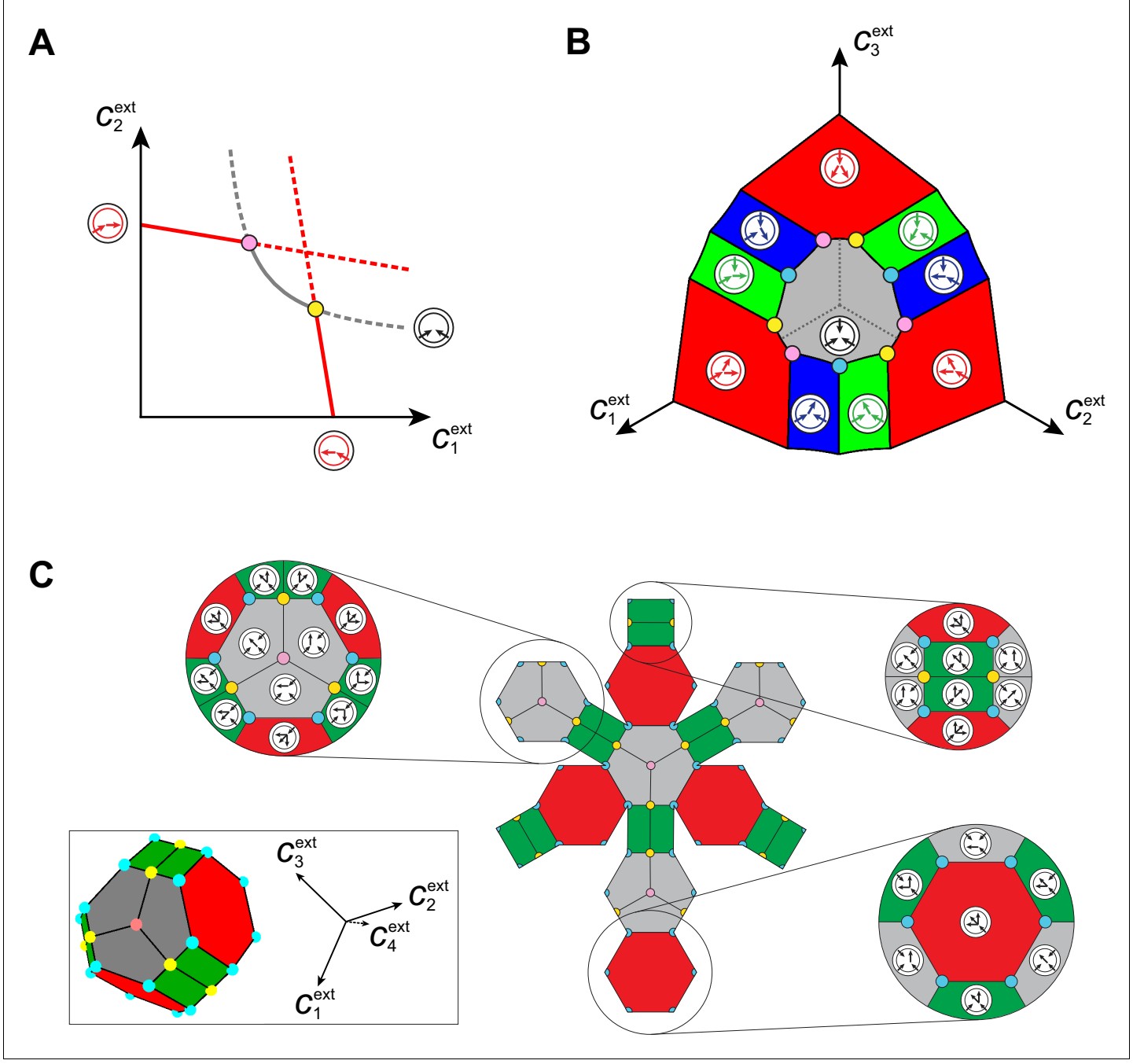

**Figure 6.** Emergence of microbial cartels at steady state. For large enough supply rates, population dynamics drive the external building-block concentrations towards steady-state values $c_1^\star, \ldots, c_p^\star$ that satisfy growth rate equals death rate, $G(c_1^\star, \ldots, c_p^\star) = \delta$. Consortia emerge at concentrations for which distinct metabolic classes are jointly optimal. Cartels are consortia with at least $p$ distinct metabolic classes. (A) For $p = 2$, a pure-converter strategy is optimal on each of the red curves, while a pure-importer strategy is optimal on the grey curve. Cartels with two distinct cell types exist at the intersection of the grey curve and a red curve. (B) For $p = 3$, a pure-converter strategy is optimal on the red patches, mixed strategies are optimal on the blue and green patches, while a pure-importer strategy is optimal on the grey patch. There are two types of cartels at the intersection of 3 patches: 6 distinct cartels with well-ordered external concentrations (yellow and pink), e.g. $c_1^{ext} > c_2^{ext} > c_3^{ext}$, and 3 distinct cartels with degenerate external concentration ordering (cyan), e.g. $c_1^{ext} = c_2^{ext} > c_3^{ext}$. (C) Graph structure of microbial cartels for $p = 4$ building blocks. As cartels can be labelled by ordering of resource availability, their graph structure is closely related to permutohedron solids, such as the truncated octahedron for $p = 4$ (inset: the interior of the truncated octahedron for $p = 4$ corresponds to the grey patch shown in (B) for $p = 3$) In addition to the metabolic types shown, each cartel includes a pure-importer strategy, so that blue and pink cartels have 4 distinct types while yellow cartels have 5 distinct types. In all panels, the circular arrow diagrams depict the metabolic strategies present.

## Relevance of microbial cartels

Microbial cartels only exist for specific external building-block concentrations (*cf.* the intersection points in *Figure 6A and B*). *Can competitive population dynamics lead to these cartels for generic supply conditions*? To answer this question, we compute the set of supply conditions compatible with the emergence of a cartel. We label a microbial cartel $\Sigma^\star$ by its associated external concentrations $c_1^\star, \ldots, c_p^\star$, which satisfy a specific (possibly degenerate) order relation. At concentrations $c_1^\star, \ldots, c_p^\star$, cartel cell types $\sigma \in \Sigma^\star$ jointly achieve the optimal growth rate and are therefore the only surviving cell types. The per-cell fluxes $\phi_{i,\sigma}^\star$ experienced by these cell types take fixed values that can be obtained via *Equation (5)*. Then, the resulting flux-balance equations for extracellular building blocks,

$$s_i(\{n_\sigma\}) = \mu c_i^\star + \frac{v}{V - Nv} \sum_{\sigma \in \Sigma^\star} n_\sigma \phi_{i,\sigma}^\star, \tag{8}$$

yield the supply rates as a function of the populations $n_\sigma > 0$, $\sigma \in \Sigma^\star$. In fact, *Equation (8)* defines the sector of supply rates compatible with the existence of the cartel $\Sigma^\star$ as a $p$-dimensional cone. Crucially, although $c_1^\star, \ldots, c_p^\star$ specify isolated points in the space of external concentrations, the cartel sectors have finite measure in the space of supply rates, showing that cartels can arise for generic conditions.

In *Figure 7*, we illustrate the supply sectors associated with cartels. What is the relation between supply sectors and steady-state external concentrations? To answer this question, we consider in *Figure 7A–E* the case of $p = 2$ building blocks and first focus on supply rates for which the optimal cell type is either a pure converter (white dots in *Figure 7A–B*) or a pure importer (white dots in *Figure 7C–D*). With only one cell type present and fixed $c_1^{\text{ext}}$ and $c_2^{\text{ext}}$, varying $n_\sigma$ in *Equation (8)* defines a half-line in the supply plane (cf. *Figure 7B* and *Figure 7D*). This half-line originates from the point $(\mu c_1^{\text{ext}}, \mu c_2^{\text{ext}})$, which are the supply rates that first support a nonzero population $n_\sigma$ for steady-state concentrations $c_1^{\text{ext}}$ and $c_2^{\text{ext}}$. Thus, all supply half-lines originate from a surface that is identical to the steady-state external concentrations, simply rescaled by the external building block leakage rate $\mu$. Changing the supply rates transverse to such a half-line yields different steady-state concentrations $c_1^{\text{ext}}$ and $c_2^{\text{ext}}$. In particular, one can increase the supply of one block until another optimal type can invade, i.e. until one reaches the cartel-specific concentrations $c_1^\star$ and $c_2^\star$ (pink dots in *Figure 7A* and *Figure 7C*). At the corresponding point $(\mu c_1^\star, \mu c_2^\star)$ in the supply space, the pure-converter half-line and the pure-importer half-line have distinct direction vectors, i.e. co-optimal cell types $\sigma$ have distinct per-cell fluxes $\phi_\sigma$: e.g. for $s_1 > s_2$, a pure-converter $\sigma$ consumes block 1 to produce and leak block 2, i.e. $\phi_{1,\sigma} > 0$ and $\phi_{2,\sigma} < 0$. By contrast, a pure importer $\sigma$ consumes all building blocks according to the biomass stoichiometry, i.e. $\phi_{1,\sigma} = \phi_{2,\sigma}$. For each point $(\mu c_1^\star, \mu c_2^\star)$ in the supply space, these distinct per-cell fluxes $\phi_\sigma$ define conic regions where a pure importer can invade a pure-converter population and a pure converter can invade a pure-importer population, i.e. where a cartel is stable. These cones are therefore cartel supply sectors (pink and yellow sectors in *Figure 7E*).

The above argument can be generalized for $p > 2$ building blocks by considering the metabolic fluxes of optimal cell types (cf. *Figure 7F* for $p = 3$). For all values of $p$, we find that supply sectors associated with cartels define non-overlapping cones (see Appendix 7). Moreover cones associated with two connected cartels, i.e. cartels that share at least $p - 1$ metabolic classes, have parallel facets in the limit of large budget $E \gg \beta$. As a consequence, at a fixed overall rate of building-block supply $s = s_1 + \ldots + s_p$, the fraction of supply conditions for which no cartel arises becomes negligible with increasing overall supply rate $s$. For instance, for large rate $s$, every building block has to be supplied at exactly the same rate for a single pure-importer strategy to dominate rather than a cartel. Therefore, for very generic conditions, a cartel will arise and drive the external building-block concentrations toward cartel-specific values, thereby precluding invasion by any other metabolic strategy. This ability to eliminate competition is reminiscent of the role of cartels in human economies, motivating the name 'cartels' for stable microbial consortia that include at least $p$ distinct metabolic strategies.

At supply conditions for which no cartel arises, an optimal cell type or a consortium of cell types dominates at steady state but these cell types cannot control external resource availability. Indeed, a consortium that is not a cartel cannot compensate for changes in supply conditions via population

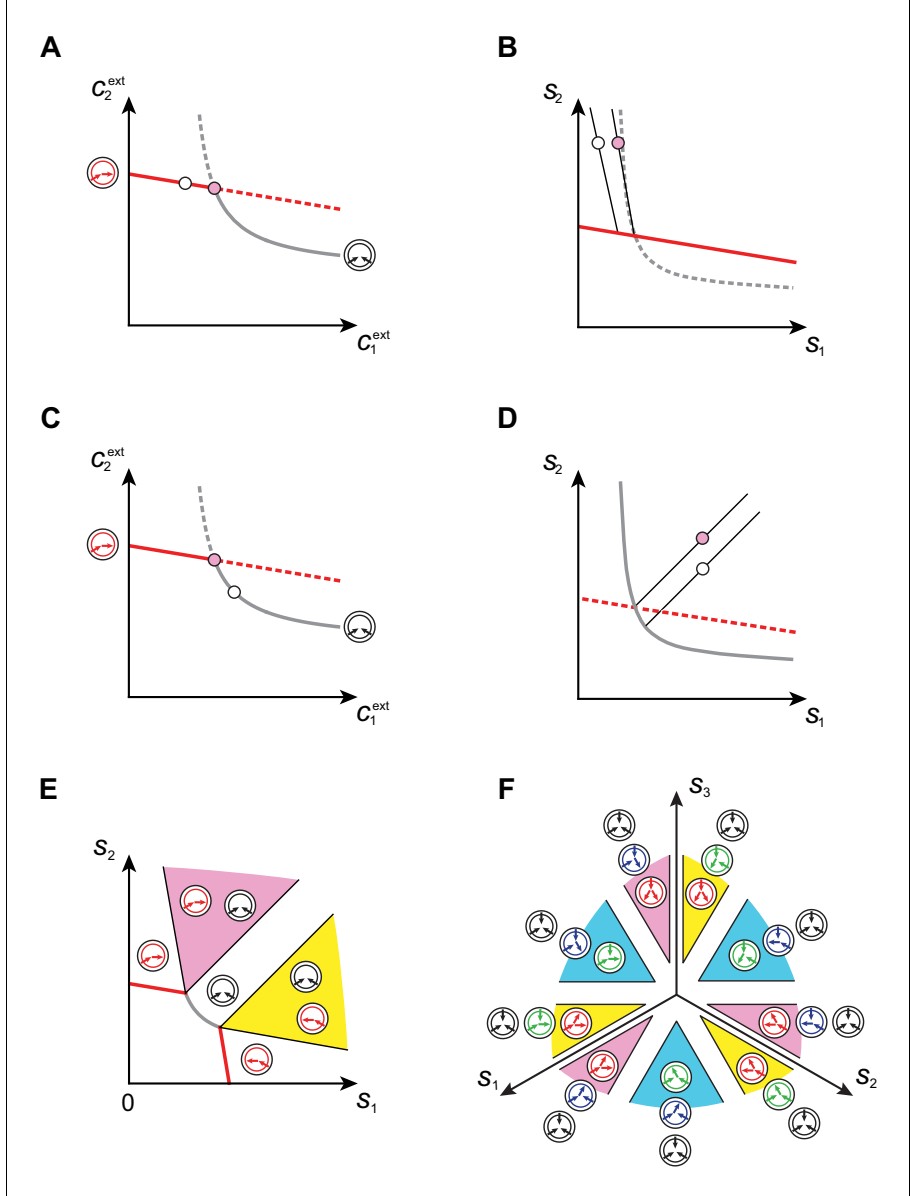

**Figure 7.** Supply sectors associated with microbial cartels. (A–B): For steady-state concentrations $c_1^{ext} < c_2^{ext}$, only a converter strategy can survive (white dot in A). The corresponding set of supply rates $s_1$ and $s_2$ lie on a line (labelled by a white dot in B). Increasing the supply rate $s_1$ causes concentration $c_1^{ext}$ to increase, until a pure-importer strategy can survive at $c_1^\star$ and $c_2^\star$ (pink dot in A). Any further increase of $s_1$ no longer affects $c_1^\star$ and $c_2^\star$ and is solely dedicated to biomass growth. (C–D): For steady-state concentrations $c_1^{ext} \simeq c_2^{ext}$, only a pure-importer strategy can survive (white dot in C). The corresponding set of supply rates $s_1$ and $s_2$ lie on a line (labelled by a white dot in D). Increasing the supply rate $s_2$ causes concentration $c_2^{ext}$ to increase, until a converter strategy can survive at $c_1^\star$ and $c_2^\star$ (pink dot in C). Any further increase of $s_2$ no longer affects $c_1^\star$ and $c_2^\star$ and is solely dedicated to biomass growth. (E–F): Supply conditions compatible with the emergence of a cartel for (E) $p = 2$ and (F) $p = 3$. The set of supply rates for which cartels can arise define non-overlapping polyhedral $p$-dimensional cones, with parallel or diverging faces between neighboring cartels, i.e. cartels that share $p - 1$ metabolic classes. For $p = 2$, the boundaries of the pink cartel supply sector correspond to the lines labelled by pink dots in (B) and (D). Outside of these cones, only fewer than $p$ strategies can survive.

dynamics. In particular, simply multiplicatively increasing the building-block supply augments the steady-state biomass of a consortium but also modifies steady-state resource availabilities, and therefore the distributions of enzymes that optimally exploit these resources. In other words, for consortia that are not cartels, optimal metabolic strategies must be fine-tuned to specific supply conditions.

By contrast, within a cartel supply-sector, any increase of the building-block supply is entirely directed toward the cartel's growth of biomass. Remarkably, it appears that microbial cartels automatically achieve maximum carrying capacity, i.e. they optimally exploit the resource supply. At steady state, the total number of cells $N$ is related to supply rates $s_i$ via the overall conservation of building blocks by

$$N = \frac{V}{v}\left(1 + \frac{p\delta(1-f)}{\sum_i s_i - \mu \sum_i c_i^{\text{ext}}}\right)^{-1},\tag{9}$$

which implies that maximizing biomass yield at fixed supply rates $s_i$ amounts to minimizing the overall external building-block concentrations $\sum_i c_i^{\text{ext}}$. In **Figure 8**, for $p = 2$ building blocks, we use the preceding equivalence to show that in each cartel sector, no consortium can yield a larger steady-state biomass than the supply-specific cartel. For a homogeneous growth-rate function such as **Equation (1)**, this result generalizes to arbitrary $p$ if we conjecture that (i) in a cartel supply sector, the cartel's metabolic classes can invade any other consortium, and that (ii) the maximum-growth-rate function associated with a given metabolic class has the property of diminishing returns (see Appendix 7). Intuitively, conjecture (i) means that the emergence of a cartel does not depend on the history of appearance of distinct metabolic classes and conjecture (ii) means that beating diminishing returns requires a switch of metabolic classes. Together, these conjectures ensure that adding a new metabolic class when possible implies a decrease in the total abundance of building blocks, i.e. a better use of resources. Because a better use of resources is equivalent to a steady-state biomass

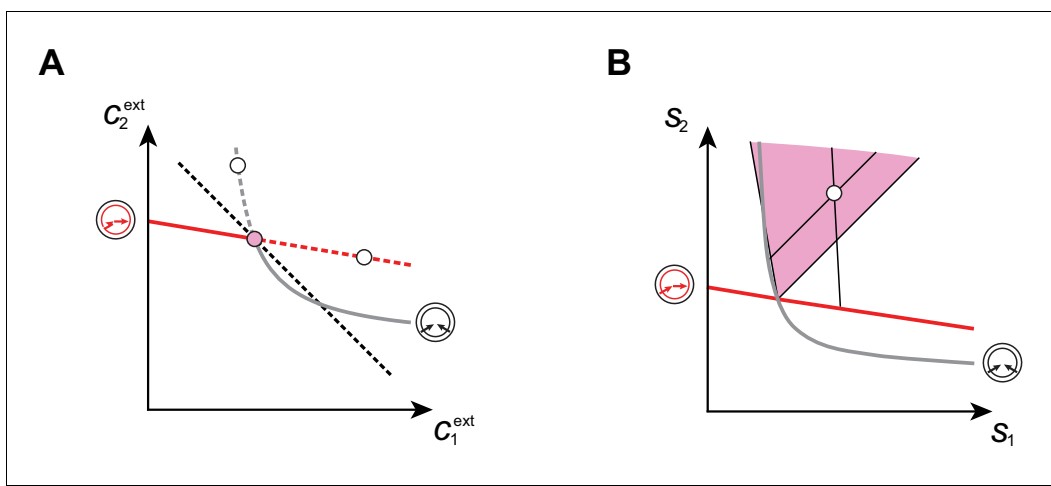

**Figure 8.** Cartels yield optimal biomass at steady supply. Steady-state concentrations $c_1^{\text{ext}}$ and $c_2^{\text{ext}}$ that satisfy $c_1^{\text{ext}} + c_2^{\text{ext}} > c_1^\star + c_2^\star$ (above the dashed black line in **A**) imply smaller biomass yields than achieved by the microbial cartel that exists for $c_1^\star$ and $c_2^\star$ (pink dot in **A**). The supply sector associated with the cartel defines a cone (pink region in **B**). For given supply rates in the cartel supply sector (white dot in **B**), the black lines represent the supply sectors of the pure-importer strategy and of the pure-converter strategy that are optimal when present alone (as opposed to being in a cartel). The intersection of these non-cartel supply sectors (black lines in **B**) with the steady-state curves (red and grey curves in **B**) define concentrations $c_1^{\text{ext}}$ and $c_2^{\text{ext}}$ for which $c_1^{\text{ext}} + c_2^{\text{ext}} > c_1^\star + c_2^\star$ (white dots in **A**). This result is generic for any supply rates in the cartel supply sector; thus a pure-importer or a pure-converter strategy alone leads to steady-state concentrations with a smaller biomass yield than the cartel. We did not take into account the other converter strategy, belonging to the other cartel (yellow sector in **Figure 7E**), since this cartel can only be optimal for $c_1^{\text{ext}} > c_2^{\text{ext}}$, which never happens for $s_2 > s_1$.

increase (by virtue of building-block conservation), this establishes that competing microbes achieve the global collective optimum by forming cartels.

## Discussion

Building on a physical model for metabolic fluxes, which importantly includes a finite enzyme budget, we showed that competitive population dynamics leads to the emergence of microbial cartels. Cartels are defined as consortia of at least as many distinct optimal cell types — each with a fixed metabolic strategy — as there are shared resources. Constituent cell types of a cartel are optimal because they achieve the fastest possible growth rate for that cartel's self-regulated external resource concentrations, and can therefore resist invasion by metabolic competitors. Within this framework, the benefit of metabolic diversity to the participating cells stems from the ability of cartels to control resource availability (*Sanchez and Gore, 2013*). In particular, cartels maintain fixed external resource concentrations by adjusting their populations to compensate for slow changes in supply. For steady supply, the emergence of microbial cartels at long times is independent of the specific dynamics of competition, which may reflect invasions by existing competitors and/or the appearance of mutant strategies. Strikingly, our results support the conclusion that such cartels of competing microbes achieve the optimal collective carrying capacity, as if led by an 'invisible hand' to efficiently exploit the resources (*Smith, 1776*).

### Assumptions and scope of the model

For simplicity, we assumed linear metabolic fluxes and uniform enzymatic rates, production costs, and building-block stoichiometries. However, the emergence of optimal cartels does not rely on these assumptions. Even allowing for fluxes that are nonlinear (e.g. Michaelis-Menten) with respect to building-block concentrations, microbes must utilize their enzymes in the linear regime to be metabolically optimal: Because resources are depleted by competitive growth between metabolic classes, fluxes mediated by saturated enzymes do not limit growth. Cells can improve their growth rate by reallocating their enzyme budget from saturated enzymes to the unsaturated enzymes mediating growth-limiting linear fluxes. Moreover, independent of rates, production costs, and stoichiometries, optimal metabolic types must consist of non-overlapping trees of conversions. Indeed, the optimality of such metabolic networks, obtained from transport-network theory, only requires the linearity of metabolic fluxes with respect to enzyme concentrations. As a result, optimal metabolic types, as well as cartels, can still be enumerated. Interestingly, we discovered that distinct cartels can arise for very similar external building-block availabilities, and cartels can even merge under special conditions. In an extended model that includes fluctuations, e.g. in enzyme expression (*Wang and Zhang, 2011*; *Kiviet et al., 2014*), we expect 'ghosts' of these neighboring cartels associated with similar resource availabilities to persist against the background of the dominant cartel. As our primary concern is the emergence of a division of labor, we consider only relatively large populations of cells for which we can neglect stochastic population fluctuations.

### Realistic metabolic networks

What relevance might our results have for real metabolic networks? Microbes regulate metabolic processes via complex networks with, e.g., multistep reaction chains and metabolic branch points (*Almaas et al., 2004*). However, there is evidence of optimal partitioning of enzymes in these real networks: microbes produce components of multiprotein complexes in precise proportion to their stoichiometry, whereas they produce components of functional modules differentially according to their hierarchical role (*Li et al., 2014*). Recent experimental studies have revealed that optimal metabolic flux partitioning is an operating principle for resource allocation in the proteome economy of the cell (*Hui et al., 2015*; *Hermsen et al., 2015*). Provided optimality considerations apply to real metabolic networks, the approach we have taken can provide insight into flux partitioning and division of labor in microbial communities. For instance, we expect that for a group of interconvertible resources that are collectively growth limiting, the expressed metabolic network should have the topological properties discussed above — no 'futile cycles' and no 'convergent pathways'. Such predictions are not at odds with the existence of well-known metabolic cycles such as the TCA cycle and the GOGAT cycle because these cycles are not futile but rather are energy yielding or assimilatory, respectively. Our predictions apply directly to irreversible conversion processes, e.g via chains

of reactions with committed steps, as well as to reversible chains of reactions, for which the only cycles in optimal metabolic networks are two-cycles due to reversibility. The overall acyclic nature of anabolic fluxes can be tested experimentally by measuring reaction fluxes in metabolic networks, e.g. using isotope tracers and mass spectrometry.

## Spatial and temporal heterogeneities

Abiotic and biotic processes controlling resource turnover in nutrient reservoirs, such as the ocean or soil sediments, operate on many different temporal and spatial scales (*Braswell et al., 1997*; *Whitman et al., 1998*). In our framework, steady but spatially inhomogeneous supply of diffusive building blocks should lead to the tiling of space by locally dominant cartels. Because of our model cells' ability to shape their environment, we expect sharp transitions between neighboring tiles, consisting of cartels that differ by a single metabolic class. We expect spatial tiling to emerge in real microbial communities growing in inhomogeneous conditions, e.g. in a gradostat with spatially structured nutrient supply (*Lovitt and Wimpenny, 1981*). In such spatial communities, the detection of well-delimited patches of resource availabilities, with specific nutrient ratios, would be evidence of spatial tiling by microbial cartels.

The spatial structure of microbial communities may also reflect the extracellular division of labor. Extracellular division of labor involves metabolic pathways with obligatory external reactions, i.e. with enzymes that are public goods. In a homogeneous environment, 'cheating' cell types that do not produce the public good are always at an advantage and their introduction causes the collapse of the entire population. In our framework, we expect producer cartels to spatially segregate from neighboring non-producer cartels (*Drescher et al., 2014*).

Temporally varying supply can also be addressed within our framework. For supply fluctuations on long timescales $\gg 1/\delta$ (the lifetime of a cell), the population dynamics within cartels keeps resource levels fixed, whereas fluctuations on short timescales $\ll 1/\delta$ are self-averaging. In practice, slow supply fluctuations can arise due to seasonal biogeochemical cycles (*Schoener, 2011*), while fast supply fluctuations can arise from the transient biomass release upon cell death (*Yoshida et al., 2003*). The effect of supply fluctuations occurring on timescales $\sim 1/\delta$, which includes day-night cycles, is more complex. Transport-network theory predicts that fluctuating resource conditions select for networks with metabolic cycles, whose structures depend on the statistics of the driving fluctuations (*Katifori et al., 2010*; *Corson, 2010*). Characterizing the benefit of cycles in such networks may well reveal new optimization principles that underlie the microbial metabolic diversity.

Microbes also adjust to fluctuating conditions by switching their metabolic type via gene regulation instead of relying on population dynamics. Within our framework, to consistently implement the optimal mix of metabolic strategies, the role of sensing and regulation is then primarily to determine the relevant 'supply sector' by assessing the relative abundance of various resources. Thus, in principle, division of labor within a single species can lead to cartels with distinct metabolic strategies associated with distinct phenotypic states. However, the persistence of cartels requires the coexistence of all cartel strategies, which within a single species could be facilitated by cell-to-cell communication (quorum sensing). We therefore anticipate that extension of our analysis to fluctuating supply conditions may provide insight into the design principles underlying regulation and signaling in microbial communities.

## Acknowledgements

This work was supported by the DARPA Biochronicity program under Grant D12AP00025, by the National Institutes of Health under Grant R01 GM082938, and by the National Science Foundation under Grant NSF PHY11-25915. We thank Bonnie Bassler, William Bialek, Curt Callan, and Simon Levin for many insightful discussions.

# Additional information

## Funding

| Funder | Grant reference number | Author |
|---|---|---|
| Defense Advanced Research Projects Agency | D12AP00025 | Thibaud Taillefumier<br>Ned S Wingreen |
| National Institutes of Health | R01 GM082938 | Thibaud Taillefumier<br>Yigal Meir<br>Ned S Wingreen |
| National Science Foundation | NSF PHY11-25915 | Anna Posfai<br>Ned S Wingreen |

The funders had no role in study design, data collection and interpretation, or the decision to submit the work for publication.

## Author contributions

TT, Conceptualization, Software, Formal analysis, Investigation, Methodology, Writing—original draft, Writing—review and editing; AP, Formal analysis, Validation; YM, Supervision; NSW, Conceptualization, Supervision, Validation, Writing—original draft, Writing—review and editing

## Author ORCIDs

Thibaud Taillefumier, http://orcid.org/0000-0003-3538-6882

Ned S Wingreen, http://orcid.org/0000-0001-7384-2821

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

## Appendix 1:Resource-limited growth-model

In this section, starting from simple biological considerations, we develop a physical model for the growth of microbes that produce biomass from a finite set of resources. In section "Nonlinear biomass-production model", we present a simple model for cellular biomass production, e.g. protein synthesis, from intracellularly available building blocks, e.g. amino acids. In section "Internal flux-balance equations", we specify the balance of fluxes associated with the import and processing of building blocks prior to biomass production. In section "Positive monotonic cell growth", we show that biomass production and the intracellular building-block concentrations are increasing functions of the influxes of building blocks.

### Nonlinear biomass-production model

We denote by $g$ the growth rate averaged over a population of cells, which has the units of inverse time, and by $r$ the rate of biomass production , which has units of biomass per second. Both these rates are related by a simple relation. To see this, observe that neglecting the fluctuations in cellular biomass, e.g. due to individual cell divisions, justifies the adoption of a continuous model where the cellular biomass $b$, i.e. the concentration of protein, is a characteristic of cells that remains constant. Then, taking into account the dilution by cell growth, the stationarity of cellular biomass, $\dot{b} = r - gb = 0$, directly implies a proportionality relation: $g = r/b$. In particular, setting the concentration units of our model to satisfy $b = 1$ allows us to formally equate the growth rate $g$ and the rate of biomass production $r$.

Importantly, we consider that biomass is produced by incorporating $p$ building blocks into biologically functional units. Accordingly, we denote by $b_i$, $1 \leq i \leq p$, the concentration of building block $i$ in cellular biomass. In the context of protein synthesis, $b_i$ corresponds to the cellular concentration of amino acid $i$ incorporated in proteins. Achieving a rate of growth $g$ requires cells to consume building block $i$ at rate $b_i g$ to satisfy their fixed building-block requirements. As the consumption of building blocks, and thus biomass production, is limited by the internal availability of each building block, we model the rate of growth as a function $g(c_1, \ldots, c_p)$ of the internal concentrations of building blocks $c_i$, $1 \leq i \leq p$. We assume that $g(c_1, \ldots, c_p) = 0$ whenever any building block is lacking, i.e. when $\min(c_1, \ldots, c_p) = 0$ and that $\partial_{c_i} g > 0$ so that elevated internal concentrations of building blocks promote cell growth. We expect the above assumptions to generally hold for building blocks that are absolutely required for cell growth.

A relevant choice for such a rate function is to take $g(c_1, \ldots, c_p)$ proportional to the harmonic mean of the internal concentrations

$$g \propto \left( \frac{b_1}{c_1} + \ldots + \frac{b_p}{c_p} \right)^{-1}. \tag{A1}$$

Such a rate function adequately models protein biosynthesis, where biomass production results from the sequential incorporation of amino acids, each amino acid being incorporated at a rate proportional to its concentration. The time to produce a unit of biomass is the sum of the incorporation time of each building block $i$, which is proportional the relative requirement for each building block $b_i$, in turn yielding the overall rate of growth as the harmonic mean *Equation (A1)*. Tellingly, the harmonic mean function has the quasi-concave property, whereby its level sets $g(c_1, \ldots, c_n) > r$ are convex sets. This mathematical property can be intuitively interpreted in terms of diminishing marginal utility: the benefit of increasing the availability of a resource decreases with its abundance. In the interest of

analytical tractability, we will also consider the 'minimum' model, where only the scarcest resource is rate limiting, leading to a growth-rate function

$$g \propto \min\left(\frac{c_1}{b_1}, \ldots, \frac{c_p}{b_p}\right). \tag{A2}$$

Such a rate function corresponds to the extreme case when increasing the abundance of resources that are not the most rate limiting is unproductive. Observe that, to fully define a growth-rate function $g$, we need to specify a kinetic rate constant relating $g$ to the right-hand sides of *Equation (A1)* and *Equation (A2)*.

## Internal flux-balance equations

In order to produce biomass, a cell can import a building block $i$ from the external medium, or produce it by internally converting another building block $j$. As each import/conversion process is enzymatically controlled, a metabolic strategy is specified by the allocation of the enzyme budget to these various processes. For simplicity, we adopt the enzyme budget constraint:

$$\sum_i \alpha_i + \sum_{i \neq j} \kappa_{ij} \leq E, \tag{A3}$$

where the constants $\alpha_i$ and $\kappa_{ij}$ are enzymatic activities, assumed proportional to the concentrations of enzymes, associated with the import of resource $i$ ($\alpha_i$) and the conversion of $j$ into $i$ ($\kappa_{ij}$), and where $E$ denotes the total enzyme budget.

We assume a separation of timescales between fast metabolic reactions and slow biomass production. As a result, we consider that over the timescale of cellular growth the internal concentrations reach their stationary state: $\dot{c}_i = 0$. Assuming that every enzymatic process operates in the linear regime, the internal building-block concentrations $c_i$ satisfy $p$ flux-balance equations, one for each building block:

$$(\beta + \alpha_i)c_i^{\text{ext}} - \left(g + \beta + \sum_{j \neq i} \kappa_{ji}\right)c_i + \sum_{j \neq i} \kappa_{ij}c_j - b_i g = 0, \tag{A4}$$

where $g$ is the cellular rate of growth, $c_i^{\text{ext}}$ is the external building-block concentration, and the flux $b_i g$ specifies the building-block consumption rate in units of concentration per second. The above set of flux-balance equations defines a network of metabolic reactions, where the only nonlinearity of the model is due to the growth-rate function $g$. For every building block, we model passive transport across the cell membrane via the leak rate $\beta$. Finally, in writing *Equation (A4)*, we made the assumption that the overwhelming majority of cellular building blocks are incorporated in the biomass, i.e. $b_i \geq c_i$. This biologically relevant approximation justifies neglecting the dilution rate $c_i g$ due to cell growth by comparison with $b_i g$, the consumption rate of building block $i$.

## Positive monotonic cell growth

Physically, we expect that for given external building-block concentrations $c_i^{\text{ext}}$, the set of internal flux-balance *Equation (A4)* specifies a unique set of positive internal concentrations $c_i$, and that the cellular growth is an increasing function of the resource influxes $(\beta + \alpha_i)c_i^{\text{ext}}$. Using the monotonic property of growth rate $g$ together with conservation of building-block

fluxes, it is indeed possible to show that our growth model exhibits the desired property of positive monotonic cell growth.

To demonstrate this point, it is convenient to consider the biomass production rate as a free parameter $\gamma$ (independent of the requirement that $\gamma = g(c_1, \ldots, c_p)$), in which case the system of **Equation (A4)** becomes linear, yielding internal concentrations $c_1(\gamma), \ldots, c_p(\gamma)$. In turn, the solutions to the nonlinear problem are obtained from the self-consistent relation $\gamma = g(c_1(\gamma), \ldots, c_p(\gamma))$, that states that the growth rate $\gamma$ is achievable by the parametrized internal concentrations $c_1(\gamma), \ldots, c_p(\gamma)$. These parametrized concentrations $c_1(\gamma), \ldots, c_p(\gamma)$ can be found as

$$\left(c_1(\gamma), \ldots, c_p(\gamma)\right) = K^{-1}\left((\beta + \alpha_1)c_1^{\text{ext}} - b_1\gamma, \ldots, (\beta + \alpha_p)c_p^{\text{ext}} - b_p\gamma\right) \tag{A5}$$

where $K$ is the matrix of enzymatic activities defined as

$$K_{ij} = \begin{cases} \beta + \sum_{k \neq i} \kappa_{ki} & \text{if } i = j \\ -\kappa_{ij} & \text{if } i \neq j \end{cases}. \tag{A6}$$

Because $K$ is strictly diagonally dominant, i.e. $K_{ii} > \sum_{j \neq i} |K_{ij}|$ for all $i$, with negative off diagonal coefficients, it is a monotone matrix, which means that all the coefficients of $K^{-1}$ are positive. Thus, the internal concentrations $c_1(\gamma), \ldots, c_p(\gamma)$ are decreasing functions of $\gamma$. Moreover, $(\beta + \alpha_i)c_i^{\text{ext}} - b_i\gamma$ is positive for small enough $\gamma \geq 0$ and negative for large enough $\gamma$. Thus, defining

$$\gamma_0 = \inf\left\{\gamma > 0 \,\middle|\, \min_i c_i(\gamma) = 0\right\}, \tag{A7}$$

the function $g(\gamma) = g(c_1(\gamma), \ldots, c_p(\gamma))$ is decreasing on $(0, \gamma_0)$, from $g(0) > 0$ to $g(\gamma_0) = 0$. In particular, the equation $g(\gamma) - \gamma = 0$ admits a unique positive solution $\gamma^\star$ in $(0, \gamma_0)$, which is the actual cellular growth rate ($\gamma^\star = g(\gamma^\star)$). Moreover, the corresponding internal concentrations $c_i(g^\star)$ are all positive. Indeed, suppose $c_i(\gamma^\star) < 0$, then there is $\gamma$, $0 < \gamma < \gamma^\star$, such that $c_i(\gamma) = 0$, implying $g(\gamma) - \gamma = -\gamma < 0$, which contradicts the uniqueness of $\gamma^\star$. This shows that there is a unique set of positive internal concentrations satisfying **Equation (A4)**. Importantly, this reasoning also shows that an increase in the influx of a resource leads to an increase in internal concentration, and therefore an increase in growth rate. For instance, if the influx $(\beta + \alpha_i)c_i^{\text{ext}}$ is increased by an amount $\epsilon > 0$, then the new function $g_\epsilon(\gamma)$ is strictly larger the original function $g(\gamma)$. Thus, $\gamma^\star$ satisfies $0 = g(\gamma^\star) - \gamma^\star < g_\epsilon(\gamma^\star) - \gamma^\star$, which implies that the growth rate associated with $g_\epsilon(\gamma)$, defined as the solution to $g_\epsilon(\gamma) - \gamma = 0$, is strictly larger than the growth rate $\gamma^\star$ associated with $g(\gamma)$.

# Appendix 2: Population dynamics of metabolic strategies

In this section, we establish a model for the population dynamics of competing metabolic strategies, e.g. cell types, in a microbial colony. In section "External flux-balance equations", we model the external metabolic fluxes at the population level, where different cell-types compete for steadily supplied building blocks. In section "Competitive growth of metabolic strategies", we model the growth of metabolically competing cell types, ensuring the overall conservation of building-block fluxes. In section "System of ordinary differential equations", we show that our model reduces to a set of coupled ordinary differential equations, which prescribe bounded population dynamics.

## External flux-balance equations

A metabolic strategy is an assignment $\{\alpha_{i,\sigma}, \kappa_{ij,\sigma}\}$ that satisfies the enzyme budget constraint **Equation (A3)**, thus defining a cell type $\sigma$. We consider that these different cell types $\sigma$ are growing in a homogeneous environment of volume $\Omega$. We denote the dimensionless population count of cell type $\sigma$ by $n_\sigma$ and the total population count of cells by $N = \sum_\sigma n_\sigma$. For simplicity, we also consider that every cell type has the same average lifetime. The inverse of this lifetime defines the cellular death rate $\delta$, which is independent of the cell type and assumed much slower than the timescales associated with metabolic processes such as diffusion, interconversion, and passive/active transport.

In the volume $\Omega$, we consider that the $p$ building blocks are steadily supplied with rate by volume $s_i$ and can be lost, e.g. via degradation or diffusion out of the volume at a rate $\mu$. In the absence of microbes, the change in external concentration of building block $i$ due to supply and loss obeys $\dot{c}_i^{\text{ext}} = s_i - \mu c_i^{\text{ext}}$, so that we simply have $c_i^{\text{ext}} = s_i/\mu$ at steady state. When present, microbes modifies this steady state via building-block intake during growth and via biomass release upon cell death. We denote by $f$ the fraction of biomass recycled upon cell death ($0 \leq f < 1$). Then, $n_\sigma$ cells of type $\sigma$ create a net flux $n_\sigma \phi_{i,\sigma}$ for each building block $i$, where the per-cell flux $\phi_{i,\sigma}$ equals the net individual metabolic intake of building block $i$ minus the rate of release of building block $i$ from the biomass of dying cells:

$$\phi_{i,\sigma} = (\beta + \alpha_{i,\sigma}) c_i^{\text{ext}} - \beta c_{i,\sigma} - \delta(fb_i + c_{i,\sigma}) \tag{A8}$$

$$\approx (\beta + \alpha_{i,\sigma}) c_i^{\text{ext}} - \beta c_{i,\sigma} - \delta fb_i. \tag{A9}$$

In theory, upon death, cells release the free internal building blocks $i$ that are present in the cell at the time of death, as well as a fraction $f$ of the building blocks $i$ that were incorporated in the cellular biomass. In practice, as cell growth sets the slow timescale of our model ($\delta \ll \beta$), we can neglect the fluxes due to the release of the internal pool of building blocks upon cell death.

Importantly, individual cell fluxes $\phi_{i,\sigma}$ contribute to changing the external concentration $c_i^{\text{ext}}$ via a geometric factor. The conservation of the number of building blocks determines this geometric factor to be $\omega/(\Omega - N\omega)$, the ratio of the average individual cellular volume $\omega$ and of the cell-free volume $\Omega - N\omega$. Accordingly, the smaller the fractional volume of a cell type, the less that cell type can change the shared external concentration via metabolic exchanges. Thus the total number of cells has to satisfy $N < \Omega/\omega$, where $\Omega/\omega$, the number of cells that would fill the volume $\Omega$, plays the role of a carrying capacity. Moreover, the change in the cell-free volume $\Omega - N\omega$ due to cell growth affects the change in external building-block concentration $c_i^{\text{ext}}$ via another geometric term. Indeed, denoting by $M_i$ the number of molecules of external building block $i$ in the cell-free volume, we have

$$\dot{c}_i^{\text{ext}} = \frac{d}{dt}\left(\frac{M_i}{\Omega - N\omega}\right) = \frac{\dot{M}_i}{\Omega - N\omega} + \frac{\dot{N}\omega}{\Omega - N\omega}c_i^{\text{ext}}. \tag{A10}$$

In the equation above, the first term of the right-hand side arises from the supply and loss of building blocks and from the fluxes created from the various cell types, while the last term is purely geometrical in nature and vanishes at steady state. Writing these terms explicitly, the stationary condition for the external building-block concentrations $c_i^{\text{ext}}$ on the timescale of cellular growth ($\dot{c}_i^{\text{ext}} = 0$) leads to $p$ external flux-balance equations that govern the external availability of building blocks:

$$0 = s_i - \left(\mu - \frac{\dot{N}\omega}{\Omega - N\omega}\right)c_i^{\text{ext}} - \frac{\omega}{\Omega - N\omega}\left(\sum_\sigma n_\sigma \phi_{i,\sigma}\right). \tag{A11}$$

The above equations shows that microbial growth affects the external availability of building blocks via both the metabolic fluxes the microbes create and the change in external cell-free volume. In the next section, we show that the latter geometric contribution is negligible when compared to the former.

## Competitive growth of metabolic strategies

In the previous section, we presented a model for the external metabolic fluxes in a homogeneous environment where populations of different cell-types compete for steadily supplied building blocks. When growing, cells compete with each other by depleting the pool of external building blocks: as the population $n_\sigma$ of cell type $\sigma$ grows, the flux of nutrients imported by these cells increases, thereby reducing the availability of the particular mix of nutrients they feed on. This competition for building blocks couples the growth of different microbial strategies $\sigma$ according to the population dynamics

$$\frac{\dot{n}_\sigma}{n_\sigma} = g_\sigma - \delta. \tag{A12}$$

When used in combination with the internal and external flux-balance **Equations (A4) and (A11)**, the above population dynamics allows us to justify the neglect of variations in external cell-free volume due to cell growth.

We justify the neglect of variations in external cell-free volume due to cell growth by analyzing overall building-block fluxes at the population level. The per-cell fluxes for cell-type $\sigma$ are

$$\phi_{i,\sigma} = (\beta + \alpha_{i,\sigma})c_i^{\text{ext}} - \beta c_{i,\sigma} - \delta f b_i. \tag{A13}$$

Then, summing the internal flux-balance **Equation (A4)** for every building block leads to the relation between the growth rate of a cell type $\sigma$ and its per-cell building-block fluxes:

$$\left(\sum_i b_i\right)g_\sigma = \sum_i \phi_{i,\sigma} + \delta f \sum_i b_i. \tag{A14}$$

In turn, summing the external flux-balance **Equation (A11)** for every building block leads to the overall conservation of building-block fluxes

$$\sum_i s_i - \left( \mu - \frac{\dot{N}\omega}{\Omega - N\omega} \right) \sum_i c_i^{\text{ext}} - \frac{\omega}{\Omega - N\omega} \left( \sum_i \sum_\sigma n_\sigma \phi_{i,\sigma} \right) = 0 . \tag{A15}$$

Using relation **Equation (A14)**, the above conservation of building-block fluxes can be rewritten as

$$\sum_i s_i - \left( \mu - \frac{\omega}{\Omega - N\omega} \dot{N} \right) \sum_i c_i^{\text{ext}} + \frac{\omega}{\Omega - N\omega} \sum_i b_i \sum_\sigma n_\sigma ( \delta f - g_\sigma ) = 0 . \tag{A16}$$

As the growth **Equation (A12)** imply that $\sum_\sigma n_\sigma g_\sigma = \dot{N} + \delta N$, upon substitution in **Equation (A16)**, we finally obtain the conservation of building-block fluxes as

$$\sum_i s_i - \mu \sum_i c_i^{\text{ext}} - \frac{\omega}{\Omega - N\omega} \left( \dot{N} \left( \sum_i b_i - \sum_i c_i^{\text{ext}} \right) + N\delta(1-f) \sum_i b_i \right) = 0 . \tag{A17}$$

We consider cell types for which, as a result of active import, internal building-block availability exceeds the external building-block availability, i.e. $c_i^{\text{ext}} < c_{i,\sigma} \ll b_i$. Thus, in **Equation (A17)**, the overall fluxes of building blocks required for growth vastly exceed the effect of variations in free volume due to cell growth: $\dot{N}\omega \sum_i b_i \gg \dot{N}\omega \sum_i c_i^{\text{ext}}$. This justifies neglecting the purely geometric term due to variations in free volume in the external flux-balance **Equation (A11)**.

## System of ordinary differential equations

On the timescale of cell growth, set by the average lifetime of a cell $\approx 1/\delta$, the internal and external concentrations can be considered as fast variables that have reached steady state. In particular, $\{c_i^{\text{ext}}\}$ and $\{c_{i,\sigma}\}$ satisfy the set of flux-balance **Equations (A4) and (A11)**. In Section "Positive monotonic cell growth", we have established that, given external concentrations $c_i^{\text{ext}}$, the internal concentrations $c_{i,\sigma}$ of a cell type $\sigma$ can be determined by solving the system of **Equation (A4)**, which specifies the functions $c_{i,\sigma}(c_1^{\text{ext}}, \dots, c_i^{\text{ext}})$. Then, the growth rate of cell type $\sigma$,

$$g_\sigma = g\big( c_{1,\sigma}(c_1^{\text{ext}}, \dots, c_p^{\text{ext}}), \dots, c_{p,\sigma}(c_1^{\text{ext}}, \dots, c_p^{\text{ext}}) \big) , \tag{A18}$$

only depends on the external concentrations of building blocks $c_i^{\text{ext}}$. Solving **Equation (A11)** yields the external concentrations $c_i^{\text{ext}}$ can be obtained as functions of $\{n_\sigma\}$, the population vector of cell types. Thus, **Equation (A12)** governing the growth of the competing metabolic strategies constitute a set of ordinary differential equations coupled via the external concentrations of building blocks, considered as $p$ auxiliary variables. In this section, we show that the corresponding population dynamics is bounded with (i) finite overall population count $N$ and (ii) finite positive external concentrations $c_i^{\text{ext}}$. We devote the rest of this section to show these two points.

i.  Neglecting the variations in free volume due to cell growth, the overall conservation of building blocks implies that the total cell population $N$ remains strictly below the carrying capacity $\Omega/\omega$, ensuring that our population dynamics model is well-posed. To see this, remark that, by positivity of the external building-block concentrations $c_i^{\text{ext}}$, we have

$$\dot{N} = \frac{\Omega - N\omega}{\omega} \frac{\sum_i s_i - \mu \sum_i c_i^{\text{ext}}}{\sum_i b_i} - N\delta(1-f)$$

$$\leq \frac{\Omega}{\omega} \frac{\sum_i s_i}{\sum_i b_i} - N\left(\delta(1-f) + \frac{\sum_i s_i}{\sum_i b_i}\right)$$

The above inequality implies that the total cell population is decreasing for any $N$ larger than

$$\frac{\Omega}{\omega}\left(1 + \frac{\delta(1-f)\sum_i b_i}{\sum_i s_i}\right)^{-1} < \frac{\Omega}{\omega}, \tag{A19}$$

and that, for any initial population satisfying $N(0) < \Omega/\omega$, the left-hand side of the above expression is an upper-bound of the long-term population dynamics.

ii.  Neglecting the variations in free volume due to cell growth, the external flux-balance *Equation (A11)* reads

$$s_i = \mu c_i^{\text{ext}} + \frac{\omega}{\Omega - N\omega}\sum_\sigma n_\sigma \phi_{i,\sigma}(c_1^{\text{ext}}, \dots, c_p^{\text{ext}}), \tag{A20}$$

where the per-cell building block fluxes are defined as in *Equation (A13)*. Summing the above equations for different building blocks, we obtain the overall conservation of building blocks as

$$\sum_i s_i = \mu \sum_i c_i^{\text{ext}} + \frac{\omega}{\Omega - N\omega}\sum_i b_i \sum_\sigma n_\sigma \left(g_\sigma(c_1^{\text{ext}}, \dots, c_p^{\text{ext}}) - \delta f\right). \tag{A21}$$

By positivity of the growth function for each cell types, i.e. $g_\sigma \geq 0$, it follows that

$$\sum_i s_i \geq \mu \sum_i c_i^{\text{ext}} - \frac{N\omega\delta f}{\Omega - N\omega}\sum_i b_i. \tag{A22}$$

In turn, the boundedness of the overall population of cell types $N$ implies the boundedness of the external building block concentrations $c_i^{\text{ext}}$ via

$$c_i^{\text{ext}} \leq \frac{1}{\mu}\left(\sum_i s_i + \frac{N\omega\delta}{\Omega - N\omega}\sum_i b_i\right). \tag{A23}$$

# Appendix 3: Optimization over network topologies

In this section, we explain how the optimization of microbial growth rate over the set of metabolic strategies allowed by the enzyme budget constraint determines the nature of microbial coexistence at steady state. In section "Maximum-growth-rate function", we introduce the maximum growth-rate function, which fully characterizes microbial coexistence at steady state. In section "Metabolic classes", we explain how to compute the maximum growth-rate function, by focusing on the analysis of a constrained number of metabolic types. In section "Network-theory analysis", we identify the metabolic types that are relevant to our optimization problem by identifying the allowed topological structures of their metabolic networks.

## Maximum-growth-rate function

For steady building-block supply, extensive numerical simulations suggest that the population dynamics of a finite number of metabolic strategies is globally convergent, i.e. for any initial conditions, the population vector $\{n_\sigma(t)\}$ tends to a limit $\{n_\sigma^\infty\}$, avoiding oscillatory or chaotic dynamics. Because the growth of certain cell types can benefit or impair the growth of other cell types, we cannot use standard population dynamics arguments to prove the global convergence of our model. However, we believe that global convergence holds and that two key features explain this simple asymptotic behavior. First, because of our assumption of a separation of timescales, the internal concentrations are effectively instantaneously determined by the external concentrations and no imbalance can build up, as would be required for oscillatory dynamics. Second, because of the strict conservation of building-block fluxes, each step toward the production of biomass entails some building-block waste, making the system dissipative.

Positive global convergence for steady building-block supply greatly simplifies the problem of determining which metabolic strategies survive competition to coexist on long timescales. To understand this point, we first need to introduce the maximum-growth-rate function $G$, defined as a function of steady-state external building-block concentrations by

$$G\left(c_1^{\mathrm{ext}},\ldots,c_p^{\mathrm{ext}}\right) = \max_\sigma g_\sigma\left(c_1^{\mathrm{ext}},\ldots,c_p^{\mathrm{ext}}\right). \tag{A24}$$

Notice that in the above definition strategies $\sigma$ belong to a compact continuous set determined by the budget constraint **Equation (A3)**, which justifies that $\sup_\sigma g_\sigma\left(c_1^{\mathrm{ext}},\ldots,c_p^{\mathrm{ext}}\right)$ is attained and is therefore a $\max$. The maximum growth-rate function $G$ entirely determines the long-time structure of coexisting strategies at stationary state. Indeed, the boundedness of the population dynamics imposes that survival at stationary state implies

$$G\left(c_1^{\mathrm{ext}},\ldots,c_n^{\mathrm{ext}}\right) \leq \delta, \tag{A25}$$

where $\delta$ is the cell death rate, which we take as a constant. Otherwise, there is a strategy such that $g_\sigma - \delta > 0$, yielding a diverging population $n_\sigma$. Moreover, if $G < \delta$, we have $g_\sigma - \delta < 0$ for each strategy, yielding a vanishing population. Thus, the steady-state external building-block concentrations $c_i^\star$ are specified as positive numbers satisfying

$$G\left(c_1^\star,\ldots,c_p^\star\right) = \delta. \tag{A26}$$

Moreover, for each set of admissible steady-state concentrations $c_i^\star$, the compatible set of coexisting metabolic strategies is given by

$$\{\sigma^\star\} = \arg\max_\sigma g_\sigma\left(c_1^\star, \ldots, c_p^\star\right) = \delta, \tag{A27}$$

where $\arg\max_\sigma g_\sigma$ denotes the set of strategies for which maximum growth is acheived. Relation *Equation (A27)* states that, at fixed steady-state external concentrations $c_1^\star, \ldots, c_p^\star$, the surviving strategies are those ones for which $g_\sigma\left(c_1^\star, \ldots, c_p^\star\right)$ is optimal. In particular, coexistence occurs when this maximum is attained for different strategies.

## Metabolic classes

To compute the maximum-growth-rate function, we need to optimize the growth rate $g_\sigma$ over the continuous set of admissible metabolic strategies for arbitrary fixed external conditions. This optimization can be carried out in two steps: First, by optimizing the growth rate over metabolic network topologies, one can identify a finite number of metabolic classes, defined as classes of strategies utilizing the same subset of transporters/enzymes, which are potentially optimal. Second, within these metabolic classes, one can characterize optimal strategies for fixed external concentrations. As a result, the maximum-growth-rate function $G(c_1^\star, \ldots, c_p^\star)$ can be computed as a maximum over a finite number of metabolic classes.

Formally, metabolic classes are defined as equivalence classes, whereby two strategies $\sigma$ and $\sigma'$ belong to the same metabolic class $\mathcal{M}$ if and only if $\alpha_i = 0 \Leftrightarrow \alpha_i' = 0$ and $\kappa_{ij} = 0 \Leftrightarrow \kappa_{ij}' = 0$, where $\alpha_i$, $\alpha_i'$, $\kappa_{ij}$, and $\kappa_{ij}'$ are enzyme activities. How many metabolic classes do we need to consider? There are $p$ import enzymes and $p(p-1)$ interconversion enzymes, for a total of $p^2$ enzymes. Thus, there are at most $2^{p^2}$ metabolic classes according to whether or not each type of enzyme is present ($\alpha_i > 0$ or $\kappa_{ij} > 0$). However, only a subset of metabolic classes are potentially optimal. For instance, the class for which $\alpha_i = \kappa_{ij} = 0$ for all $i, j$, is clearly not optimal. More tellingly, irrespective of the external availability of building blocks, numerical simulations show that optimal strategies generally belong to a very restricted set of metabolic classes. Next, we will identify this reduced set of metabolic classes using arguments inspired by analytic network theory.

Our strategy to discard a metabolic class $\mathcal{M}$, and thus reduce the number of relevant classes, is as follows. We will first consider a representative strategy $\sigma$ in $\mathcal{M}$ with budget $E = \sum_i \alpha_i + \sum_{ij} \kappa_{ij}$ at arbitrary external concentrations $c_i^{\text{ext}}$. The growth rate $g_\sigma$ of strategy $\sigma$ only depends on the internal concentrations $c_{i,\sigma}$. For the same external concentrations $c_i^{\text{ext}}$, we will show that one can always find a strategy $\sigma'$ from another metabolic class $\mathcal{M}'$ achieving the same internal concentrations but using a smaller budget $E' = \sum_i \alpha_i' + \sum_{ij} \kappa_{ij}' < E$. Then, by virtue of the positive monotony of our cell growth model, one can reallocate the saved budget $E - E'$ to building-block import and increase building-block influxes, thus yielding a strategy $\sigma'$ that outperforms $\sigma$ for the same budget constraint. Since the existence of a more 'economical' strategy $\sigma'$ contradicts the optimality of metabolic class $\mathcal{M}$, we can restrict our consideration to the metabolic classes other than $\mathcal{M}$. In the following, we will use the above procedure to show that optimal strategies belong to a very restricted set of metabolic classes.

## Network-theory analysis

Topologically, the most generic metabolic class is the one that utilizes every import and conversion enzyme, i.e. $\alpha_i, \kappa_{ij} > 0$ for all $i$ and $j$, as any other class can be obtained from it by setting some $\alpha_i$ or $\kappa_{ij}$ to zero. In this section, starting from the most generic metabolic class, we show that the assumption of optimal growth implies that many $\alpha_i$ and $\kappa_{ij}$ vanish, thereby restricting candidate topologies for optimal networks.

## Optimal networks are 'forests of trees'

Inspired by classic network-theory arguments, we show that at steady state the network of import and interconversion enzymes associated with optimal strategies is a directed forest of trees rooted in some or all of the external building blocks.

Consider a strategy $\sigma$ that has a cycle of order 2, e.g. enzymes interconverting building block $i$ into building block $j$ and building block $j$ into building block $i$:

$$\sigma : c_i \xrightarrow{\kappa_{ji}} c_j \quad \text{and} \quad c_j \xrightarrow{\kappa_{ij}} c_i \tag{A28}$$

Altering the enzymatic activities according to $\kappa_{ij} \leftarrow \kappa_{ij} + \delta\kappa_{ij} = \kappa'_{ij}$ and $\kappa_{ji} \leftarrow \kappa_{ji} + \delta\kappa_{ji} = \kappa'_{ji}$ with

$$\delta\kappa_{ji}\, c_i = \delta\kappa_{ij}\, c_j \,, \tag{A29}$$

leads to the same net flux between $i$ and $j$. As a result, the net building-block fluxes are preserved for the same internal concentrations $c_i$ and $c_j$. The corresponding altered enzyme budget is $E' = E + \delta E$ with

$$\delta E = \delta\kappa_{ji}\left(1 + \frac{c_i}{c_j}\right), \tag{A30}$$

showing that reducing the enzymatic activity $\kappa_{ji}$ leads to a smaller enzyme budget if $\kappa_{ij} \geq 0$. Thus, one can always find a more economical strategy $\sigma'$ that uses only one enzyme to perform an interconversion( the same reasoning applies to $\kappa_{ij}$). This shows that the network of interconversion reactions of optimal strategies has no cycle of order 2, so we can restrict our consideration to metabolic classes satisfying $\kappa_{ij}\kappa_{ji} = 0$ for all $i$ and $j$.

We can generalize the above argument to show that an optimal strategy cannot exhibit interconversion cycles of any order (see **Appendix 3—figure 1**). To see this, consider a interconversion network of a strategy $\sigma$ with no 2-cycles and suppose that its undirected graph has a 3-cycle, e.g. of the form

$$\sigma : c_i \xrightarrow{\kappa_{ji}} c_j \,, \quad c_j \xrightarrow{\kappa_{kj}} c_k \,, \quad c_i \xrightarrow{\kappa_{ki}} c_k \,. \tag{A31}$$

Then, altering the enzymatic activities according to

$$\delta\kappa_{ji}\, c_i = \delta\kappa_{kj}\, c_j = -\delta\kappa_{ki}\, c_i \,, \tag{A32}$$

leaves the net internal fluxes unchanged, and the altered strategy $\sigma'$ uses a budget $E' = E + \delta E$ with

$$\delta E = \delta\kappa_{ji}\left(\frac{c_i}{c_j}\right). \tag{A33}$$

Thus, for generic conditions, one can always form a more economical strategy $\sigma'$ by either reducing or increasing $\kappa_{ji}$, until one of the activities along the cycle becomes zero. This shows that the network of interconversion reactions of optimal strategies has no cycle of order 3, so that we can restrict ourselves to metabolic classes satisfying $\kappa_{ij}\kappa_{ji} = 0$ and $\kappa_{(ij)}\kappa_{(jk)}\kappa_{(ki)} = 0$ for all $i$, $j$, and $k$, where $\kappa_{(ij)} = \max(\kappa_{ij}, \kappa_{ji})$. The above argument directly generalizes to cycles of any order showing that, if a strategy is optimal, the undirected graph of its interconversion network has no cycles. Therefore, the undirected graph of an optimal interconversion

network is a 'forest of trees'. Observe that, in particular, this result implies that cycles are never optimal.

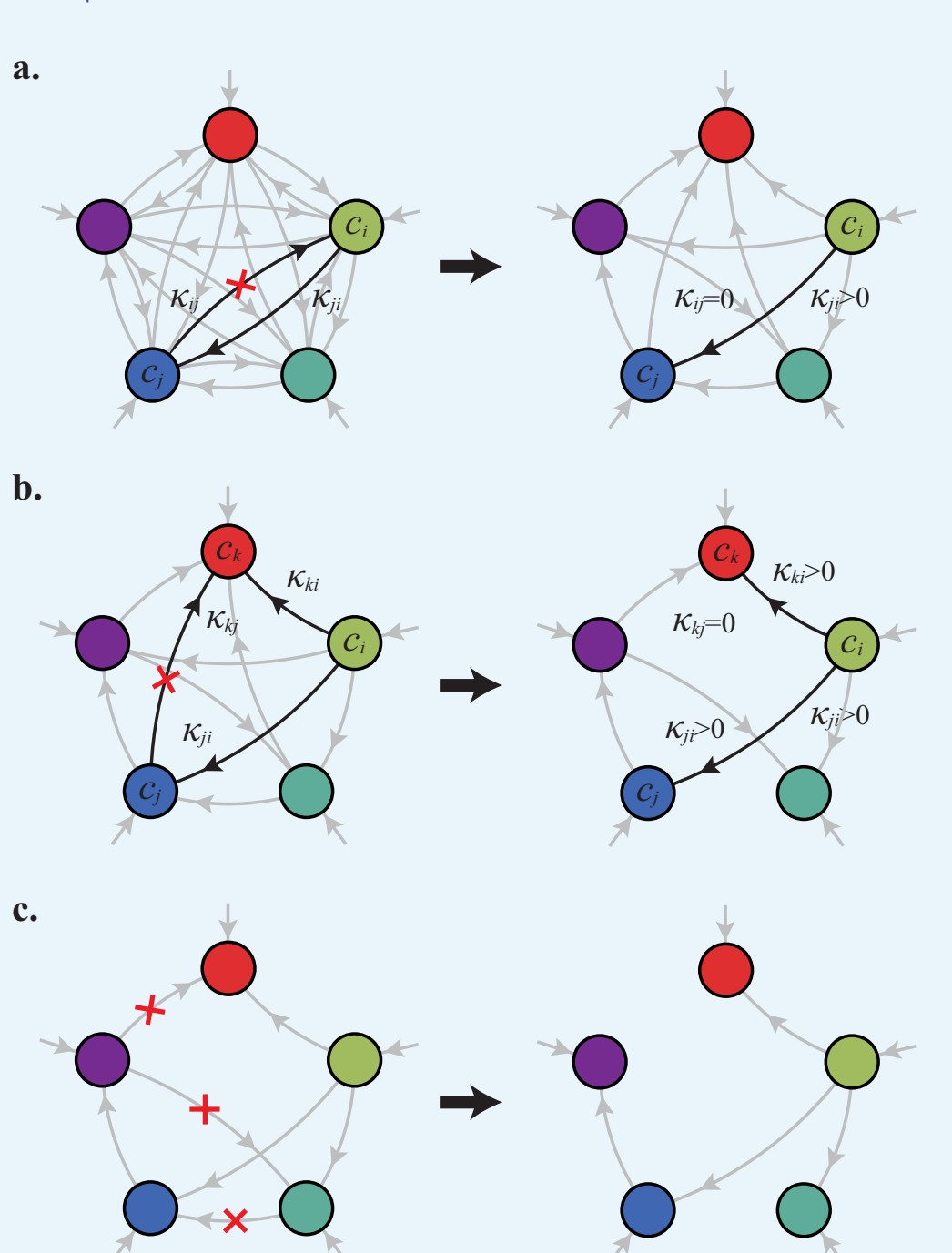

**Appendix 3—figure 1.** For fixed external concentrations, optimal networks are 'forests of trees'. (a) Optimal metabolic classes cannot have topological 2-cycles. If cell type $\sigma$ (left) is such that the net conversion flux from block $i$ to block $j$ is positive, i.e. $\kappa_{ji}c_i > \kappa_{ij}c_j$, a cell type $\sigma'$ (right) that only differs from $\sigma$ by $\kappa'_{ij} = 0$ and $\kappa'_{ji} = \kappa_{ji} - \kappa_{ij}c_j/c_i$ achieves the same growth rate as $\sigma$ but more economically. (b) Optimal metabolic classes cannot have topological 3-cycles. For cell type $\sigma$ (left) with a topological 3-cycle, a cell type $\sigma'$ (right) that differs from $\sigma$ by $\kappa'_{ki} \neq \kappa_{ki}$, $\kappa'_{ji} \neq \kappa_{ji}$ and $\kappa_{kj} = 0$, achieves the same growth rate as $\sigma$ but more economically. (c) More generally, optimal metabolic classes have no topological cycles, i.e. the graphs of their metabolic networks have a tree structure.

In the following, exploiting the same procedure as above, we show that optimal networks satisfy an additional topological property: building blocks are accumulated internally from a unique external source (see **Appendix 3—figure 2**). In graph theory, the defining property of a forest is that, given any two nodes, there is at most one path joining them. For our network of import and interconversion enzymes, this property implies that, given two external building-block concentrations, e.g. $c_i^{\text{ext}}$ and $c_j^{\text{ext}}$, there is at most one chain of processes whose undirected path links $c_i^{\text{ext}}$ and $c_j^{\text{ext}}$. For simplicity, consider a strategy $\sigma$ for which there exists such a path of length $3$, e.g. corresponding to:

$$\sigma : c_i^{\text{ext}} \xrightarrow{\alpha_i} c_i \ , \quad c_i \xrightarrow{\kappa_{ji}} c_j \ , \quad c_j^{\text{ext}} \xrightarrow{\alpha_j} c_j \ . \tag{A34}$$

Then, altering the enzymatic activities according to

$$\delta\alpha_i \, c_i^{\text{ext}} = \delta\kappa_{ji} \, c_i = -\delta\alpha_j \, c_j^{\text{ext}} \tag{A35}$$

leaves the net internal fluxes unchanged and the altered strategy $\sigma'$, uses a budget $E' = E + \delta E$ with

$$\delta E = \delta\kappa_{ji} \left( 1 + \frac{c_i}{c_i^{\text{ext}}} - \frac{c_i}{c_j^{\text{ext}}} \right) . \tag{A36}$$

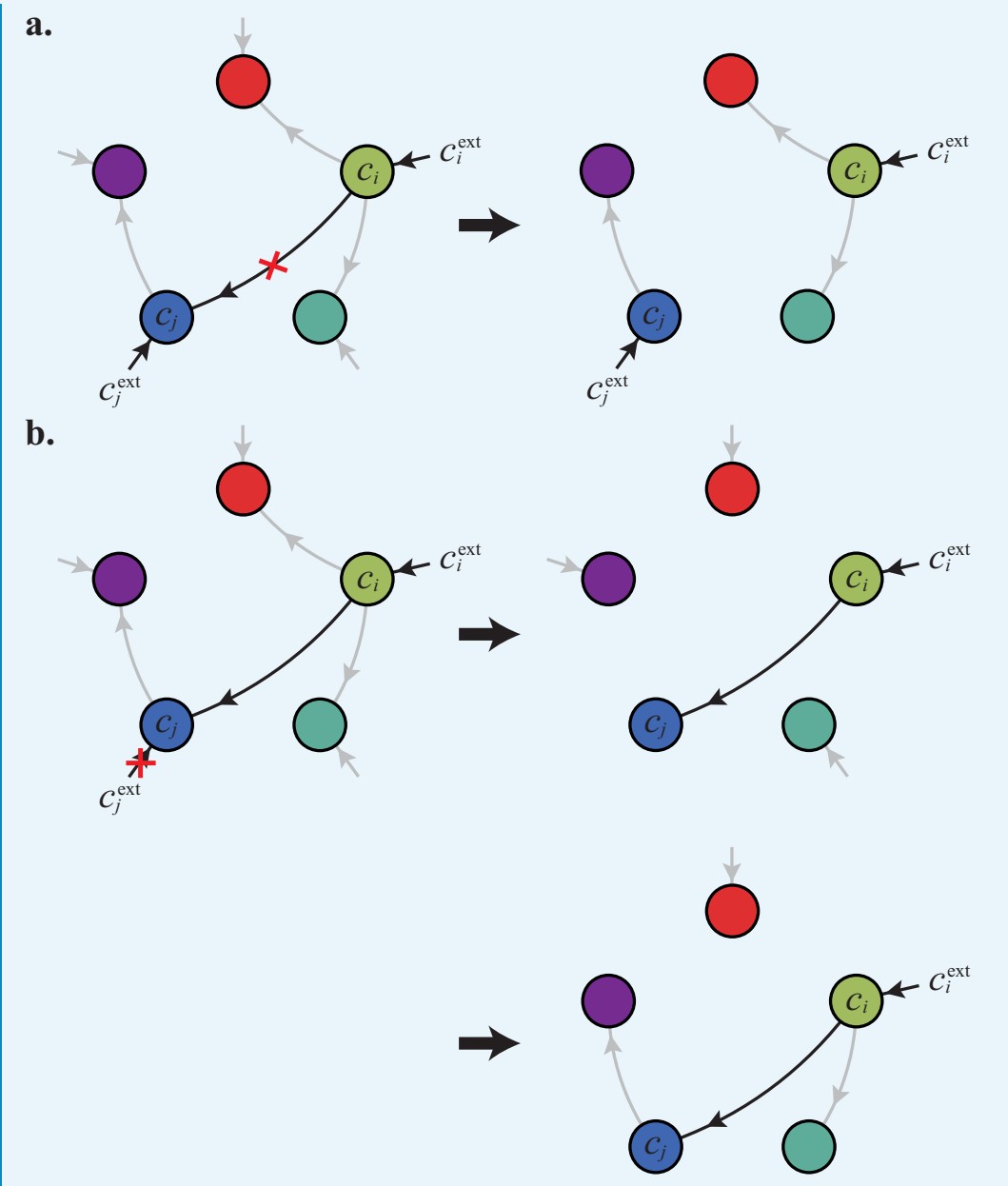

**Appendix 3—figure 2.** For fixed external concentrations, optimal metabolic classes use a single precursor for each converted building block. If cell type $\sigma$ (left) accumulates block $j$ by import and by conversion from $i$, there is always a more economical strategy $\sigma'$ (right) for which either $\kappa_{ji} = 0$ (**a**) or $\alpha_j = 0$ (**b**).

Thus, for generic conditions, one can always form a more economical strategy $\sigma'$ by either reducing or increasing $\kappa_{ji}$ according to the sign of the term in between parenthesis in *Equation (A35)*, until one of the activities along the path becomes zero. In all rigor, this last point requires that building-block import can be set to zero, which is not realistic in the presence of passive leakage. However, including passive imports does not change our result as long as the building-block internal concentrations are larger than the building-block external concentrations, which is the biologically relevant case. In any case, if a network of import and interconversion enzymes is optimal, there is at most one path connecting a building block to an external resource. Moreover, such a path has to exist and has to flow from the external building-block source for the internal building-block concentration to be

non-zero. As a result, the network of import and interconversion enzymes forms a forest of directed trees rooted in the external building-block concentrations.

## Optimal networks convert at most one building block

Using arguments inspired by network theory, we have shown that, in optimal networks of import and conversion enzymes, each building block is made from a single external source, via a single enzymatic chain. This constraint on the topology of optimal networks drastically reduces the number of metabolic classes to inspect for optimal strategies. In the following, we will show that it is possible to further restrict the set of candidate classes. Specifically, we will show that optimal strategies have a single non-trivial tree of depth one, i.e. the building blocks that are not directly imported are all made in one step from a single imported building block. In other words, optimal strategies convert at most one building block.

To prove the above claim, we first show that, for a complete set of import and interconversion enzymes, trees originating from an imported building block are at most of depth one (see *Appendix 3—figure 3*). Consider a strategy $\sigma$ that processes a precursor building block $i$ via a tree of interconversions of depth at least $2$. Necessarily, there is a building block $k$ made after two successive interconversions. Call $j$ the intermediary building block between $i$ and $k$. We thus have

$$\sigma: \quad c_i \xrightarrow{\kappa_{ji}} c_j \quad \text{and} \quad c_j \xrightarrow{\kappa_{kj}} c_k \,,$$

where $c_i$, $c_j$, and $c_k$ denote steady-state internal concentrations. Now consider a strategy $\sigma'$ that is the same as $\sigma$ except that $k$ is directly made from $i$, i.e. $\kappa'_{kj} = 0$ and $\kappa'_{ki} > 0$, and possibly $\kappa'_{ji} \neq \kappa_{ji}$:

$$\sigma': \quad c_i \xrightarrow{\kappa'_{ji}} c_j \quad \text{and} \quad c_i \xrightarrow{\kappa'_{ki}} c_k \,.$$

Choosing $\kappa'_{ji}$ and $\kappa'_{ki}$ such that

$$\kappa_{ji}\, c_i - \kappa_{kj}\, c_j = \kappa'_{ji}\, c_i \tag{A37}$$

$$\kappa'_{ki}\, c_i = \kappa_{kj} c_j \tag{A38}$$

leaves the net fluxes into each internal building block unchanged and the altered strategy $\sigma'$, which grows as fast as $\sigma$, uses a budget

$$E' = \kappa'_{ji} + \kappa'_{ki} = \kappa_{ji} < \kappa_{ji} + \kappa_{kj} = E \,. \tag{A39}$$

Thus, one can always form a more economical strategy $\sigma'$ by replacing a two-step synthesis process by a one-step synthesis process. Recursive application of the above argument shows that, if not imported, building blocks should be made from their imported precursor in as few steps as possible. In particular, we deduce that optimal networks have trees of depth one, i.e. any converted building block is made in one step from its precursor.

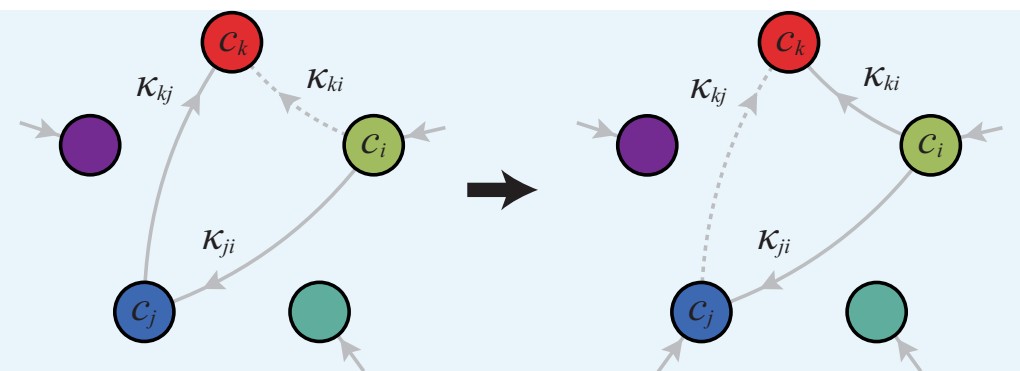

**Appendix 3—figure 3.** Optimal metabolic classes convert building blocks in the minimum number of steps. If cell type $\sigma$ (left) accumulates block $k$ via a 2-step conversion from block $i$, there is always a more economical strategy $\sigma'$ (right) that converts block $i$ directly into block $k$.

To complete the proof of our claim, we now show that an optimal strategy uses at most a single imported building block as a precursor for the synthesis of non-imported blocks (see **Appendix 3—figure 4**). Consider a strategy $\sigma$ that uses two imported building blocks: building block $i$ as a precursor for $q_i$ blocks, and building block $j$ as a precursor for $q_j$ blocks. By symmetry, in optimal strategies, the $q_i$ blocks made from $i$ are processed identically at concentration $c_i$, as well as the $q_j$ blocks made from $j$ at concentration $c_j$. Thus, the internal flux-balance equations for the precursors read

$$(\beta + \alpha_i)c_i^{\text{ext}} - \left(\beta + \sum_k \kappa_{ki}\right)c_i - b_i g = 0, \tag{A40}$$

$$(\beta + \alpha_j)c_j^{\text{ext}} - \left(\beta + \sum_l \kappa_{lj}\right)c_j - b_j g = 0, \tag{A41}$$

while the internal flux-balance equations for the end products read

$$\beta c_k^{\text{ext}} + \kappa_{ki}c_i - \beta c_k - b_k g = 0, \tag{A42}$$
$$\beta c_l^{\text{ext}} + \kappa_{lj}c_j - \beta c_l - b_l g = 0, \tag{A43}$$

where $\kappa_{ki}$ and $\kappa_{lj}$ are the enzymatic activities associated with the conversion of $i$ and $j$ into an end product, respectively. Algebraic manipulations of the flux-balance equations allow one to express the enzymatic activities $\alpha_i$, $\alpha_j$, $\kappa_{ki}$, and $\kappa_{lj}$ in terms of the internal and external building-block concentrations and of the rate of biomass production $g$. For instance, we have:

$$\alpha_i = \left(b_i g + \beta(c_i - c_i^{\text{ext}}) + \sum_k \left(b_k g + \beta(c_k - c_k^{\text{ext}})\right)\right)/c_i^{\text{ext}} \tag{A44}$$

$$\kappa_{ki} = \left(b_k g + \beta(c_k - c_k^{\text{ext}})\right)/c_i \tag{A45}$$

From there, one can show that the budget $E_i = \alpha_i + \sum_k \kappa_{ki}$ allocated to the $i$ pathways is

$$E_i = \sum_k \left(b_k g + \beta(c_k - c_k^{\text{ext}})\right)\left(\frac{1}{c_i} + \frac{1}{c_i^{\text{ext}}}\right) + \frac{b_i g + \beta c_i}{c_i^{\text{ext}}} - \beta, \tag{A46}$$

while the budget $E_j = \alpha_j + \sum_l \kappa_{lj}$ allocated to the $j$ pathways is

$$E_j = \sum_l \left(b_l g + \beta(c_l - c_l^{\text{ext}})\right)\left(\frac{1}{c_j} + \frac{1}{c_j^{\text{ext}}}\right) + \frac{b_j g + \beta c_j}{c_j^{\text{ext}}} - \beta.$$ (A47)

Now consider a strategy $\sigma'$ for which the internal concentrations are the same as $\sigma$, except that it only uses $i$ as a precursor. Accordingly, we have the new internal flux-balance equations for imported blocks $i$ and $j$

$$(\beta + \alpha_i')c_i^{\text{ext}} - \left(\beta + \sum \kappa_{ki}' + \sum \kappa_{li}'\right)c_i - b_i r = 0,$$ (A48)

$$(\beta + \alpha_j')c_j^{\text{ext}} - \beta - b_j r = 0,$$ (A49)

an the new internal flux-balance equations for the end products

$$\beta c_k^{\text{ext}} + \kappa_{ki}'c_i - \beta c_k - b_k r = 0,$$ (A50)

$$\beta c_l^{\text{ext}} + \kappa_{li}'c_i - \beta c_l - b_l r = 0,$$ (A51)

where $\alpha_i'$, $\alpha_j'$, $\kappa_{ki}'$ and $\kappa_{li}'$ denote the enzymatic activity of strategy $\sigma'$. The budget $E_i' = \alpha_i' + \sum_k \kappa_{ki}' + \sum_l \kappa_{li}'$ allocated to the $i$ pathways is

$$E_i' = \frac{b_i g + \beta c_i}{c_i^{\text{ext}}} - \beta + \left(\sum_k \left(b_k g + \beta(c_k - c_k^{\text{ext}})\right) + \sum_l \left(b_l g + \beta(c_l - c_l^{\text{ext}})\right)\right)\left(\frac{1}{c_i} + \frac{1}{c_i^{\text{ext}}}\right),$$ (A52)

while the budget $E_j' = \alpha_j$ allocated to the $j$ pathways is

$$E_j' = \frac{b_j g + \beta c_j}{c_j^{\text{ext}}} - \beta.$$ (A53)

The overall difference in budget $\delta E = E - E' = E_i + E_j - E_i' - E_j'$ reads

$$E - E' = \sum_l \left(b_l g + \beta(c_l - c_l^{\text{ext}})\right)\left[\left(\frac{1}{c_j} + \frac{1}{c_j^{\text{ext}}}\right) - \left(\frac{1}{c_i} + \frac{1}{c_i^{\text{ext}}}\right)\right],$$ (A54)

Therefore, as $b_l g + \beta(c_l - c_l^{\text{ext}}) \geq 0$ by **Equations (A43) and (A51)**, one can always form a more economical strategy $\sigma'$ by either setting $p_i$ or $p_j$ to zero according to the sign of the expression between brackets in **Equation (A54)**. Thus, when considering two sources of building blocks, it is always more economical to use a single external building block for conversion. This argument shows that strategies with optimal growth convert at most one building block.

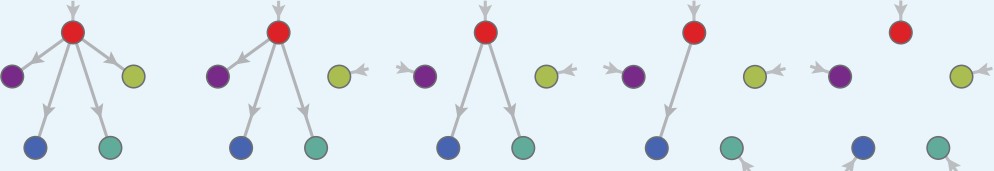

**Appendix 3—figure 4.** Optimal metabolic classes can only have a single tree of direct conversion(s). For $p$ building block, there are $p$ such network topologies. Taking into account building-blocks permutations leads to $1 + p(2^{p-1} - 1)$ metabolic types.

# Appendix 4: The analytically solvable minimum model

Analytically identifying the strategies with optimal growth rate at fixed external concentrations proves to be an arduous task for a general growth model, or even for the harmonic-mean model *Equation (A1)*. However, for the 'minimum' model it is possible to fully characterize optimal strategies. Namely, it is possible to specify their exact allocation of enzymes. Clearly, optimal strategies utilize their entire enzyme budget so that the budget constraint is actually an equality. In the following, considering strategies for which

$$\sum_i \alpha_i + \sum_{i,j} \kappa_{ij} = E, \tag{A55}$$

we establish closed-form expressions for the maximum-growth-rate function at fixed external concentrations for the minimum model. i.e. for

$$g(c_1, \ldots, c_p) = \gamma \min\left(\frac{c_1}{b_1}, \ldots, \frac{c_p}{b_p}\right), \tag{A56}$$

where $\gamma$ is a kinetic rate constant. In turn, we use these closed-form expressions to visualize graphically the structure of microbial coexistence in the space of external building-block concentrations for $p \leq 3$.

## Pure-importer strategy

Consider a strategy $\sigma$ belonging to the 'pure-importer' metabolic type that actively imports every building block, i.e. does not convert building blocks. Define the auxiliary variable $m = \min\left(c_1/b_1, \ldots, c_p/b_p\right)$. If strategy $\sigma$ is optimal, its growth rate satisfies $g = \gamma m = \gamma c_i/b_i$ for all $i$. Indeed, suppose there exists an $i$ for which $c_i/b_i > m$. Consider the set of indices $J$ for which $j$ is such that $c_j/b_j = m$. For $j \in J$, the internal flux-balance equations are

$$(\beta + \alpha_j)c_j^{\text{ext}} - (\beta + \gamma)b_j m = 0, \tag{A57}$$

while for indices $i \notin J$, the internal flux-balance equations are

$$(\beta + \alpha_i)c_i^{\text{ext}} - \beta c_i + \gamma b_i m = 0, \tag{A58}$$

Thus, for all $j \in J$ and $i \notin J$, we have

$$\frac{c_i}{b_i} = \frac{(\beta + \alpha_i)c_i^{\text{ext}} - b_i \gamma m}{\beta b_i} > \frac{(\beta + \alpha_j)c_j^{\text{ext}}}{b_j(\beta + \gamma)} = \frac{c_j}{b_j} = m. \tag{A59}$$

If there are $q$ indices in $J$, one can form the strategy $\sigma'$ from $\sigma$ by taking an amount $\delta E/(p - q)$ from every $\alpha_i$, $i \notin J$, and by distributing an amount $\delta E/q$ to every $\alpha_j$, $j \in J$. The new strategy $\sigma'$ uses the same budget as strategy $\sigma$ and, for small enough $\delta E > 0$, every new import activity is positive. Moreover, every $c_i'/b_i$, $i \notin J$, is a decreasing function of $\delta E$ and every $c_j'/b_j$, $j \in J$ is an increasing function of $\delta E$. Thus, for small enough $\delta E > 0$, we have $m' = \min\left(c_1'/b_1, \ldots, c_p'/b_p\right) > m$, which contradicts the optimality of $\sigma$ and shows that $m = c_i/b_i$ for all $p$ internal building-block concentrations.

Since $m = c_i/b_i$ for all $i$, the internal flux-balance equations of an optimal strategy actively importing every building block are

$$(\beta + \alpha_i)c_i^{\text{ext}} - (\beta + \gamma)b_i m = 0.$$ (A60)

Considering *Equation (A60)* for two indices $i$ and $j$ and equating $m$ yields

$$\left(\frac{\beta + \alpha_i}{\beta + \gamma}\right)\left(\frac{c_i^{\text{ext}}}{b_i}\right) = \left(\frac{\beta + \alpha_j}{\beta + \gamma}\right)\left(\frac{c_j^{\text{ext}}}{b_j}\right).$$ (A61)

Thus, the import activities $\alpha_j$ can all be expressed in term of the single import activity $\alpha_i$ via

$$\alpha_j = (\beta + \alpha_i)\frac{b_j c_i^{ext}}{b_j c_j^{ext}} - \beta.$$ (A62)

In turn, we obtain a simple expression for $\alpha_i$ from the budget constraint

$$\sum_j \alpha_j = (\beta + \alpha_i)\left(1 + \sum_{j\neq i}\frac{b_j c_i^{\text{ext}}}{b_i c_j^{\text{ext}}}\right) - \beta p = E,$$ (A63)

which leads to a closed form expression for the optimal growth rate

$$g_{\sigma^\star} = \gamma m = \frac{\gamma}{\beta + \gamma}\left(\frac{(\beta + \alpha_i)c_i^{\text{ext}}}{b_i}\right) = \frac{\gamma}{\beta + \gamma}\left(\frac{E + \beta p}{\sum_i b_i/c_i^{\text{ext}}}\right).$$ (A64)

For the above expression to be valid, one also needs to verify the positivity of the enzymatic activities

$$\alpha_i = \left(\frac{E + \beta p}{1 + \sum_{j\neq i}\frac{b_j c_i^{\text{ext}}}{b_i c_j^{\text{ext}}}}\right) - \beta > 0,$$ (A65)

which is always true at steady state if the budget $E$ is large enough compared with the passive leakage rate $\beta$.

## Interconversion strategies

Consider a strategy $\sigma$ that actively imports $q < p$ building blocks. Denote by $S$ the set of indices of actively imported blocks. There is a unique building block $i \in S$ that is used to produce the $p - q$ building blocks that are not imported. Denote by $C$ the set of indices of converted blocks. Reasoning by contradiction as in Section "Pure-importer strategy", one can show that optimal strategies are such that $g = \gamma m = \gamma c_j/b_j$ for all $j \neq i$, where $m = \min\left(c_1/b_1, \ldots, c_p/b_p\right)$. The key distinction from the case of the pure importer is that we possibly have $c_i/b_i > m$. Such a case emerges for moderate enzyme budget satisfying $E \leq p(\beta + \gamma)$.

**Case** $c_i/b_i > m$:

In this case, the internal flux-balance equations of an optimal strategy actively importing building blocks $j \in S$, while using $i$ as sole precursor to building blocks $j \in C$, read

$$(\beta + \alpha_i)c_i^{\text{ext}} - \left(\beta + \sum_{j \in C} \kappa_{ji}\right)c_i - \gamma b_i m = 0, \tag{A66}$$

$$\beta c_j^{\text{ext}} + \kappa_{ji}c_i - (\beta + \gamma)b_j m = 0, \quad j \in C, \tag{A67}$$

$$(\beta + \alpha_j)c_j^{\text{ext}} - (\beta + \gamma)b_j m = 0, \quad j \notin C, \quad j \neq i. \tag{A68}$$

Notice that relation **Equation (A67)** allows one to express $m$ as a function of $c_i$. Substituting the corresponding expression for $m$ in the condition $c_i/b_i > m$ yields

$$\kappa_{ji} < (\beta + \gamma)\left(1 - \frac{\beta}{\beta + \gamma}\frac{c_j^{\text{ext}}}{c_i}\right), \tag{A69}$$

which can be satisfied for positive $\kappa_{ji}$ when the internal building-block concentration $c_i$ exceeds the external building-block concentration $c_j^{\text{ext}}$. Then, using **Equation (A66)**, one can write **Equation (A67)** as

$$\beta c_j^{\text{ext}} + \frac{\kappa_{ji}}{\beta + \sum_{j \in C}\kappa_{ji}}\left[(\beta + \alpha_i)c_i^{\text{ext}} - \gamma b_i m\right] - (\beta + \gamma)b_j m = 0. \tag{A70}$$

Solving for $\tilde{\kappa}_{ji} = \kappa_{ji}/(\beta + \sum_{j \in C}\kappa_{ji})$ in the above equation yields:

$$\tilde{\kappa}_{ji} = \frac{(\beta + \gamma)b_j m - \beta c_j^{\text{ext}}}{(\beta + \alpha_i)c_i^{\text{ext}} - \gamma b_i m}. \tag{A71}$$

Then, using **Equation (A71)** and the relation

$$\sum_{j \in C}\kappa_{ji} = \beta\frac{\sum_{j \in C}\tilde{\kappa}_{ji}}{1 - \sum_{j \in C}\tilde{\kappa}_{ji}}, \tag{A72}$$

one obtains an expression for the enzymatic activity $\kappa_{ji} = (\beta + \sum_{j \in C}\kappa_{ji})\tilde{\kappa}_{ji}$ as a function of $m$:

$$\kappa_{ji}(m) = \frac{\beta(\beta + \gamma)b_j m - \beta^2 c_j^{\text{ext}}}{(\beta + \alpha_i)c_i^{\text{ext}} + \beta\sum_{j \in C}c_j^{\text{ext}} - \left(\gamma b_i + (\beta + \gamma)\sum_{j \in C}b_j\right)m}. \tag{A73}$$

If $q > 1$, adopting the reasoning from Section "Pure-importer strategy" to optimize over the $\alpha_j, j \in S, j \neq i$, at fixed $\alpha_i$ and $\kappa_{ij}$, we have:

$$m = \frac{1}{\beta + \gamma}\left(\frac{(\beta + \alpha_j)c_j^{\text{ext}}}{b_j}\right) \tag{A74}$$

$$= \frac{1}{\beta + \gamma}\left(\frac{E + \beta(q - 1) - \alpha_i - \sum_{j \in C}\kappa_{ji}(m)}{\sum_{j \in S \setminus \{i\}}b_j/c_j^{\text{ext}}}\right). \tag{A75}$$

Solving this quadratic equation for $m$, it can be seen that the larger root gives a negative value for $\kappa_{ji}$. Hence it is the smaller root that defines the function $m(\alpha_i)$, which, in turn, can be optimized over the import enzymatic activity $\alpha_i$ to obtain the optimal growth rate for a metabolic class with interconversions:

$$g_{\sigma^\star} = \gamma \frac{u}{v}\left(1 - \sqrt{1 - \frac{w}{u^2}}\right), \tag{A76}$$

where the reduced variables $u$, $v$, and $w$ are

$$u = (E+\beta q)c_i^{\text{ext}} + \beta \sum_{j\in C} c_j^{\text{ext}} + \frac{2\beta \sum_{j\in C} b_j}{\sum_{j\in S\setminus\{i\}} b_j/c_j^{\text{ext}} + \left(\sum_{j\in C} b_j + \frac{\gamma}{\beta+\gamma}b_i\right)/c_i^{\text{ext}}}, \tag{A77}$$

$$v = (\beta+\gamma)\left(\sum_{j\in S\setminus\{i\}} b_j c_i^{\text{ext}}/c_j^{\text{ext}} + \sum_{j\in C} b_j + \frac{\gamma}{\beta+\gamma}b_i\right), \tag{A78}$$

$$w = \left[(E+\beta q)c_i^{\text{ext}} + \beta \sum_{j\in C} c_j^{\text{ext}}\right]^2 + 4\beta^2 c_i^{\text{ext}} \sum_{j\in C} c_j^{\text{ext}}. \tag{A79}$$

Similarly, for $q = 1$, writing the enzyme budget constraint $E = \sum_{j\neq i} \kappa_{ji}(m) + \alpha_i$ leads to an expression for $m$ in terms of $\alpha_i$. In turn, the optimization of $m(\alpha_i)$ over the import enzymatic activity $\alpha_i$ yields the same expression as in **Equation (A76)**, with **Equation (A77)**, **Equation (A78)**, and **Equation (A79)**, where the only imported block is $i$, i.e. $S \setminus \{i\} = \emptyset$.

One can check that when considered as functions of the external concentrations, the reduced variable $u$ is homogeneous of degree one, the reduced variable $v$ is homogeneous of degree zero and $w$ is homogeneous of degree two. Therefore, the optimal growth rate is homogeneous of degree one, as expected from the homogeneity of the $\min$ function. (In the harmonic-mean model, the rate function $g$ is also homogeneous of degree one).

**Case $c_i/b_i = m$:**

In this case, the internal flux-balance equations of an optimal strategy importing building blocks $j \in S$, while using $i$ as sole precursor to building blocks $j \in C$, reads

$$(\beta+\alpha_i)c_i^{\text{ext}} - \left(\beta + \sum_{j\in C} \kappa_{ji} + \gamma\right)b_i m = 0, \tag{A80}$$

$$\beta c_j^{\text{ext}} + \left(\kappa_{ji}b_i - (\beta+\gamma)b_j\right)m = 0, \quad j \in C, \tag{A81}$$

$$(\beta+\alpha_j)c_j^{\text{ext}} - (\beta+\gamma)b_j m = 0, \quad j \notin C, \quad j \neq i. \tag{A82}$$

From **Equation (A81)**, we deduce that the conversion activities satisfy

$$\kappa_{ji} = \frac{1}{b_i}\left((\beta+\gamma)b_j - \frac{\beta}{m}c_j^{\text{ext}}\right), \tag{A83}$$

and, using the above result in **Equation (A80)**, we find

$$\alpha_i = (\beta+\gamma)\left(b_i + \sum_{j\in C} b_j\right)\frac{m}{c_i^{\text{ext}}} - \beta\left(1 + \sum_{j\in C} \frac{c_j^{\text{ext}}}{c_i^{\text{ext}}}\right). \tag{A84}$$

If $q>1$, adopting the reasoning from Section "Pure-importer strategy to optimize over the $\alpha_j$, $j \in S, j \neq i$, at fixed $\alpha_i$ and $\kappa_{ij}$, we still have:

$$m = \frac{1}{\beta + \gamma} \left( \frac{E + \beta(q-1) - \alpha_i - \sum_{j \in C} \kappa_{ji}}{\sum_{j \in S \setminus \{i\}} b_j / c_j^{\text{ext}}} \right). \tag{A85}$$

Finally, substituting for $\kappa_{ji}$ and $\alpha_i$ in the above equation, $m$ is determined as the unique positive solution of a quadratic equation, which yields:

$$g_{\sigma^\star} = \gamma \frac{-v + \sqrt{v^2 - 4uw}}{2u} \tag{A86}$$

where the reduced variables $u$, $v$, and $w$ are

$$u = -(\beta + g) \left( \sum_{j \in S \setminus \{i\}} \frac{b_j}{c_j^{\text{ext}}} + \sum_{j \in C} \frac{b_j}{c_i^{\text{ext}}} + \frac{b_i}{c_i^{\text{ext}}} \right) \tag{A87}$$

$$v = E + \beta \left( q + \sum_{j \in C} \frac{c_j^{\text{ext}}}{c_i^{\text{ext}}} \right) - (\beta + \gamma) \sum_{j \in C} \frac{b_j}{b_i}, \tag{A88}$$

$$w = \frac{\beta}{b_i} \sum_{j \in C} c_j^{\text{ext}}. \tag{A89}$$

Similarly, for $q = 1$, writing the enzyme budget constraint $E = \sum_{j \neq i} \kappa_{ji}(m) + \alpha_i$ leads to an expression of $m$ in terms of $\alpha_i$. In turn, the optimization of $m(\alpha_i)$ over the import enzymatic activity $\alpha_i$ yields the same expression as in *Equation (A86)*, with *Equation (A87)*, *Equation (A88)*, and *Equation (A89)*, where $S \setminus \{i\} = \emptyset$.

## Coexistence at steady state for $p \leq 3$ building blocks

As the solution to $G(c_1^{\text{ext}}, \ldots, c_p^{\text{ext}}) = \max_j g_{\sigma_j^\star}(c_1^{\text{ext}}, \ldots, c_p^{\text{ext}}) = \delta$, the set of steady-state external concentrations $c_1^\star, \ldots, c_p^\star$ defines a multi-patched hypersurface. Each hypersurface patch corresponds to the set of external building-block concentrations for which a given metabolic type achieves the optimal rate of biomass production. Coexistence of multiple strategies, belonging to different metabolic types, occurs for external concentrations at the intersection of patches. As we consider $p$ building blocks, we expect consortia to be generically made of $p$ strategies, each belonging to a different metabolic type. Moreover, consortia with $q$ coexisting strategies occur locally at the intersection of $q$ hypersurfaces in a $p$-dimensional space of external concentrations, determining a $p - q$ dimensional set. In particular, consortia with $p$ coexisting strategies occur for a set of isolated external building-block concentrations. We refer to such consortia as microbial cartels. In the following, we show the generic occurrence of cartels and characterize their structure by specifying their composition and their associated external building-block concentrations.

Exploiting analytical expressions for the optimal growth of various metabolic classes, one can visually inspect the hypersurface defined by $c_1^\star, \ldots, c_p^\star$ for $p \leq 3$. Although our optimization is valid for any stoichiometric coefficients $b_i$, for simplicity, we consider the symmetric case where building blocks have the same biomass stoichiometry, i.e. for which $b_i = 1$ for all $i$.

For $p = 2$, there are two particular sets of building-block external concentrations for which microbial cartels are optimal. By symmetry with respect to building block permutations, it is enough to characterize one cartel, e.g., the one occurring for $c_1^\star > c_2^\star$. In this case, the microbial cartel is composed of a converting strategy, one that imports block 1 to synthesize block 2, and a pure-importer strategy (see *Appendix 4—figure 1*).

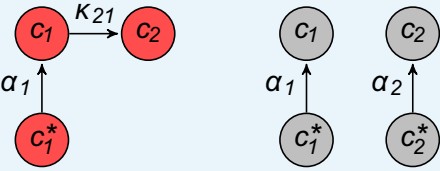

**Appendix 4—figure 1.** Microbial cartels for 2 building blocks.

Considering the case $p = 3$ reveals a more complex picture as shown in *Appendix 4—figure 5a* and *Appendix 4—figure 5b*. For moderate enzyme budgets $E \lesssim p(\beta + \gamma)$, there are 9 cartels of 3 coexisting strategies (see *Appendix 4—figure 5a*), whereas, for large enzyme budgets $E \gg p(\beta + \gamma)$, 3 pairs of consortia merge, yielding 3 cartels of 3 coexisting strategies and 3 cartels of 4 coexisting strategies (see *Appendix 4—figure 5b*). In any case, microbial cartels can be categorized based on symmetry considerations. For instance, for moderate enzyme budget, 6 cartels are associated to strictly ordered external concentrations, e.g. $c_1^\star > c_2^\star > c_3^\star$, whereas the other 3 cartels are associated with degenerate order of the type $c_1^\star = c_2^\star > c_3^\star$, for which there are two most abundant building blocks. The strictly ordered cartel corresponding to $c_1^\star > c_2^\star > c_3^\star$ is made of the 3 metabolic types shown in *Appendix 4—figure 2*,

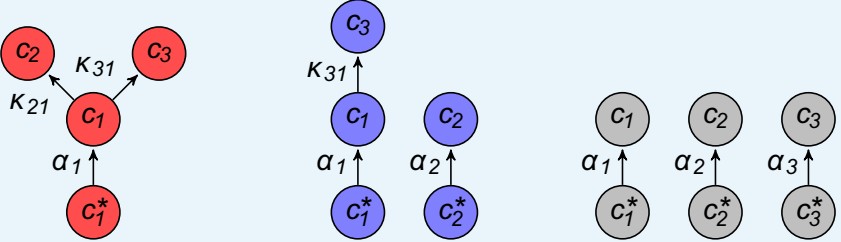

**Appendix 4—figure 2.** Microbial cartel for 3 building blocks with well-ordered concentrations.

whereas the marginal cartel corresponding to $c_1^\star = c_2^\star > c_3^\star$ comprises the metabolic types shown in *Appendix 4—figure 3*.

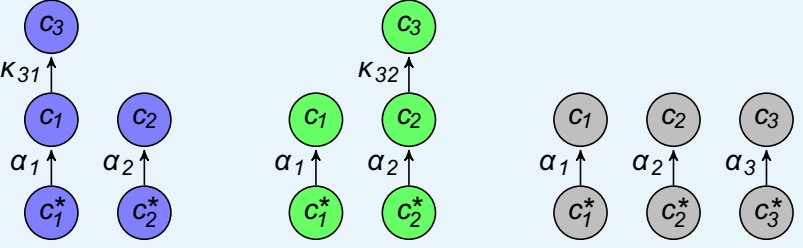

**Appendix 4—figure 3.** Microbial cartel for 3 building block with degenerate order among building-block concentrations.

Observe that the set of 6 strictly ordered cartels is symmetric with respect to building block permutations, as is the set of 3 marginal cartels. For large enzyme budget, the merging of

pairs of strictly ordered cartels leads to 3 cartels associated to orders of the type $c_1^\star > c_2^\star = c_3^\star$, for which the comprised metabolic types are shown in *Appendix 4—figure 4*.

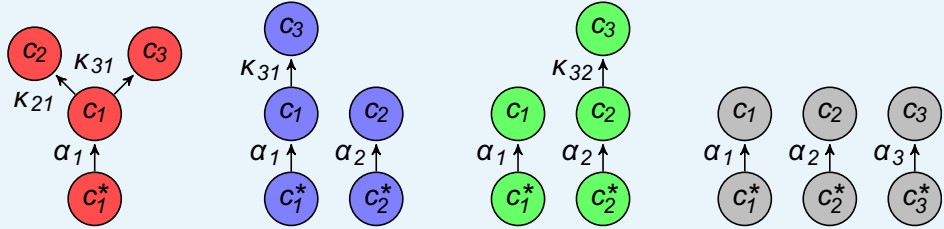

**Appendix 4—figure 4.** Merged microbial cartel for 3 building blocks.

As shown later, such cartels with are not generic as they only arise for the minimum model whose growth-rate function $g(c_1, \ldots, c_p) = \gamma \min(c_1, \ldots, c_p)$ is not differentiable when $c_i = c_j$ for $i \neq j$.

The low-dimensional examples $p = 2$ or $3$ suggest a general structure for the symmetric $p$-dimensional case, i.e. $b_i = 1$ for $1 \leq i \leq p$. If the external concentrations satisfy the order relation $c_1^{\text{ext}} > c_2^{\text{ext}} > c_3^{\text{ext}} > \ldots > c_p^{\text{ext}}$, there exists a microbial cartel with $p$ metabolic strategies. Specifically, these strategies belong to the metabolic class that converts building block 1 into the $p - 1$ other building blocks, the metabolic class that converts building block 1 into the $p - 2$ least abundant building blocks and import building block 2, the metabolic class that converts building block 1 into the $p - 3$ least abundant building blocks and import building blocks 2 and 3, ..., and the pure-importer metabolic class. Moreover, we conjecture that cartels also emerge for degenerate order relations of the type with $c_1^{\text{ext}} = \ldots = c_q^{\text{ext}} > c_{q+1}^{\text{ext}} > \ldots > c_p^{\text{ext}}$.

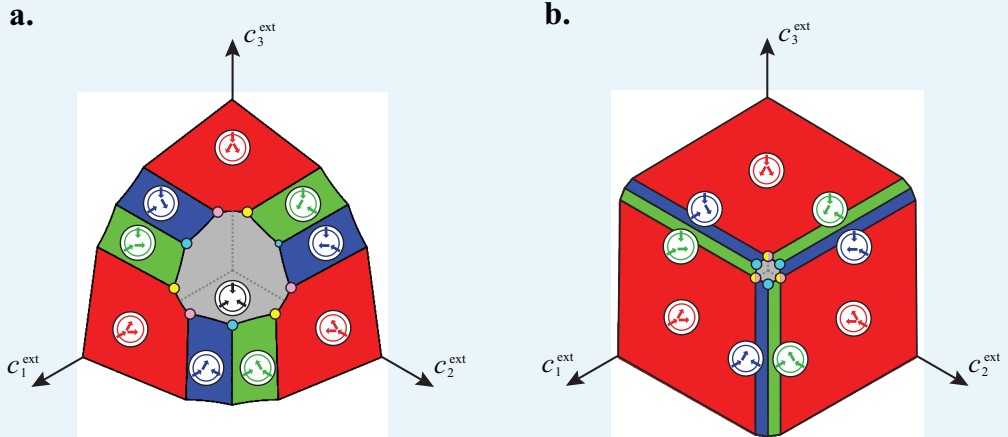

**Appendix 4—figure 5.** For large enough supply rates, population dynamics drive the external building-block concentrations towards steady-state values $c_1^\star, \ldots, c_p^\star$ that satisfy growth rate equals death rate, $G(c_1^\star, \ldots, c_p^\star) = \delta$. Consortia emerge at concentrations for which distinct metabolic classes are jointly optimal. For $p = 3$, a pure-converter strategy is optimal on the red patches, mixed strategies are optimal on the blue and green patches, while a pure-importer strategy is optimal on the grey patch. There are two types of cartels at the intersection of 3 patches: 6 distinct cartels with well-ordered external concentrations (yellow and pink), e.g. $c_1^{\text{ext}} > c_2^{\text{ext}} > c_3^{\text{ext}}$, and 3 distinct cartels with degenerate external concentration ordering (cyan), e.g. $c_1^{\text{ext}} = c_2^{\text{ext}} > c_3^{\text{ext}}$. For both panels, we take $\beta = 1$, $\gamma = 2$, $\delta = 0.2$, $b_1 = b_2 = b_3 = 1$. a. For a moderate enzyme budget $E \leq p(\beta + \gamma)$, there are 9 cartels of 3

coexisting strategies. b. For a large enzyme budget $E = 100$, 3 pairs of consortia merge, yielding 3 cartels of 3 coexisting strategies and 3 cartels of 4 coexisting strategies.

# Appendix 5: Optimal strategies within metabolic classes

Unfortunately, finding the optimal metabolic strategies in closed form proves intractable for the harmonic-mean model. However, we expect the structure of microbial cartels suggested by the analysis of the minimum model to hold for more generic growth-rate function. To justify this point, we need to reformulate the optimization of the growth rate at fixed external building-block concentrations into a geometric problem in the space of internal building-block concentrations. From there, we show that: (i) Optimal strategies utilize the most available building block for conversion and that converted blocks are the least available. (ii) The order of internal abundances of building blocks is the same as the order of external abundances, except that converted building blocks are equally abundant internally. The first property directly implies that only certain metabolic classes can be jointly optimal for the same external concentrations, limiting the number of possible cartels. The second property allows one to discard the occurrence of degenerate cartels for which internal concentrations are equal in the harmonic-mean model.

## Geometric formulation of metabolic optimization

In this section, we formulate the optimization of the cellular growth rate at fixed external building-block concentrations as a geometric problem in the space of internal building-block concentrations.

On one hand, define $\mathcal{C}_g$, the set of internal building-block concentrations for which the rate of growth exceeds a given rate $g$, i.e.

$$\mathcal{C}_g = \left\{ c_1, \ldots, c_p \,|\, g(c_1, \ldots, c_p) \geq r \right\}. \tag{A90}$$

Then, by the monotonicity of the rate function $g$, we have that

$$\mathcal{C}_{g'} \subset \mathcal{C}_g \quad \text{with} \quad g \leq g', \quad \lim_{g \to \infty} \mathcal{C}_g = \varnothing \quad \text{and} \quad \lim_{g \to 0} \mathcal{C}_g = \mathbb{R}^p_+. \tag{A91}$$

Observe that considering the case where building blocks are symmetric with respect to permutations implies that the growth-rate function is symmetric in its arguments. On the other hand, observe that, if a strategy $\sigma$ has internal concentrations $c_1, \ldots, c_p$ and grows at rate $g$, such a strategy necessarily has an enzyme budget

$$E_{g,\mathcal{M}}(c_1, \ldots, c_p) = \sum_{j \in C} \left( g + \beta(c_j - c_j^{\text{ext}}) \right) \left( \frac{1}{c_i} + \frac{1}{c_i^{\text{ext}}} \right) + \sum_{j \notin C} \left( \frac{g + \beta c_j}{c_j^{\text{ext}}} - \beta \right) \tag{A92}$$

where $C$ denotes the set of building blocks converted from building block $i$. Notice that the above expression for the enzyme budget only depends on the topology of the metabolic networks, and thus is characteristic of the metabolic class $\mathcal{M}$. Then, define $\mathcal{E}_{g,\mathcal{M}}$, the set of internal building-block concentrations for which the required budget $E_{g,\mathcal{M}}(c_1, \ldots, c_p)$ satisfies the budget constraint, i.e

$$\mathcal{E}_{g,\mathcal{M}} = \left\{ c_1, \ldots, c_p \,|\, E_{g,\mathcal{M}}(c_1, \ldots, c_p) \leq E \right\}. \tag{A93}$$

As the required budget is linearly increasing in $g$ at fixed external concentrations, we also have that

$$\mathcal{E}_{g,\mathcal{M}} \subset \mathcal{E}_{g',\mathcal{M}} \quad \text{with} \quad g \leq g'. \tag{A94}$$

Moreover, (i) for large enough rate $g$, there are no positive internal concentrations for which the budget constraint is satisfied and (ii) at zero growth rate, there always are positive internal building-block concentrations for which the budget constraint is satisfied, i.e.

$$\lim_{g \to \infty} \mathcal{E}_{g,\mathcal{M}} \cap \mathbb{R}^p_+ = \varnothing \quad \text{and} \quad \mathcal{E}_{0,\mathcal{M}} \cap \mathbb{R}^p_+ \neq \varnothing. \tag{A95}$$

The properties **Equation (A91)**, **Equation (A94)**, and **Equation (A95)** show that the optimal growth rate achievable by a metabolic class $\mathcal{M}$ can be defined as the maximal value $g>0$ for which $\mathcal{C}_g$ and $\mathcal{E}_g$ have a contact point

$$\sup_{\sigma \in \mathcal{M}} g_\sigma(c_1^{\text{ext}}, \dots, c_p^{\text{ext}}) = \sup\left\{ g \, | \, \mathcal{C}_g \cap \mathcal{E}_{g,\mathcal{M}} \neq \varnothing \right\}. \tag{A96}$$

Such a contact point defines the internal concentrations for an optimal strategy in metabolic class $\mathcal{M}$ at fixed external building-block concentrations. In particular, the uniqueness of the contact point implies that there is a unique optimal strategy within a given metabolic class. By this argument, one can see that there is unique optimal pure-importer strategy. Indeed, the quasi-concave property of the growth-rate function $g$ means that the sets $\mathcal{C}_g$ are convex, while for a pure-importer strategy, the enzyme budget $E_{g,\mathcal{M}}$ is a linear function of the internal concentrations and sets $\mathcal{E}_{g,\mathcal{M}}$ are (convex) hyperplanes. Then, the uniqueness of the optimal pure-importer strategy follows from the uniqueness of the contact point between two convex sets.

In the following, we will use a similar geometrical approach to specify the optimal pure-converter strategy which proves crucial to discard the occurence of degenerate optimal strategies for which all internal building-block concentrations are equal. Establishing this result requires first to relate the topology of optimal metabolic strategies to the relative external and internal abundances of building blocks. The next section is devoted to establish that relation.

## Building-block concentrations for optimal strategies

In this section, we exploit our geometric formulation of metabolic optimization to infer properties about the optimal strategies of a metabolic class as defined in **Equation (A96)**.

### Conditions on external building blocks

Here, we show that if the external concentrations satisfy the order relation $c_1^{\text{ext}} > c_2^{\text{ext}} \geq c_3^{\text{ext}} \geq \dots \geq c_p^{\text{ext}}$, then at most $p$ metabolic classes can coexist: the metabolic class $\mathcal{M}_1$ that converts building block 1 into the $p-1$ other building blocks, the metabolic class $\mathcal{M}_2$ that converts building block 1 into the $p-2$ least abundant building blocks and imports building block 2, the metabolic class $\mathcal{M}_3$ that converts building block 1 into the $p-3$ least abundant building blocks and imports building blocks 2 and 3, ..., and the pure-importer metabolic class $\mathcal{M}_p$. Observe that we assume a strict ordering between the concentrations of the two most abundant building blocks.

First, we show that at fixed external concentrations, it is always more economical to use the most abundantly available building block in $i \notin C$ as a precursor for conversion. To see this, consider $\mathcal{M}$ and $\mathcal{M}'$ the metabolic classes that respectively use $i$ and $j$ as a precursor for building blocks $k \in C$. Then, defining $c'_1, \dots, c'_p$ as the transposition of $c_1, \dots, c_p$ such that $c'_i = c_j$ and $c'_j = c_i$, one can see that relation **Equation (A92)** implies that

$$E_{g,\mathcal{M}'}(c'_1,\ldots,c'_p) - E_{g,\mathcal{M}}(c_1,\ldots,c_p) = \sum_{k \in C} \left(g + \beta(c_k - c_k^{\text{ext}})\right)\left(\frac{1}{c_j^{\text{ext}}} - \frac{1}{c_i^{\text{ext}}}\right). \tag{A97}$$

Thus, if $c_i^{\text{ext}} > c_j^{\text{ext}}$, for any strategy in $\mathcal{M}'$, i.e. for any internal concentrations $c'_1,\ldots,c'_p$, we have

$$E_{g,\mathcal{M}'}(c'_1,\ldots,c'_p) \geq E_{g,\mathcal{M}}(c'_1,\ldots,c'_p), \tag{A98}$$

showing that there is a more economical strategy in $\mathcal{M}$ with internal concentration $c_1,\ldots,c_p$ such that $g(c_1,\ldots,c_p) = g(c'_1,\ldots,c'_p)$ by symmetry of the growth rate function.

Second, we show that it is always more economical to produce the least abundant building blocks by conversion. To see this, consider a metabolic class $\mathcal{M}$ that converts building block $i$ into building block $j$ and the metabolic class $\mathcal{M}'$ that only differs from $\mathcal{M}$ by the fact that it imports building block $j$. Then, for fixed internal concentrations $c_1,\ldots,c_p$, the difference of the enzyme budgets associated to each metabolic class satisfies:

$$E_{g,\mathcal{M}}(c_1,\ldots,c_p) - E_{g,\mathcal{M}'}(c_1,\ldots,c_p) = \left(g + \beta(c_j - c_j^{\text{ext}})\right)\left(\frac{1}{c_i} + \frac{1}{c_i^{\text{ext}}} - \frac{1}{c_j^{\text{ext}}}\right). \tag{A99}$$

If importing building block $j$ is optimal, we necessarily have

$$E_{g,\mathcal{M}}(c_1,\ldots,c_p) \geq E_{g,\mathcal{M}'}(c_1,\ldots,c_p) \quad \text{i.e.} \quad \frac{1}{c_i} + \frac{1}{c_i^{\text{ext}}} \geq \frac{1}{c_j^{\text{ext}}}, \tag{A100}$$

whereas, if converting $j$ is optimal, we necessarily have

$$E_{g,\mathcal{M}}(c_1,\ldots,c_p) \leq E_{g,\mathcal{M}'}(c_1,\ldots,c_p) \quad \text{i.e.} \quad \frac{1}{c_i} + \frac{1}{c_i^{\text{ext}}} \leq \frac{1}{c_j^{\text{ext}}}. \tag{A101}$$

Thus, for a metabolic class $\mathcal{M}$ converting $i$ in $j \in C$ to be optimal, we necessarily have

$$\frac{1}{c_i} + \frac{1}{c_i^{\text{ext}}} \leq \frac{1}{c_j^{\text{ext}}}, j \in C, \quad \text{and} \quad \frac{1}{c_i} + \frac{1}{c_i^{\text{ext}}} \geq \frac{1}{c_k^{\text{ext}}}, k \notin C. \tag{A102}$$

In particular, the above inequalities implies $c_k^{\text{ext}} > c_i^{\text{ext}}$. In other words, at fixed external concentrations, optimal metabolic classes are such that every imported building block has higher external concentration than the external concentration of any converted building block. Therefore, for the metabolic class that converts building block $i$ into building blocks $j \in C$ to be optimal, we need that $c_i^{\text{ext}} = \max_k c_k^{\text{ext}}$ and $\max_{j \in C} c_j^{\text{ext}} \leq \min_{k \notin C} c_k^{\text{ext}}$.

Consider now that the external concentrations satisfy the order relation $c_1^{\text{ext}} > c_2^{\text{ext}} \geq c_3^{\text{ext}} \geq \ldots \geq c_p^{\text{ext}}$. Then, only metabolic classes that use building block 1 as precursor for conversion can be optimal as $c_1^{\text{ext}} = \max_j c_j^{\text{ext}}$. Moreover, if a metabolic strategy imports a building block besides building block 1, it necessarily imports the second most abundant building block, i.e. 2, so that $\max_{j>2} c_j^{\text{ext}} \leq c_2^{\text{ext}}$. Similarly, if a metabolic strategy imports $q$ building blocks, it necessarily imports the $q$ most abundant building blocks, so that $\max_{j>q} c_j^{\text{ext}} \leq \min_{k \leq q} c_k^{\text{ext}}$. The above argument implies that degenerate cartels with coexisting optimal strategies belonging to more than $p$ metabolic classes can only happen for degenerate order relation on the external building-block concentration, i.e. when there are $i \neq j$ such that $c_i^{\text{ext}} = c_j^{\text{ext}}$.

## Conditions on internal building blocks

Here, we show that if a metabolic class $\mathcal{M}$ is optimal for external concentrations satisfying, say, $c_1^{\text{ext}} \geq c_2^{\text{ext}} \geq c_3^{\text{ext}} \geq \ldots \geq c_p^{\text{ext}}$, then the internal concentrations of the optimal strategy in $\mathcal{M}$ satisfy an order relation. Specifically, for the metabolic class $\mathcal{M}_q$ that comprises strategies importing the $q$ most abundant building blocks, the optimal strategy is such that $c_1 \geq c_2 \geq \ldots \geq c_{q+1} = \ldots = c_p$. In other words, the internal building-block abundances mirror the external abundances except for the converted building blocks, which all have the same concentration.

Proving the above point requires similar arguments as for the case of the external building-block abundances. Indeed, at fixed external concentrations, a strategy in metabolic class $\mathcal{M}_q$ growing at rate $g$ and having internal concentrations $c_1, \ldots, c_p$ requires an enzyme budget

$$E_{g,\mathcal{M}_q}(c_1,\ldots,c_p) = \sum_{j>q}\left(g + \beta(c_j - c_j^{\text{ext}})\right)\left(\frac{1}{c_1} + \frac{1}{c_1^{\text{ext}}}\right) + \sum_{j\leq q}\left(\frac{g + \beta c_j}{c_j^{\text{ext}}} - \beta\right). \tag{A103}$$

Then, defining $c_1', \ldots, c_p'$ as the transposition of $c_1, \ldots, c_p$ such that $c_1' = c_j$ and $c_j' = c_1$, with $j \leq q$, one can see that relation **Equation (A103)** implies that

$$E_{g,\mathcal{M}_q}(c_1',\ldots,c_p') - E_{g,\mathcal{M}_q}(c_1,\ldots,c_p) \tag{A104}$$

$$= \sum_{k\in C}\left(g + \beta(c_k - c_k^{\text{ext}})\right)\left(\frac{1}{c_1'} - \frac{1}{c_1}\right) + \beta\left(\frac{c_1' - c_1}{c_1^{\text{ext}}} + \frac{c_j' - c_j}{c_j^{\text{ext}}}\right), \tag{A105}$$

$$= \sum_{k\in C}\left(g + \beta(c_k - c_k^{\text{ext}})\right)\left(\frac{1}{c_j} - \frac{1}{c_1}\right) + \beta(c_j - c_1)\left(\frac{1}{c_1^{\text{ext}}} - \frac{1}{c_j^{\text{ext}}}\right), \tag{A106}$$

which has the sign of $c_1 - c_j$ as $c_1^{\text{ext}} \geq c_j^{\text{ext}}$. Thus, we necessarily have $c_1 = \max_j c_j$. Moreover, defining $c_1', \ldots, c_p'$ as the transposition of $c_1, \ldots, c_p$ such that $c_i' = c_j$ and $c_j' = c_i$, with $1 < i, j \leq q$, one can see that relation **Equation (A103)** implies that

$$E_{g,\mathcal{M}_q}(c_1',\ldots,c_p') - E_{g,\mathcal{M}_q}(c_1,\ldots,c_p) = \beta(c_j' - c_j)\left(\frac{1}{c_i} + \frac{1}{c_i^{\text{ext}}}\right), \tag{A107}$$

$$= \beta(c_j - c_i)\left(\frac{1}{c_i^{\text{ext}}} - \frac{1}{c_j^{\text{ext}}}\right). \tag{A108}$$

which has the sign of $c_i - c_j$ if $c_i^{\text{ext}} \geq c_j^{\text{ext}}$. Thus, we necessarily have $c_1 \geq c_2 \geq c_3 \geq \ldots \geq c_q$. In turn, defining $c_1', \ldots, c_p'$ as the transposition of $c_1, \ldots, c_p$ such that $c_i' = c_j$ and $c_j' = c_i$, with $2 < i \leq q$ and $j > q$, one can see that relation **Equation (A103)** implies that

$$E_{g,\mathcal{M}_q}(c_1',\ldots,c_p') - E_{g,\mathcal{M}_q}(c_1,\ldots,c_p) \tag{A109}$$

$$= \beta(c_j' - c_j)\left(\frac{1}{c_1} + \frac{1}{c_1^{\text{ext}}}\right) + \beta\frac{c_i' - c_i}{c_i^{\text{ext}}}, \tag{A110}$$

$$= \beta(c_i - c_j)\left(\frac{1}{c_1} + \frac{1}{c_1^{\text{ext}}} - \frac{1}{c_i^{\text{ext}}}\right), \tag{A111}$$

which has the sign of $c_i - c_j$ as we have $1/c_1 + 1/c_1^{\text{ext}} \geq 1/c_i^{\text{ext}}$ by **Equation (A102)**. Thus, we necessarily have $c_1 \geq c_2 \geq c_3 \geq \ldots \geq c_q \geq \max_{j>q} c_j$.

Finally, at fixed $c_1, \ldots c_q$, the sets defined by $E_{g,\mathcal{M}}(c_{q+1}, \ldots, c_p) < E$ are (convex) $(p-q)$-dimensional planes, implying the unicity of the contact point $c_{q+1}, \ldots, c_p$ with the convex sets $\mathcal{C}_g$ for optimal strategies. Together with the symmetry by permutations of $c_{q+1}, \ldots, c_p$, this

uniqueness directly implies $c_{q+1} = c_{q+2} = \ldots = c_p$. This concludes our proof that for external concentrations such that $c_1^{\text{ext}} > c_2^{\text{ext}} \geq c_3^{\text{ext}} \geq \ldots \geq c_p^{\text{ext}}$, the internal concentrations of optimal strategies satisfy $c_1 \geq c_2 \geq \ldots \geq c_{q+1} = \ldots = c_p$.

## Optimal pure-converter strategy

In this section, we specify the optimal pure-converter strategy in our geometric setting, which proves crucial to discard the occurrence of degenerate cartels for general growth functions, which includes the harmonic-mean model.

Consider an optimal pure-converter strategy importing building block 1 with $c_1^{\text{ext}} = \max_j c_j^{\text{ext}}$. We have established that, for this strategy to be optimal in metabolic class $\mathcal{M}_1$, we necessarily have $c_2 = \ldots = c_p$. Denoting $c = c_2 = \ldots = c_p$, such a strategy growing at rate $g$ requires an enzyme budget

$$E_{g,\mathcal{M}_1}(c_1, c) = (p-1)\big(g + \beta(c - c^{\text{ext}})\big)\left(\frac{1}{c_1} + \frac{1}{c_1^{\text{ext}}}\right) + \frac{g + \beta c_1}{c_1^{\text{ext}}} - \beta, \qquad \text{(A112)}$$

where we have defined $c^{\text{ext}} = \sum_{j>1} c_j^{\text{ext}}/(p-1) \leq c_1^{\text{ext}}$. As an optimal strategy necessarily utilizes all its enzyme budget, i.e. $E_{g,\mathcal{M}_1} = E$, we deduce from the above relationship that

$$c = c^{\text{ext}} + \frac{1}{\beta}\left(\frac{1}{p-1}\left(\frac{c_1 c_1^{\text{ext}}}{c_1 + c_1^{\text{ext}}}\right)\left(E + \beta - \frac{g + \beta c_1}{c_1^{\text{ext}}}\right) - g\right). \qquad \text{(A113)}$$

The function $c = c_g(c_1)$ is the boundary of the set $\mathcal{E}_{g,\mathcal{M}_1}(c_1, c) \leq E$. Because $c_g(c_1)$ is concave in $c_1$ for $c \leq c_1 \leq 0$, the contact point $(c, c_1)$ with the (convex) set $g(c_1, c, \ldots, c) \geq g$ is unique. Therefore, there is a unique optimal pure-converter strategy.

In principle, the contact point corresponding to that optimal strategy either occurs at an endpoint $c = c_1$ or for $c < c_1$ as an interior point. However, if $g$ is symmetric differentiable, the level sets of $g(c_1, c, \ldots, c)$ in the $(c_1, c)$-plane have a slope $-1/(p-1)$ on the diagonal $c = c_1$, whereas one can check that the slope of $c_g(c_1)$ is larger than $-1/(p-1)$ on the diagonal $c = c_1$. Actually, one can show that the latter slope tends toward $-1/(p-1)$ from above for increasing enzyme budget $E$. This relation between slopes on the diagonal $c = c_1$, together with the convexity properties of $g(c_1, c, \ldots, c)$ and $c_g(c_1)$, implies that the contact point is always an interior point: for optimal pure-converter strategies, we always have $c_1 > c_2 = \ldots = c_p$.

The fact that $c_1 > c_2 = \ldots = c_p$ at optimum does not necessarily holds if $g(c_1, \ldots, c_p)$ is non-differentiable when $c_i = c_j$ for $i \neq j$, as in the minimum model. Indeed, for the minimum model, the slope of the level sets of $g(c_1, c, \ldots, c)$ to the right of the diagonal $c_1 = c$ is zero and for large enough enzyme budget $E$, the contact point is an endpoint. Then we have $c_1 = c_2 = \ldots = c_p$ at optimum. This case of equal internal concentrations in optimal strategies is the reason for the occurrence of degenerate cartels with pure-converter strategies for the minimum model.

## Appendix 6: The selection of cartels for generic growth functions

In this section, we characterize the emergence of microbial cartels for growth-rate functions satisfying $g(c_1, \ldots, c_p) < \lambda \min(c_1, \ldots, c_p)$ for some $\lambda > 0$ and having the quasi-concave property, which includes the harmonic-mean model. We distinguish between external building-block concentrations that satisfy (i) a strict ordering relation, e.g. $c_1^{\mathrm{ext}} > c_2^{\mathrm{ext}} > \ldots > c_p^{\mathrm{ext}}$, or (ii) a degenerate ordering relation with equality between the concentrations of the most abundant building blocks, e.g. $c_1^{\mathrm{ext}} = \ldots = c_q^{\mathrm{ext}} > c_{q+1}^{\mathrm{ext}} > \ldots > c_p^{\mathrm{ext}}$, with $1 < q < p$. We also introduce the graph structure of microbial cartels in order to define a notion of neigborhood between cartels.

## Microbial cartels for strict ordering

Here, we characterize the structure of microbial cartels arising for external concentrations satisfying $c_1^{\mathrm{ext}} > c_2^{\mathrm{ext}} \geq \ldots \geq c_p^{\mathrm{ext}}$ for everywhere differentiable growth-rate function $g(c_1, \ldots, c_p)$ (as for the harmonic-mean model). Specifically, we show that cartels exist for external concentrations $c_1^\star, \ldots, c_p^\star$ such that for all metabolic class $\mathcal{M}_i$, $1 \leq i \leq p$, we have

$$\sup_{\sigma \in \mathcal{M}_i} g_\sigma(c_1^\star, \ldots, c_p^\star) = \delta, \tag{A114}$$

and such that the following order relations holds:

$$\frac{1}{c_1^{(1)}} \leq \frac{1}{c_2^\star} - \frac{1}{c_1^\star} < \frac{1}{c_1^{(2)}} \leq \frac{1}{c_3^\star} - \frac{1}{c_1^\star} < \frac{1}{c_1^{(3)}} \leq \ldots < \frac{1}{c_1^{(p)}}. \tag{A115}$$

In particular, observe that the set of steady-state external concentrations $c_1^\star > c_2^\star > \ldots > c_p^\star$ is strictly ordered. We prove the above proposition by an iterative procedure which requires to first establish that there is a unique optimal pure-converter strategy with internal concentrations satisfying $c_1 > c_2 = \ldots = c_p$.

First, observe that as import from the external pool is the only source of internal building block 1 for metabolic class $\mathcal{M}_1$, the optimal growth rate

$$\sup_{\sigma \in \mathcal{M}_1} g_\sigma(c_1^{\mathrm{ext}}, \ldots, c_p^{\mathrm{ext}}), \tag{A116}$$

defines an increasing function of $c_1^{\mathrm{ext}}$ that is zero when $c_1^{\mathrm{ext}} = 0$. Then, by uniqueness of the optimal strategy in metabolic class $\mathcal{M}_1$ for fixed concentrations $c_2^{\mathrm{ext}}, \ldots, c_p^{\mathrm{ext}}$, there is a unique $c_1^\star > 0$ such that

$$\sup_{\sigma \in \mathcal{M}_1} g_\sigma(c_1^\star, c_2^{\mathrm{ext}}, \ldots, c_p^{\mathrm{ext}}) = \delta, \quad i < q, \tag{A117}$$

thereby defining a function $c_1^\star(c_2^{\mathrm{ext}}, \ldots, c_p^{\mathrm{ext}})$. Moreover, the internal concentrations $c_1, \ldots, c_p$ of the optimal strategy in $\mathcal{M}_1$, correspond to an interior point in the sense that $c_1 < c_2 = \ldots = c_p$.

Suppose there are $q - 1$ functions $c_1^\star(c_q^{\mathrm{ext}}, \ldots, c_p^{\mathrm{ext}}), \ldots, c_{q-1}^\star(c_q^{\mathrm{ext}}, \ldots, c_p^{\mathrm{ext}})$ such that for all $c_q^{\mathrm{ext}}, \ldots, c_p^{\mathrm{ext}}$, we have

$$\sup_{\sigma \in \mathcal{M}_i} g_\sigma(c_1^\star, \ldots, c_{q-1}^\star, c_q^{\text{ext}}, \ldots, c_p^{\text{ext}}) = \delta, \quad i < q, \tag{A118}$$

and such that we have the following order relation

$$\frac{1}{c_1^{(1)}} \le \frac{1}{c_2^\star} - \frac{1}{c_1^\star} < \frac{1}{c_1^{(2)}} \le \frac{1}{c_3^\star} - \frac{1}{c_1^\star} < \frac{1}{c_1^{(3)}} \le \ldots < \frac{1}{c_1^{(q-1)}}, \tag{A119}$$

$$c_{q-1}^{(q-1)} = \ldots = c_p^{(q-1)}, \tag{A120}$$

where $c_i^{(q')}$ denotes the internal concentration of building block $i$ in the optimal strategy of metabolic class $\mathcal{M}_{q'}$. As import from the external pool is the only source of internal building block $q'$ for metabolic class $\mathcal{M}_{q'}$, all strategies in $\mathcal{M}_{q'}$, $q' \ge q$, have zero growth rate if $c_q^{\text{ext}} = 0$.

For fixed $c_{q+1}^{\text{ext}}, \ldots, c_p^{\text{ext}}$, one can increase $c_q^{\text{ext}}$ until

$$\frac{1}{c_q^{\text{ext}}} = \frac{1}{c_1^{(q-1)}} + \frac{1}{c_1^\star}, \tag{A121}$$

since $c_1^{(q-1)}$ and $c_1^\star$ are both bounded from below. To see that $c_1^{(q-1)}$ and $c_1^\star$ are bounded from below, observe that the growth-rate function for the harmonic-mean model satisfies

$$g(c_1, \ldots, c_p) \le \gamma \min\left(\frac{c_1}{b_1}, \ldots, \frac{c_p}{b_p}\right). \tag{A122}$$

Thus, independently of its metabolic strategy, a cell that grows at rate $\delta$ has internal concentrations that are bounded below by $c_i/b_i > \delta$ for all $i$. Moreover, for all metabolic classes $\mathcal{M}_{q'}$, $1 \le q' \le p$, the only source of internal building block 1 is via import form the external pool. In particular, we have $(\alpha_1 + \beta)c_1^{\text{ext}} - \beta c_1 > 0$, which implies that the concentrations $c_1^{\text{ext}}$ for which a cell can grow at rate $\delta$ are bounded below by

$$c_1^{\text{ext}} > \frac{\beta c_1}{\alpha_1 + \beta} > \frac{\beta c_1}{E + \beta} > \frac{\beta}{E + \beta}\frac{\delta}{b_1}. \tag{A123}$$

This shows that, as function of $c_{q+1}^{\text{ext}}, \ldots, c_p^{\text{ext}}$, both $c_1^{(q-1)}$ and $c_1^\star$ are bounded from below by positive constants that are independent of $c_q^{\text{ext}}$, which justifies that equality **Equation (A121)** holds for some $c_q^{\text{ext}} > 0$.

When equality **Equation (A121)** holds, we know that for any strategy in metabolic class $\mathcal{M}_{q-1}$, there is a strategy in metabolic class $\mathcal{M}_q$ that grows at the same rate, i.e. $\delta$. Then, one can consider $c_q^\star$, the smallest concentration $c_q^{\text{ext}}$ for which a metabolic strategy in $\mathcal{M}_q$ grows with rate $\delta$:

$$c_q^\star = \inf\left\{c_q^{\text{ext}} > 0 \,\middle|\, \sup_{\sigma \in \mathcal{M}_q} g_\sigma(c_1^\star, \ldots, c_{q-1}^\star, c_q^{\text{ext}}, \ldots, c_p^{\text{ext}}) = \delta\right\}, \tag{A124}$$

For such $c_q^\star$, metabolic classes $\mathcal{M}_{q-1}$ and $\mathcal{M}_q$ are jointly optimal, which implies by inequalities **Equation (A102)** that

$$\frac{1}{c_1^{(q-1)}} \le \frac{1}{c_q^\star} - \frac{1}{c_1^\star} \le \frac{1}{c_1^{(q)}}. \tag{A125}$$

Moreover, $c_q^\star$ is thus-defined as a function of $c_{q+1}^{\text{ext}}, \ldots, c_p^{\text{ext}}$, which allows one to define $c_1^\star, \ldots, c_{q-1}^\star$ as functions of $c_{q+1}^{\text{ext}}, \ldots, c_p^{\text{ext}}$ alone by

$$c_{q'}^\star \overset{\text{def}}{=} c_{q'}^\star \left( c_q^\star(c_{q+1}^{\text{ext}}, \ldots, c_p^{\text{ext}}), c_{q+1}^{\text{ext}}, \ldots, c_p^{\text{ext}} \right), \quad q' < q. \tag{A126}$$

Finally, we show that the last inequality in **Equation (A125)** is strict by a geometric argument. As the contact point between $\mathcal{C}_g$ and $\mathcal{E}_{g,\mathcal{M}_{q-1}}$ is an interior point with $c_1 > \ldots > c_{q-1} = \ldots = c_p$, the gradient of the growth-rate function $g(c_1, \ldots, c_p)$ is proportional to the gradient of the required enzyme budget $E_{g,\mathcal{M}_{q-1}}(c_1, \ldots, c_p)$ at the contact point. However, we have

$$\nabla E_{g,\mathcal{M}_q}\Big|_{c_1, \ldots, c_p} = \begin{bmatrix} \dfrac{\beta}{c_1^\star} - \dfrac{1}{c_1^2} \displaystyle\sum_{j=1, j>q} \left( g + \beta(c_j - c_j^{\text{ext}}) \right) \\ \dfrac{\beta}{c_2^\star} \\ \vdots \\ \dfrac{\beta}{c_q^\star} \\ \beta\left( \dfrac{1}{c_1^\star} + \dfrac{1}{c_1} \right) \\ \vdots \\ \beta\left( \dfrac{1}{c_1^\star} + \dfrac{1}{c_1} \right) \end{bmatrix}. \tag{A127}$$

For fixed external concentrations $c_{q+1}^{\text{ext}}, \ldots, c_p^{\text{ext}}$, if $c_q^{\text{ext}}$ is such that equality **Equation (A121)** holds, i.e.

$$c_q^{\text{ext}} = \left( \frac{1}{c_1^{(q)}} + \frac{1}{c_1^\star} \right)^{-1}, \tag{A128}$$

there is a metabolic strategy in metabolic class $\mathcal{M}_q$ that grows at rate $\delta$ with the same internal concentrations as the optimal strategy in $\mathcal{M}_q$ (which also grows at rate $\delta$). At these internal concentrations, $\nabla E_{g,\mathcal{M}_q}$ is proportional to $\nabla g$ and we have

$$\left[ \nabla E_{g,\mathcal{M}_{q-1}} - \nabla E_{g,\mathcal{M}_q} \right]_j = \begin{cases} \left( g + \beta(c_q - c_q^{\text{ext}}) \right) / \left( c_1^{(q-1)} \right)^2 & \text{if } j = 1, \\ 0 & \text{if } j \neq 1. \end{cases} \tag{A129}$$

The fact that $\left[ \nabla E_{g,\mathcal{M}_{q-1}} - \nabla E_{g,\mathcal{M}_q} \right]_1 > 0$ implies that the contact point between sets $\mathcal{E}_{g,\mathcal{M}_{q-1}}$ and $\mathcal{C}_g$ cannot be a contact point between sets $\mathcal{E}_{g,\mathcal{M}_q}$ and $\mathcal{C}_q$, and that $c_q^\star$, as defined by **Equation (A124)**, is such that

$$c_q^{\text{ext}} < \left( \frac{1}{c_1^{(q)}} + \frac{1}{c_1^\star} \right)^{-1}. \tag{A130}$$

In other words, the last inequality in **Equation (A125)** is strict and the contact point $\mathcal{E}_{g,\mathcal{M}_q}$ and $\mathcal{C}_q$ between is an interior point in the sense that $c_1 > c_2 > \ldots > c_q = \ldots = c_p$.

Iterating on the above argument for $q = 2, \ldots, p$ demonstrates the existence of a sequence $c_1^\star > \ldots > c_p^\star$ for which there are $p$ jointly optimal metabolic strategies, each belonging to metabolic class $\mathcal{M}_i$, $1 \leq i \leq p$, thereby forming a microbial cartel.

## Microbial cartels for degenerate ordering

Above, we characterized the cartels that arise for strict order of the external concentrations $c_1^{\text{ext}} > \ldots > c_2^{\text{ext}} > \ldots > c_p^{\text{ext}}$. Here, we show that other cartels emerge at the frontier between sectors defined by distinct orders, i.e. when the most abundantly available building blocks can have the same external concentration.

Consider an order relation of the type $c_1^{\text{ext}} = \ldots = c_q^{\text{ext}} > c_{q+1}^{\text{ext}} \ldots > c_p^{\text{ext}}$ with $1 \leq q < p$. The structure of the 'degenerate cartels' arising for the above ordering can be infer from the cartels associated to strict ordering by using two facts:

i.   Optimal metabolic classes necessarily import all building block $1 \leq i \leq q$. To see this, consider block $i$ and block $j$ among the most abundant building blocks $1 \leq i, j \leq q$. By inequality **Equation (A101)**, for a metabolic class converting $i$ into $j$ to be optimal, we necessarily have

$$\frac{1}{c_i} + \frac{1}{c_i^{\text{ext}}} \leq \frac{1}{c_j^{\text{ext}}}, \tag{A131}$$

which contradicts that $c_i^{\text{ext}} = c_j^{\text{ext}}$. Thus, optimal metabolic classes import all building block $1 \leq i \leq q$.

ii.  There is a degeneracy in the choice of the precursor. This point follows from the fact that optimal strategies use a single most abundant building block as precursor for converted building blocks. For order of the type $c_1^{\text{ext}} = \ldots = c_q^{\text{ext}} > c_{q+1}^{\text{ext}} \ldots > c_p^{\text{ext}}$, optimal metabolic classes can use any block $1 \leq i \leq q$ as a precursor.

Using facts (i) and (ii), one can adapt the argument of the above section for strict order $c_1^{\text{ext}} > \ldots > c_2^{\text{ext}} > \ldots > c_p^{\text{ext}}$ to characterize cartels for order of the type $c_1^{\text{ext}} = \ldots = c_q^{\text{ext}} > c_{q+1}^{\text{ext}} \ldots > c_p^{\text{ext}}$. In fact, one can show that for some external concentrations $c_1^\star = \ldots = c_q^\star > c_{q+1}^\star \ldots > c_p^\star$, there are degenerate cartels comprising strategies that use block $q'$, $1 \leq q' \leq q$ as precursor for conversion of the $q''$ least abundantly available block with $q + q'' \leq p$. In particular, one can consider degenerate cartels as the merger of $q$ consortia that share an identical pure-importer strategy, but that each comprise strategies using a specific block $q'$, $1 \leq q' \leq q$ as precursor for converted blocks. Moreover, using the same notations as for cartels associated to strict ordering, the consortia using block $q'$, $1 \leq q' \leq q$ as precursor includes metabolic classes $\mathcal{M}_q, \ldots, \mathcal{M}_p$, and we have

$$\frac{1}{c_{q'}^{(q)}} \leq \frac{1}{c_{q+1}^\star} - \frac{1}{c_{q'}^\star} < \frac{1}{c_{q'}^{(q+1)}} \leq \frac{1}{c_{q+2}^\star} - \frac{1}{c_{q'}^\star} < \frac{1}{c_{q'}^{(q+2)}} \leq \ldots < \frac{1}{c_{q'}^{(p)}}. \tag{A132}$$

where $c_{q'}^{(q'')}$ denotes the internal concentration of block $q'$ for the strategy that converts block $q'$ into blocks $\geq q''$. Note that $c_{q'}^{(q'')}$ does not depends on which block $q'$ is used as a precursor since $c_1^\star = \ldots = c_q^\star$. Thus, relation **Equation (A132)** characterizes all the order properties of internal and external concentrations in degenerate microbial cartels. Interestingly, degenerate cartels comprise more than $p$ distinct cell types for $p > 3$. Actually, if the $q$ most abundant building blocks have equal concentration, there are $1 + q(p - q)$ coexisting strategies: there are $p - q$ possible sets of converted building blocks, that can each be

converted from one of the $q$ most abundant blocks, yielding $q(p-q)$ mixed strategies in addition to the pure-importer strategy.

## Graph structure of microbial cartels

To establish the existence of cartels, we have distinguished between non-degenerate cartels, i.e. for strictly ordered external building-block concentrations, and degenerate cartels, i.e. for degenerate orders of external building-block concentrations. Here, we introduce the degree of degeneracy to categorize these cartels and to define a notion of neighborhood for cartels.

Cartel-specific concentrations $c_1^\star, \ldots, c_p^\star$ are represented by isolated point $c^\star$ in the space of external building concentrations, defined as the intersection of $p$ hypercurves. Indeed, for cartels associated to strict ordering, the concentrations $c_1^\star > \ldots > c_p^\star$ satisfy $p$ equations

$$\sup_{\sigma \in \mathcal{M}_q} g_\sigma(c_1^\star, \ldots, c_p^\star) = \delta, \quad \text{for all} \quad 1 \le q \le p, \tag{A133}$$

where $\mathcal{M}_q$, $q<p$, is the metabolic class that convert block 1 into blocks $q'>q$ and where $\mathcal{M}_p$ is the pure-importer class. For degenerate cartels associated with $q-1$ equalities between the concentrations of the most abundant building blocks, the concentrations $c_1^\star = \ldots = c_q^\star > \ldots > c_p^\star$ satisfy

$$\sup_{\sigma \in \mathcal{M}_{q'}} g_\sigma(c_1^\star, \ldots, c_p^\star) = \delta, \quad \text{for all} \quad q \le q' \le p, \tag{A134}$$

where $\mathcal{M}_{q'}$ is the metabolic class that convert one of the most abundant building block into blocks $q''>q'$ and where $\mathcal{M}_p$ is the pure-importer class. We define the order of a cartel to be one for cartel associated to strict ordering and to be the number of equally abundant building blocks for other cartels.

The set of concentrations $c_1^{\text{ext}}, \ldots, c_p^{\text{ext}}$ that satisfy $p-1$ equations among the $p$ equations defining a cartel define a one-dimensional path in the space of external building-block concentrations. Importantly, we can show that each of these paths passes through at most two cartel-specific concentrations. For instance, consider the cartel of order 1 associated to $c_1^\star > \ldots > c_p^\star$, represented by the point $c^\star$, and the path $\mathcal{P}$ defined by

$$\sup_{\sigma \in \mathcal{M}_{q' \ne q}} g_\sigma(c_1^\star, \ldots, c_p^\star) = \delta. \tag{A135}$$

The point $c^\star$, which lies on $\mathcal{P}$, divides the path in two rays depending on whether $\sup_{\sigma \in \mathcal{M}_q} g_\sigma(c_1^{\text{ext}}, \ldots, c_p^{\text{ext}}) > \delta$. Actually, among all possible cartels, there is another cartel (and only one) that is compatible with the $p-1$ **Equation (A135)**: the cartel associated with $c_1^\star > \ldots > c_{q+1}^\star > c_q^\star > \ldots > c_p^\star$ represented by the point $c^{\star\prime}$. The point $c^{\star\prime}$ also divides the path $\mathcal{P}$ in two rays depending on whether $\sup_{\sigma \in \mathcal{M}_q'} g_\sigma(c_1^{\text{ext}}, \ldots, c_p^{\text{ext}}) > \delta$, where $\mathcal{M}_q'$ is the metabolic class that converts 1 into blocks $q, q+2, \ldots, p$. By optimality of both cartels, we necessarily have $\sup_{\sigma \in \mathcal{M}_q'} g_\sigma(c^\star) < \delta$ and $\sup_{\sigma \in \mathcal{M}_q} g_\sigma(c^{\star\prime}) < \delta$. Thus, the conditions $\sup_{\sigma \in \mathcal{M}_q} g_\sigma(c_1^{\text{ext}}, \ldots, c_p^{\text{ext}}) < \delta$ and $\sup_{\sigma \in \mathcal{M}_q'} g_\sigma(c_1^{\text{ext}}, \ldots, c_p^{\text{ext}}) < \delta$ define the segment of path $\mathcal{P}$ that links the points $c^\star$ and $c^{\star\prime}$ representing both cartels. In fact, this segment is the set of concentrations for which the $p-1$ metabolic classes shared by both cartels are jointly optimal.

The above reasoning can be generalize to cartel of any order, yielding the following results:

**Cartels of order** 1: There are paths connecting the cartel of order 1 associated with $c_1^\star > \ldots > c_{q'}^\star > c_{q'+1}^\star > \ldots > c_p^\star$, to $p-2$ cartels of order 1 associated with $c_1^\star > \ldots > c_{q'+1}^\star > c_{q'}^\star > \ldots > c_p^\star$ for $1 < q' < p$. There is a path connecting the cartel of order 1 associated with $c_1^\star > c_2^\star > \ldots > c_p^\star$, to the cartel of order 2 associated with $c_1^\star = c_2^\star > \ldots > c_p^\star$.

**Cartels of order** 2: There are paths connecting the cartel of order 2 associated with $c_1^\star = c_2^\star > \ldots > c_p^\star$ to the cartels of order 1 associated with $c_1^\star > c_2^\star > \ldots > c_p^\star$ and with $c_2^\star > c_1^\star > \ldots > c_p^\star$. There are paths connecting the cartel of order 2 associated with $c_1^\star = c_2^\star > \ldots > c_{q'}^\star > c_{q'+1}^\star > \ldots > c_p^\star$ to the $p-3$ cartels of order 2 associated with $c_1^\star = c_2^\star > \ldots > c_{q'+1}^\star > c_{q'}^\star > \ldots > c_p^\star$ for $2 < q' < p$. There is a path connecting the cartel of order 2 associated with $c_1^\star = c_2^\star > \ldots > c_p^\star$ to the cartel of order 3 associated with $c_1^\star = c_2^\star = c_3^\star > \ldots > c_p^\star$.

$$\vdots \quad \vdots \quad \vdots \quad \vdots \quad \vdots \quad \vdots \quad \vdots \quad \vdots$$

**Cartels of order** $q$: There are paths connecting the cartel of order $q$ associated with $c_1^\star = \ldots = c_q^\star > \ldots > c_p^\star$ to the $q$ cartels of order $q-1$ associated with $q-2$ equalities among $c_1^\star, \ldots, c_q^\star$. There are paths connecting the cartel of order $q$ associated with $c_1^\star = \ldots = c_q^\star > \ldots > c_{q'}^\star > c_{q'+1}^\star > \ldots > c_p^\star$ to the $p-q-1$ cartels of order $q$ associated with $c_1^\star = \ldots = c_q^\star > \ldots > c_{q'+1}^\star > c_{q'}^\star > \ldots > c_p^\star$ for $q < q' < p$. There is a path connecting the cartel of order $q$ associated with $c_1^\star = \ldots = c_q^\star > \ldots > c_p^\star$ to the cartel of order $q+1$ associated with $c_1^\star = \ldots = c_{q+1}^\star > \ldots > c_p^\star$.

$$\vdots \quad \vdots \quad \vdots \quad \vdots \quad \vdots \quad \vdots \quad \vdots \quad \vdots$$

**Cartels of order** $p-1$: There are paths connecting the cartel of order $p-1$ associated with $c_1^\star = \ldots = c_{p-1}^\star > c_p^\star$ to the $p-1$ cartels of order $p-2$ associated with $p-2$ equalities among $c_1^\star, \ldots, c_{p-1}^\star$.

In particular, cartels of order 1 and $p-1$ are connected to $p-1$ cartels, while other cartels are connected to $p$ cartels. Observe that a cartel of order $q$ only joins cartels of similar order, and if possible, cartels of order $q-1$ or $q+1$. Moreover, on can verify that when a path joins two cartels of order $q$, $1 \leq q < p$, these cartels share $1 + q(p-q-1)$ strategies, whereas when a path joins a cartel of order $q$ to a cartel of order $q-1$, $1 < q < p$, these cartels share $1 + (q-1)(p-q)$ strategies. We define the graph structure of microbial cartels by considering the points representing cartel-specific concentrations as nodes and by considering paths joining cartel-specific concentrations as edges. This graph allows us to define a notion of neighborhood for cartels: two cartels are neighbors if the are connected by an edge.

## Appendix 7: Benefit of microbial consortia

In this section, we establish that, although microbial cartels exist at isolated points in the space of external building-block concentrations, cartels emerge for generic supply conditions. Then, we characterize the geometry of cartel-specific supply sectors in the space of supply rate, establishing that cartels emerge for generic building-block supply. Finally, we show that steady-state microbial consortia maximize biomass yield at fixed supply, thereby showing that microbial cartels of competing strategies achieve a collective optimum.

### Supply sectors as polyhedral cones

Here, we introduce the notion of supply sectors, i.e. the set of supply rates values for which competitive microbial dynamics yield the same steady-state external concentrations. We then discuss the geometric properties of these supply sectors explaining why the emergence of cartels is generic.

Consider an arbitrary set of steady-state external concentrations $c_1^\star, \ldots, c_p^\star$ satisfying $G(c_1^\star, \ldots, c_p^\star) = \delta$. For such concentrations, at least one cell type survives and the maximum number of surviving cell types is attained for cartel-specific values, which are isolated points in the space of external concentrations. Surviving cell types $\sigma \in \Sigma^\star$ jointly achieve the optimal growth rate for external building-block concentrations $c_1^\star, \ldots, c_p^\star$. The per-cell fluxes experienced by these cell types $\phi_{i,\sigma}^\star$ take fixed values that can be obtained via *Equation (A11)*. Then, the resulting flux-balance equations for extracellular building blocks,

$$s_i(\{n_\sigma\}) = \mu c_i^\star + \frac{v}{V - Nv} \sum_{\sigma \in \Sigma^\star} n_\sigma \phi_{i,\sigma}^\star, \tag{A136}$$

yield the supply rates as functions of the populations $n_\sigma > 0$, $\sigma \in \Sigma^\star$. Accordingly, for supply rates

$$s_i(\{n_\sigma\}_{\sigma \in \Sigma^\star}), \quad \text{with} \quad n_\sigma > 0 \quad \text{for all} \quad \sigma \in \Sigma^\star, \tag{A137}$$

the optimal strategies $\sigma^\star \in \Sigma^\star$ are the only steady-state strategies that cannot be invaded by metabolic variants. In particular, specific sets of supply rate, called supply sectors, are associated to the stable dominance of an optimal cell type or of a consortia of optimal cell types.

Mathematically, *Equation (A136)* defines the supply sector associated with the dominance of a consortium $\Sigma^\star$ as a polyhedral convex cone in the $p$-dimensional space of supply rates. Such a cone is entirely determined by its vertex $\mu c^\star$ (with coordinates $\mu c_i^\star$) and its generating set of vectors $\{\phi_\sigma^\star\}_{\sigma \in \Sigma^\star}$ (with coordinates $\phi_{i,\sigma}^\star$). Observe that for all vectors $\phi_\sigma^\star$, the conservation of building blocks at steady state implies that $\phi_\sigma^\star \cdot 1 = p(1 - f)\delta > 0$. We will say that the emergence of a consortia is generic if its associated supply sector has finite measure in the space of supply rates. A supply sector has finite measure when it is a polyhedral cone with full measure, i.e. its set of generating vectors contains a family of $p$ independent vectors.

The defining property of cartels is that they are associated with supply cones that have full dimension in the space of supply rates. To see this, consider a cartel of order one, e.g. associated with the order $c_1^\star > \ldots > c_p^\star$. We know that such a cartel contains $p$ distinct metabolic classes $\mathcal{M}_q$, $1 \leq q \leq p$, where $q$ denotes the number of imported building blocks. Moreover, using *Equation (A9)*, the steady-state per-cell fluxes for optimal strategies in $\mathcal{M}_q$, $1 \leq q \leq p$ can be specified as

$$\phi^{\star}_{i,q} = \begin{cases} (1-f)\delta + \sum_{k>q} \kappa^{(q)}_{k1} c^{(q)}_1 & \text{if} \quad i=1, \\ (1-f)\delta & \text{if} \quad 1<i\leq q, \\ (1-f)\delta - \kappa^{(q)}_{i1} c^{(q)}_1 & \text{if} \quad i>j, \end{cases} \tag{A138}$$

where $c^{(q)}_1$ and $\kappa^{(q)}_{k1}$ denotes the internal concentration of block 1 and the enzyme activity associated with conversion of block 1 into block $k$, respectively, in metabolic class $\mathcal{M}_q$. Thus, the set of generating vectors can be written

$$\phi^{\star(q)} = (1-f)\delta 1 + \sum_{k>q} \kappa^{(q)}_{k1} c^{(q)}_1 \eta_k, \tag{A139}$$

i.e. as a linear combination of $1$, the vector with unit components, and of the vectors $\eta_k$ defined for $k>1$ by

$$\eta_{i,k} = \begin{cases} 1 & \text{if} \quad i=1, \\ -1 & \text{if} \quad i=k, \\ 0 & \text{otherwise}. \end{cases} \tag{A140}$$

*Equation (A139)* shows that the coefficient matrix of the vectors $\phi_p, \phi_1 - \phi_p, \phi_2 - \phi_p, \ldots \phi_{p-1} - \phi_p$ decomposed in the basis $1, \eta_2, \ldots, \eta_p$ is triangular with positive diagonal elements. In particular, the vector space generated by $\phi_p, \phi_1 - \phi_p, \phi_2 - \phi_p, \ldots \phi_{p-1} - \phi_p$, or equivalently by $\phi_1, \ldots, \phi_p$, has full dimension. Thus, cartels of order one yield sets of independent vectors $\phi_q$ with full dimension in the space of supply rate, and therefore has a supply sector with finite measure. This shows that cartels of order one, which contain uniquely defined strategies, can arise for generic supply conditions. By similar arguments from linear algebra, one can show that the vector space generated by the vectors $\phi^{\star}$ has full dimension for cartels of any order.

At steady-state concentrations $c^{\star}_1, \ldots, c^{\star}_p$ for which no cartel arises, consortia can still emerge. Such consortia also define polyhedral cones, but these cones are generated by subsets of metabolic classes belonging to cartels. One can show that these cones have zero measure: every family of $p$ flux vectors associated with steady-state external concentrations for which no cartels arise is linearly dependent. The fact that a supply cone does not have full dimension implies that the associated consortium only arises for specific supply rates, belonging to a set of zero measure in the space of supply rate. In particular, although the metabolic classes of consortia that are not cartels may be the same for different supply rates, the steady-state external concentrations and the optimal strategies (the specific enzyme distributions) generally differ for different supply rates.

## Facets of supply sectors

As the supply sector associated with cartel $\Sigma^{\star}$ is a $p$-dimensional polyhedral cones, its facets are $(p-1)$-dimensional faces defined by subsets of the generating vectors $\{\phi_{\sigma}\}_{\sigma \in \Sigma^{\star}}$. The generating subset of a particular facet can be identified by analysis of the $p$ equations that define cartel-specific external equations: for a cartel of order $q$, these are $q-1$ equalities between concentrations and $p-q+1$ growth rate equalities (see Section Graph structure of microbial cartels). In fact, one can show that a supply sector has as many facets as there are paths $\mathcal{P}$ defined in the space of external concentrations by only $p-1$ of the $p$ equations specifying the corresponding cartel. Indeed, for any point $c$ on a path $\mathcal{P}$, there is a consortium, denoted $\Sigma(c)$, which comprises a subset of the metabolic classes of $\Sigma^{\star}$ and whose supply sectors is a $(p-1)$-dimensional cone generated by $\{\phi^{\star}_{\sigma}\}_{\sigma \in \Sigma(c)}$. Then, when $c$

tends to $c^\star$ on $\mathcal{P}$, the $(p-1)$ dimensional cone associated with $\Sigma(c)$ tends to a facet of the $p$-dimensional cone associated with cartel $\Sigma^\star$. This property actually defines the facet of a supply sector. As a result, we can specify the facets of a supply sector by enumerating the composition of consortia associated with the various paths emanating from the corresponding cartel.

To be more specific, consider a cartel $\Sigma^\star$ of order $q$ associated with $c_1 = \ldots = c_q > c_{q+1} > \ldots > c_p$. One can define the paths emanating from the point $c^\star$ associated with $\Sigma^\star$:

- There is one path, denoted $\mathcal{P}_0$, corresponding to satisfying $c_1 = \ldots = c_q$ and all the **Equation (A134)** except for the pure-importer metabolic class. Then, starting from the point $c^\star$ associated with $\Sigma^\star$, a decrease in $c_p$ along path $\mathcal{P}_{q'}$ causes the pure importer strategy to vanish at steady state. The path $\mathcal{P}_0$ extends until it reaches the boundary of the positive quadrant for $c_1^\star = \ldots = c_q^\star > c_{q+1}^\star = c_p^\star = 0$.
- There are $q$ paths, denoted $\mathcal{P}_{q'}$, $1 \le q' \le q$, corresponding to satisfying all the **Equation (A134)** and only $q-1$ equalities among $c_1 = \ldots = c_q$, i.e. there is $q'$ such that $1 \le q' \le q$ and $c_{q'} \ne c_{q''}$ for all $q'' \ne q'$, $1 \le q'' \le q$. Then, starting from the point $c^\star$, a decrease in $c_{q'}$ along path $\mathcal{P}_{q'}$ causes the $p-q$ metabolic classes of $\Sigma^\star$ that utilize block $q'$ as a precursor to vanish. These paths connect $\Sigma^\star$ to cartels of order $q-1$.
- There are $p - q - 1$ paths, denoted $\mathcal{P}_{q'}$, $q < q' < p$, corresponding to satisfying $c_1 = \ldots = c_q$ and all the **Equation (A134)** except for the growth rate equation $\sup_{\sigma \in \mathcal{M}_{q'}} g_\sigma = \delta$. Then, starting from the point $c^\star$, a decrease in $c_{q'}$ along path $\mathcal{P}_{q'}$ causes the $q$ strategies that import block $q'$ but convert block $q' + 1$ to vanish. The paths connect $\Sigma^\star$ to cartels of order $q$.
- There is one path, denoted $\mathcal{P}_p$, corresponding to satisfying $c_1 = \ldots = c_q$ and all the **Equation (A134)** except for the growth rate equation $\sup_{\sigma \in \mathcal{M}_q} g_\sigma = \delta$. Then, starting from the point $c^\star$, an increase in $c_{q+1}$ along path $\mathcal{P}_p$ causes the $q$ strategies that only imports blocks $1, \ldots q$ to vanish. This path connects $\Sigma^\star$ to cartels of order $q + 1$.

Thus we have enumerated the facets of the supply sectors associated with various cartels. In particular, cartels of order 1 and $p-1$ have $p$ facets, whereas cartels of order $q$, $1 < q < p-1$ have $p + 1$ facets.

## Space tiling by supply sectors

In the previous section, we have shown that every microbial consortium admits a supply sector that is a polyhedral convex cone. Cones that are associated with a cartels are the only ones with full dimension, thereby defining a supply sector with finite measure in the space of supply rate. Here, we characterize how cartel-specific supply sectors 'tile' the space of supply rates. Namely, we show that the vertices $\mu c^\star$ and the flux vectors $\{\phi_\sigma^\star\}_{\sigma \in \Sigma^\star}$ associated to cartels $\Sigma^\star$ specify a set of non-overlapping polyhedral convex cones that extends indefinitely for increasing supply rates. Moreover, neighboring cones of different orders have parallel facets, while neighboring cones of the same order have diverging facets.

To establish the above properties, we first show that if two distinct cartels $\Sigma^\star$ and $\Sigma^{\star\prime}$ are such that there is $i \ne j$ with $c_i^\star > c_j^\star$ and $c_i^{\star\prime} < c_j^{\star\prime}$, then one can always find an hyperplane separating supply sectors of $\Sigma^\star$ and $\Sigma^{\star\prime}$. Introducing the vector $\eta^{(ij)}$ defined by

$$\eta_k^{(ij)} = \begin{cases} 1 & \text{if } k = i, \\ -1 & \text{if } k = j, \\ 0 & \text{otherwise,} \end{cases} \tag{A141}$$

one can check that $\phi_\sigma \cdot \eta^{(ij)} \ge 0$ for all strategies $\sigma$ in $\Sigma^\star$, whereas $\phi_\sigma \cdot \eta^{(ij)} \le 0$ for all strategies $\sigma$ in $\Sigma^{\star\prime}$. Indeed, one can show the set of following inequalities. If a strategy $\sigma$ or $\sigma'$ imports both block $i$ and block $j$ without using either of them as precursors, we have

$\phi_\sigma \cdot \eta^{(ij)} = \phi_{\sigma'} \cdot \eta^{(ij)} = \delta(1-f) - \delta(1-f) = 0$. If a strategy $\sigma$ uses $i$ as a precursor ($c_i^\star > c_j^\star$), we have

$$\phi_\sigma \cdot \eta^{(ij)} \geq c_i \left( \sum_k \kappa_{ki} - \kappa_{ji} \right) = c_i \sum_{k \neq j} \kappa_{ki} > 0, \tag{A142}$$

whereas by symmetry, if a strategy $\sigma'$ uses block $j$ as a precursor ($c_i^{\star\prime} < c_j^{\star\prime}$), we have $\phi_{\sigma'} \cdot \eta^{(ij)} < 0$. If a strategy $\sigma$ imports block $i$, uses precursor $k \neq i$, and converts block $j$, ($c_i^\star > c_j^\star$), we have

$$\phi_\sigma \cdot \eta^{(ij)} = -\kappa_{jk} c_k > 0, \tag{A143}$$

whereas by symmetry, if a strategy $\sigma'$ imports block $j$, uses precursor $k \neq j$, and converts block $i$, ($c_i^{\star\prime} < c_j^{\star\prime}$), we have $\phi_{\sigma'} \cdot \eta^{(ij)} < 0$. If a strategy $\sigma$ or $\sigma'$ converts both blocks $i$ and $j$, we have

$$\phi_\sigma \cdot \eta^{(ij)} = \beta c_i^\star - \beta c_{i,\sigma} - (\beta c_j^\star - \beta c_{j,\sigma}) = \beta(c_i^\star - c_j^\star) > 0, \tag{A144}$$

$$\phi_{\sigma'} \cdot \eta^{(ij)} = \beta c_i^{\star\prime} - \beta c_{i,\sigma} - (\beta c_j^{\star\prime} - \beta c_{j,\sigma}) = \beta(c_i^{\star\prime} - c_j^{\star\prime}) < 0. \tag{A145}$$

It is easy to see that $(c^\star - c^{\star\prime}) \cdot \eta^{(ij)} = (c_i^\star - c_j^\star) - (c_i^{\star\prime} - c_j^{\star\prime}) > 0$, showing that the hyperplane passing through $(c^\star - c^{\star\prime})/2$ with normal vector $\eta^{(ij)}$ separates the supply sectors of $\Sigma^\star$ and $\Sigma^{\star\prime}$. Finally, notice that neighboring cartels of the same order are cartels for which the ordering of external building-block concentrations differs by a transposition, i.e. $c_i^\star \leftrightarrow c_j^\star$. The supply sectors of these cartels have opposite facets that are separated by an hyperplane normal to $\eta^{(ij)}$. Moreover, these facets diverges from that hyperplane with an angle prescribed by **Equation (A144)** and **Equation (A145)**. On can confirm that when the passive leak is negligible with respect to the enzyme budget $\beta \ll E$, $\beta(c_i^\star - c_j^\star)$ tends to zero and the facets become parallel.

Then, to prove that supply sectors are non-overlapping, we only need to show that distinct cartels $\Sigma^\star$ and $\Sigma^{\star\prime}$ with same degenerate orders, e.g. $c_1^\star = \ldots = c_q^\star > c_{q+1}^\star > \ldots > c_p^\star$ and $c_1^\star = \ldots = c_{q'}^\star > c_{q'+1}^\star > \ldots > c_p^\star$ with $q < q'$ have non-overlapping supply sectors. Introducing the vector $\eta^{(i)}$ defined by

$$\eta_k^{(i)} = \begin{cases} 1-p & \text{if } k = i, \\ 1 & \text{otherwise}, \end{cases} \tag{A146}$$

one can check by similar arguments as above that $\phi_\sigma \cdot \eta^{(q')} \geq 0$ for all strategies $\sigma$ in $\Sigma^\star$, whereas $\phi_\sigma \cdot \eta^{(q')} \leq 0$ for all strategies $\sigma$ in $\Sigma^{\star\prime}$. Actually, one can check an additional property by exploiting the graph structure of cartels. If $\Sigma^\star$ and $\Sigma^{\star\prime}$ are connected by an edge, i.e. if $q' = q + 1$, there is a path $\mathcal{P}$ joining $\Sigma^\star$ and $\Sigma^{\star\prime}$. In particular, a facet of $\Sigma^\star$ can be continuously mapped onto a facet of $\Sigma^{\star\prime}$ by considering the $(d-1)$-dimensional supply cones associated to consortia on $\mathcal{P}$. One can check that the generating vectors of these $(d-1)$-dimensional supply cones are actually all orthogonal to $\eta^{(q')}$. In particular, the supply sector of $\Sigma^\star$ and $\Sigma^{\star\prime}$ have parallel facets and can be separated.

## Optimal biomass yield?

In this section, we propose that microbial cartels achieve a collective optimum by yielding maximum biomass, i.e. the maximum total number of cells $N$ at fixed building-block supply

rates. At steady state, the total number of cells $N$ is related to the building-block supply rates via the overall conservation of building blocks

$$\sum_i s_i - \mu \sum_i c_i^{\text{ext}} = \frac{N\omega}{\Omega - N\omega} \times p(1-f)\delta, \qquad (A147)$$

which implies

$$N = \frac{\Omega}{\omega}\left(1 + \frac{p\delta(1-f)}{\sum_i s_i - \mu \sum_i c_i^{\text{ext}}}\right)^{-1}. \qquad (A148)$$

According to the above relation, at fixed supply rates $s_i$, maximizing biomass yield amounts to minimizing the overall building-block concentrations $\sum_i c_i^{\text{ext}}$, which implies the minimization of building-block loss via diffusion out of the volume $\Omega$. To justify that cartels achieve the maximum biomass yield, one has to show that for identical supply rates, cartels lead to lower overall building-block steady-state concentrations than consortia that are not cartels. This fact can be verified for $p = 2$ or 3 building blocks and can be justified graphically for $p = 2$. In **Appendix 7—figure 1**, we first show that increase in building-block supply is entirely dedicated to biomass growth in microbial cartels. In **Appendix 7—figure 2**, we then show that cartels are optimal in the sense that no single strategy can yield higher biomass than a cartel. More generally, although we do not have a proof of this point, we believe that cartels yield optimal biomass at fixed supply rates for an arbitrary number of building blocks $p$. In the following, we provide conditions under which the biomass optimality of cartels holds. First, we show that consortia consist of metabolic types that are subsets of cartels and that the corresponding steady-state concentrations satisfy the same weak ordering as that of the supply rates. Second, we show that the biomass yield of such consortia is smaller than that of cartels given the validity of two reasonable conjectures.

**a.**

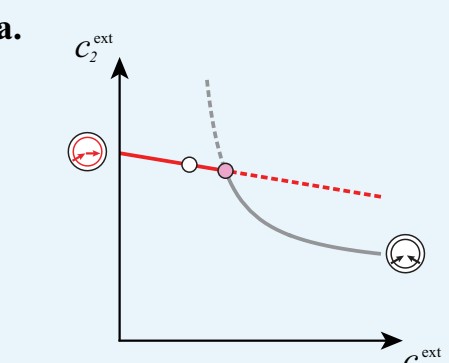

**b.**

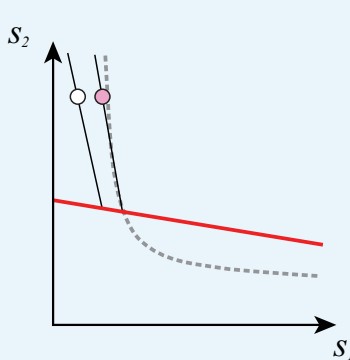

**c.**

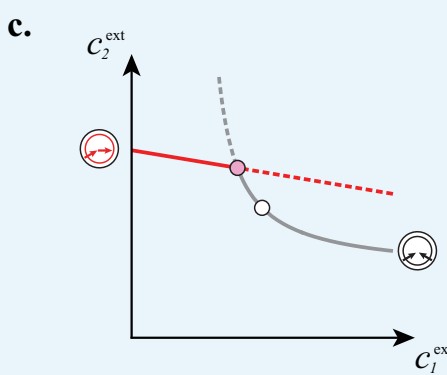

**d.**

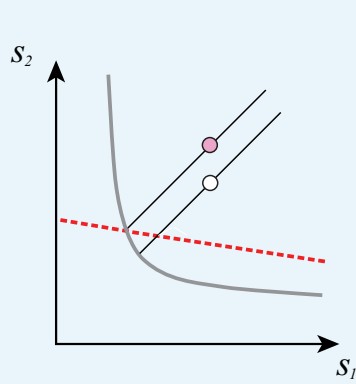

**Appendix 7—figure 1.** Cartels direct their resources toward biomass growth. **a** and **b**: For

steady-state concentrations $c_1^{\text{ext}} < c_2^{\text{ext}}$, only a converter strategy can survive (white dot in **a**). The corresponding set of supply rates $s_1$ and $s_2$ lie on a line (labelled by a white dot in **b**). Increasing the supply rate $s_1$ causes concentration $c_1^{\text{ext}}$ to increase at the expense of $c_2^{\text{ext}}$, until a pure-importer strategy can survive at $c_1^\star$ and $c_2^\star$ (pink dot in **a**). Any further increase of $s_1$ no longer affects $c_1^\star$ and $c_2^\star$ and is solely dedicated to biomass growth. **c** and **d**: For steady-state concentrations $c_1^{\text{ext}} \simeq c_2^{\text{ext}}$, only a pure-importer strategy can survive (white dot in **c**). The corresponding set of supply rates $s_1$ and $s_2$ lie on a line (labelled by a white dot in **d**). Increasing the supply rate $s_2$ causes concentration $c_2^{\text{ext}}$ to increase at the expense of $c_1^{\text{ext}}$, until a converter strategy can survive at $c_1^\star$ and $c_2^\star$ (pink dot in **c**). Any further increase of $s_2$ no longer affects $c_1^\star$ and $c_2^\star$ and is solely dedicated to biomass growth.

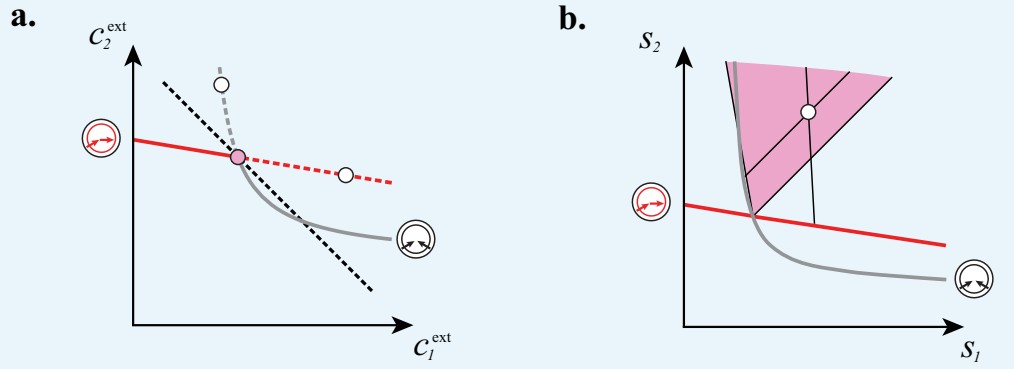

**Appendix 7—figure 2.** At fixed supply rates, microbial cartels achieve optimal biomass yield. **a** and **b**: Steady-state concentrations $c_1^{\text{ext}}$ and $c_2^{\text{ext}}$ that satisfy $c_1^{\text{ext}} + c_2^{\text{ext}} > c_1^\star + c_2^\star$ (above the dashed black line in **a** imply smaller biomass yields than achieved by the microbial cartel that exists for $c_1^\star$ and $c_2^\star$ (pink dot in **a**). The supply sector associated with the cartel defines a cone (pink region in **b**). For given supply rates in the cartel supply sector (white dot in **b**), the black lines represent the supply sectors of the pure-importer strategy and of the pure-converter strategy that are optimal when present alone (as opposed to being in a cartel). The intersection of these non-cartel supply sectors (black lines in **b**) with the steady-state curves (red and grey curves in **b**) define concentrations $c_1^{\text{ext}}$ and $c_2^{\text{ext}}$ for which $c_1^{\text{ext}} + c_2^{\text{ext}} > c_1^\star + c_2^\star$ (white dots in **a**). This result is generic for any supply rates in the cartel supply sector; thus a pure-importer or a pure-converter strategy alone leads to steady-state concentrations with smaller biomass yield than the cartel. We did not take into account the other converter strategy, belonging to the other cartel, since this cartel can only be optimal for $c_1^{\text{ext}} > c_2^{\text{ext}}$, which never happens for $s_2 > s_1$.

Consortia that are not cartels consist of optimal strategies within a metabolic class but do not necessarily contain the maximum number of metabolic classes. Consider such a consortium of optimal metabolic types for steady-state concentrations that satisfy the ordering $c_1^{\text{ext}} \geq \ldots \geq c_p^{\text{ext}}$, which is always possible via relabeling of building blocks. In Section "Microbial cartels for strict ordering" and Section "Microbial cartels for degenerate ordering", we have shown that optimal metabolic types that convert building blocks are such that the most abundant building block is converted into the least abundant building blocks. If the ordering is strict, i.e. $c_1^{\text{ext}} > \ldots > c_p^{\text{ext}}$, there are only $p$ possible optimal metabolic types $\mathcal{M}_q$, $1 \leq q \leq p$, where type $\mathcal{M}_q$ imports block $1, \ldots, q$ and converts block 1 into blocks $q+1, \ldots, p$. In particular, the metabolic types comprising a steady-state consortium are necessarily a subset of the metabolic types forming a cartel.

Moreover, for each metabolic type $\mathcal{M}_q$, the per-cell influx $\phi_{q,i}$ of block $i$ is

$$\phi_{q,i} = \begin{cases} (1-f)\delta + \sum_{j>q} \kappa_{j1}^{(q)} c_1^{(q)} & \text{if } i=1, \\ (1-f)\delta & \text{if } 1<i\le q, \\ (1-f)\delta - \kappa_{j1}^{(q)} c_1^{(q)} = \beta\left(c_j^{\text{ext}} - c_j^{(q)}\right) & \text{if } i>q. \end{cases} \tag{A149}$$

Thus, the fluxes $\phi_{q,i}$ satisfy $\phi_{q,1} > \phi_{q,2} = \ldots = \phi_{q,q} > \phi_{q,q+1} > \ldots > \phi_{q,p}$ since the internal concentrations $c_j^{(q)}$ are constant for $j>q$. In turn, the corresponding supply rates $s_i$ necessarily satisfy the same strict ordering via external flux-balance *Equation (A11)*: for $i<j$, we have

$$s_i = \mu c_i^{\text{ext}} + \sum_\sigma n_\sigma \phi_{\sigma,i} < \mu c_j^{\text{ext}} + \sum_\sigma n_\sigma \phi_{\sigma,j} = s_j, \tag{A150}$$

where $\sigma$ takes value in $\{1,\ldots,p\}$. If the ordering of external building-block concentrations is not strict, two types of equalities can occur: (i) There can be equalities between the most abundant building-block concentrations, e.g. $c_1^{\text{ext}} = \ldots = c_q^{\text{ext}}$, for which one can show that the supply rates necessarily satisfy $\min_{1\le i\le q} s_i^{\text{ext}} > s_j$ for $j>q$.(ii) There can be equalities between concentrations of building blocks that are not the most abundant, e.g. $c_i^{\text{ext}} = c_{i+1}^{\text{ext}}$, for which one can show that the supply rates necessarily satisfy $s_i = s_{i+1}$. Observe that (i) corresponds to the possible occurrence of degenerate cartels, whereas (ii) implies the absence of strategies that convert block 1 into $i+1$ while importing $i$. In any case, independent of the nature of the ordering, the metabolic types of steady-state consortia are always a subset of the metabolic types forming a cartel. Moreover, consortia of optimal metabolic types weakly preserve the ordering of building-block supplies in the sense that if $s_1 \ge \ldots \ge s_p$, we necessarily have $c_1^{\text{ext}} \ge \ldots \ge c_p^{\text{ext}}$. The order is only weakly preserved because degenerate consortia can exist for $c_1^{\text{ext}} = \ldots = c_q^{\text{ext}}$ while the supply rates are strictly ordered, e.g. $s_1 > \ldots > s_p$.

We now formulate two conjectures implying that at fixed building-block supply cartels have a higher biomass yield than consortia that are not cartels.

i.  Our first conjecture, which is the strongest, concerns the structure of the supply space. To be more specific, consider supply rates $s_i$ that lie within the supply sector of a cartel. Consortia that are not cartels are necessarily made of a subset of the metabolic classes defining that cartel. Our conjecture is to posit that, for supply rates within a cartel-specific supply sector, a consortium can only exist for steady-state concentrations $c_1^{\text{ext}}, \ldots, c_p^{\text{ext}}$ that are such that

$$G_{\mathcal{M}}(c_1^{\text{ext}}, \ldots, c_p^{\text{ext}}) = \sup_{\sigma \in \mathcal{M}} g_\sigma(c_1^{\text{ext}}, \ldots, c_p^{\text{ext}}) > \delta, \tag{A151}$$

for metabolic classes $\mathcal{M}$ that belong to the cartel but are not in the consortium. This property means that if the supply rates $s_i$ are such that a cartel emerges, consortia that are not cartels can be invaded by each missing metabolic class. In particular, this property implies that the emergence of a cartel does not depend on the history of which metabolic types are introduced first. One can confirm this property concretely for harmonic-mean model when $p=3$.

ii. Our second conjecture is about the convexity of the level sets of $G_{\mathcal{M}}$, the optimal growth rate for each metabolic type $\mathcal{M}$. One can show that for any particular strategy $\sigma$, the level sets of the growth function $g_\sigma$ inherit the convexity of the level set of the universal rate function $g$ governing biomass production. We further assume that this convexity property is inherited by the optimal growth function $G_{\mathcal{M}}$, once optimized within a given metabolic class. For $p=2$ or $3$, one can confirm that the sets of external concentrations $c_1^{\text{ext}}, \ldots, c_p^{\text{ext}}$ such that $G_{\mathcal{M}}(c_1^{\text{ext}}, \ldots, c_p^{\text{ext}}) \le \delta$ are indeed convex for the $\min$ model. From a biological standpoint, our conjecture means that within a metabolic class, although the distribution of enzymes can be tuned to external building-block availabilities, microbes still experience diminishing marginal utility. In other words, increasing building block availability yields decreasing benefit within a metabolic class and beating diminishing marginal utility requires a switch of metabolic classes.

If true, the two conjectures above imply that cartels achieve optimal biomass. To see this, consider for instance a cartel associated to a strict ordering of external concentrations, i.e. $c_1^\star > \ldots > c_p^\star$. The steady-state concentrations $c_1^\star > \ldots > c_p^\star$ are the solution of the equations

$$G_q(c_1^{\text{ext}}, \ldots, c_p^{\text{ext}}) = \delta, \qquad (A152)$$

where $G_q$ denotes the optimal growth function for metabolic class $\mathcal{M}_q$, $1 \leq q \leq p$. Our first conjecture implies that a consortium that is not a cartel exists at steady-state concentrations for which $G_q(c_1^{\text{ext}}, \ldots, c_p^{\text{ext}}) = \delta$ if metabolic class $\mathcal{M}_q$ is present and for which another metabolic class, say $\mathcal{M}_{q'}$, can invade the consortia, i.e. $G_{q'}(c_1^{\text{ext}}, \ldots, c_p^{\text{ext}}) > \delta$. Thus these steady-state concentrations belong to the nonlinear cone defined by $G_q(c_1^{\text{ext}}, \ldots, c_p^{\text{ext}}) \geq \delta$, $1 \leq q \leq p$. Showing that cartels achieve optimal biomass is then equivalent to showing that this nonlinear cone lies within the half-space $H$ defined by

$$\sum_i c_i^{\text{ext}} \geq \sum_i c_i^\star. \qquad (A153)$$

Our second conjecture about convexity implies that the nonlinear cone is contained in the linear cone originating from $c_1^\star, \ldots, c_p^\star$ and approximating the nonlinear cone defined by $G_q(c_1^{\text{ext}}, \ldots, c_p^{\text{ext}}) \geq \delta$, $1 \leq q \leq p$. Let us denote the supporting vectors generating this cone by $V_q$, $1 \leq q \leq p$, where $V_q$ is a tangent to the path defined by $G_{q' \neq q}(c_1^{\text{ext}}, \ldots, c_p^{\text{ext}}) = \delta$, $1 \leq q \leq p$ at $c_1^\star, \ldots, c_p^\star$, oriented in the direction of increasing $G_q$. One possible choice for the $V_q$ is via the relation

$$\begin{bmatrix} V_1 \\ \vdots \\ V_p \end{bmatrix} = \begin{bmatrix} \nabla G_1 |_{c_1^\star, \ldots, c_p^\star} \\ \vdots \\ \nabla G_p |_{c_1^\star, \ldots, c_p^\star} \end{bmatrix}^{-1}, \qquad (A154)$$

which ensures that $V_q$ is tangent to each hypersurface $G_{q' \neq q} = \delta$ and correctly oriented.

We now show that the linear cone generated by $V_q$ lies within the half-space $H$ by showing that there are positive numbers $\eta_q$ such that

$$V_q \cdot \mathbf{1} = \sum_i V_{q,i} = \eta_q \geq 0, \qquad (A155)$$

which can be written in matrix form as

$$\begin{bmatrix} V_1^T \\ \vdots \\ V_p^T \end{bmatrix} 1 = \begin{bmatrix} \eta_1 \\ \vdots \\ \eta_p \end{bmatrix}. \qquad (A156)$$

The key point is to realize that for both the minimum and harmonic-mean growth function, the optimal growth functions $G_q$ are homogeneous of degree one. Because of this property, at cartel specific external concentrations $c_1^\star, \ldots, c_p^\star$, we have

$$\nabla G_q |_{c_1^\star, \ldots, c_p^\star}^T \cdot c^\star = G_q(c_1^\star, \ldots, c_p^\star) = \delta, \qquad (A157)$$

which can be written in matrix form as

$$\begin{bmatrix} \nabla G_1|_{c_1^\star,\ldots,c_p^\star}^T \\ \vdots \\ \nabla G_p|_{c_1^\star,\ldots,c_p^\star}^T \end{bmatrix} \begin{bmatrix} c_1^\star \\ \vdots \\ c_p^\star \end{bmatrix} = \delta 1 \,, \tag{A158}$$

and directly shows that taking $\eta_q = c_q^\star/\delta$ satisfies relation **Equation (A156)**.

