## [Decision Letter]

Editors’ note: a previous version of this study was rejected after peer review, but the authors submitted for reconsideration. The first decision letter after peer review is shown below.]

Thank you for submitting your work entitled "Bacterial cartels at steady supply" for consideration by *eLife*. Your article has been reviewed by two peer reviewers, and the evaluation has been overseen by a Reviewing Editor and Arup Chakraborty as the Senior Editor. The following individual involved in review of your submission has agreed to reveal his identity: Eric D Kelsic (Reviewer #1).

Our decision has been reached after consultation between the reviewers. As you will see, one of the reviewers raised a number of significant concerns, many of which the other reviewer acknowledged to be important during subsequent discussion. Based on these discussions and the individual reviews below, we regret to inform you that your work will not be considered further for publication in *eLife*.

*Reviewer #1:*

The manuscript by Taillefumier et al. is well written and comprehensive, and I found the theory of bacterial cartels to be quite elegant and thought-provoking. The authors set out to understand microbial diversity by modeling communities of organisms that require a number of essential building blocks for growth. If the building blocks are present in the environment they may be imported into cells, or cells may choose to obtain the building block by enzymatically converting from one type to another. The authors model the growth potential of different strategies under the constraint of a fixed enzyme budget. Through a number of simplifying assumptions and linearizations the authors identify optimal metabolic classes and finally show that sets of organisms, termed "cartels", can optimally consume resources and prevent other species from invading. While such communities only coexist at fixed points within the space of external resource concentrations, the authors beautifully show that these points are attractors for regions within the space of supply rates, and thus cartels will frequently emerge when communities are continually challenged with organisms exhibiting various metabolic strategies.

The cartel theory makes significant progress toward understanding the diversity of microbial communities. Importantly, the work shifts the emphasis away from the number of (rather abstract) environmental resources and focuses instead on the building blocks that are essential for growth and the various strategies by which to obtain these building blocks. I find this setup more conceptually appealing. I believe that the manuscript will be of broad general interest and it would be great to see it published in *eLife*.

I have no major concerns. The manuscript is already highly polished and could be published without significant revisions.

*Reviewer #2:*

The manuscript by Wingreen and co-workers examines the structure of bacterial communities that consist of different metabolic types, under a non-trivial model that involves feedbacks from external metabolite concentrations and internal metabolic conversions and fluxes, subject to specific enzyme availability, onto the growth rates. Overall, this is an important subject for which general theoretical results could be useful for guiding experiments as well as introducing new ideas for analysis of bacterial communities.

Unfortunately, the text does an *extremely* poor job of explaining the main conclusions. One can follow the main assumptions of the model, but once the notions of consortia and cartels are introduced, it requires a good deal of work to follow which features of the model have been preserved and which ones have been conveniently thrown out the window under some new assumption that enables analysis. Over the course of the manuscript, several such assumptions are made, which effectively change the underlying model, while the key notions of consortia and cartels are defined non-rigorously and in a model-dependent manner.

The section entitled Model is the clearest section of the paper. The authors make many modeling assumptions here, some of which are questionable (see below), but these are all explained clearly.

The Results section begins with Numerical Simulations, which motivates the rest of the manuscript. The authors give too few details on how extensive their simulations are. How reproducible are they? How long have they run them? How complex a space of strategies was explored?

A) A key point at this stage of the manuscript (i.e. Results) and at the end of the Model Section is the statement that the authors are interested only in the structure of the equilibrium attained, not in the dynamics. They state: "Eventually, at long times, the surviving population will consist entirely of optimal cell types and will no longer change. It is this final population that concerns us; we only simulate metabolic competition to gain insight into the final optimized population, which is independent of the specific dynamics of the simulation."

This statement appears to be some kind of hope since it is neither (a) proven nor (b) demonstrated by extensive numerical simulations. The simulations assume that new types are only added once other types have gone extinct (subsection “Competitive population dynamics”, last paragraph), which is a pretty big assumption regarding "specific dynamics of the simulation" and which, in fact, guarantees that once a community consists of types that are not going extinct it won't have to worry about new types coming in. Indeed, the structures that are later called "cartels" in this manuscript are automatically stable under the given dynamics, due to the fact that avoidance of extinction necessarily excludes competition by new types; such dynamics are intrinsically 'cartel-friendly'. Under different dynamics, for example where new types appear at random times, or are formed by mutation from existing types, there would be no such guarantee, and it is therefore not at all clear that the final population structure (if such a stable structure even exists; see B) would not depend on the specific dynamics.

In fact, Figure 2 seems to contradict the basic assertion that a surviving population consists only of optimal types and no longer changes. In Figure 2, blue and yellow curves seem to exchange at regular intervals, and the population composition is dynamic in time. In this case, the caption states that the "external building-block concentrations fluctuate due to the invasion by and extinction of metabolic variants." Which leads to my next major point.

B) The entire analytical treatment of the manuscript is predicated on the assumption that the external concentrations Ciext are fixed. From the second paragraph of the subsection “Optimal metabolic classes” to the end of the first paragraph of the subsection “Structure of consortia”, the authors develop the basic framework in which external concentrations are self-consistently derived such that they obtain an optimal metabolic strategy under the assumption that Ciext do not fluctuate. However, they have already shown in Figure 2 that even in the simplest cases the dynamics may converge to a fluctuating solution under even their cartel-friendly dynamics (see A above).

Thus, at this point in the text, the authors have subtly shifted our attention away from the dynamics they initially presented, to some new scenario, in which they search for a certain type of optimum (i.e. one with fixed Ciext). They carry out a careful analysis of such optima (with some additional caveats) throughout the rest of the text. However, they provide no information as to whether such optima are dynamically stable at all. But if these optima are not dynamically stable, then what use is all of their analysis?

Overall, it appears that the authors have performed a nice characterization of a well-defined graph-theoretic optimization problem (stated in the aforementioned subsections). They motivated this problem in the context of a dynamical system, but they have failed to demonstrate that the dynamics converge to their optimum; or more precisely, to give a characterization of conditions (i.e. the choice of constant supply rates, initial conditions, and the exact simulation dynamics) under which their optimum is achieved. Therefore, I recommend that they submit this paper elsewhere, e.g. to a graph theory or combinatorial optimization journal, where their more rigorous results will be properly assessed by mathematicians. Biologists who read *eLife* will not understand the range of applicability, or lack thereof, of these results. I certainly do not.

Other concerns:

C) The assumption to study the "minimum model" in the last paragraph of the subsection “Structure of consortia” is quite strong. For example, the growth rate no longer depends on the *b_i_* variables that were introduced initially. This is another example of the authors presenting a much more general model than the one they ultimately analyze. For the poor reader who has made it through the Model section, but is not technically proficient to follow the derivations where the extra assumptions are buried (such as in the last paragraph of the subsection “Structure of consortia”), this can end up being misleading.

D) On the technical level, the question of the *b_i_* is important, since different bacteria may have different composition, at least with regard to a subset of key building blocks, and one would like to know if this impacts the metabolic specialization across the community to any extent. Another place where this seems to be relevant is in [Disp-formula equ3], where the term -*gb_i_* at the end of the equation corresponds to the assumption, stated earlier, that building blocks are consumed at a rate proportional to their concentration in biomass, i.e. *b_i_*. This is necessary for the steady-state growth, and will be achieved automatically at the level of the bulk population, but at the level of enzymatic activity inside cells, one expects something like -*gc_i_*. This point should be discussed, and the relevant extra assumptions that are being made here need to be pointed out.

E) In the Discussion, the authors state "In particular, cartels maintain fixed external resource concentrations by adjusting their populations to compensate for changes in supply." This is a question of dynamics, again, but dynamics have not been addressed in the analysis. Critically, the statement here is that if the external resource changes (over some unspecified timescale), the subpopulation sizes that contribute to the cartel will adjust such that the external resource concentrations will be maintained. Since the change of subpopulation size is what appears to be controlling the resource concentrations in such cases, the authors appear to be violating their own "separation of timescales" assumption which they made: "since the lifetime of a cell is much larger than the timescales associated with metabolic processes… separation of timescales justifies steady-state approximation for the fast variables: (ddt)ciσ=0∧(ddt)ciext=0

F) Consortia and Cartels are never rigorously defined before they are used. The terms are typically used first and defined later. For example, Cartels are actually defined only in Figure 4 caption, and later in the Discussion. The term consortia is used in the first sentence of the subsection “Structure of consortia” in a loose sense, i.e. interchangeably with "community composition", but that is inconsistent with the use of the term in the definition of cartels as "consortia with at least p distinct metabolic classes" (Caption, Figure 4), since it lacks the important notion of optimal types, which appears in the definition in the Discussion (first paragraph).

Even after reading these sections multiple times, it seems that the authors are missing something in their verbal definitions. There is no mention of "co-optimality" in their final definition: "Cartels are defined as consortia of at least as many distinct optimal cell types – each with a fixed metabolic strategy – as there are shared resources". The definition really should include the fact that all of these types contribute to the same optimum.

It is striking that the authors neglected to give a rigorous mathematical definition of either Cartel or Consortia, for example in terms of a big set C = {σ: condition is true}.

Additional points:

G) The derivation of the growth rate function, [Disp-formula equ1], assumes an assembly line for biomass in which reactions do not occur in parallel, at least for the most part. This might be fine for building up a protein in a sequential manner from amino-acids, but many other reactions that build biomass involve building blocks that can themselves be formed from other constituents along different *parallel* pathways.

H) Interconversion reactions, represented by Κ*_ij_*, could in reality involve multiple reactants, for example any metabolic reaction of the form A + B <-> C, and these would introduce non-linearities that are not captured by the equations.

I) Taking the death rate to be independent of the phenotype and the external conditions seems to be a very strong assumption [Disp-formula equ6].

J) One could not tell in the last paragraph of the subsection “Competitive population dynamics” whether the simulation captures finite population size effects or not, i.e. demographic fluctuations of size 1 / N.

K) The numerical simulations do not involve mutation, in that new types are introduced randomly rather than as mutated version of existing types. One expects that mutational dynamics could have different stability criteria. Also, if a random type is introduced, presumably it arrives from outside the community, and therefore why should it be introduced in small numbers? Is that a reasonable assumption for bacteria? It doesn't seem intuitive.

L) What happens to a cartel if one member is removed? Does it matter which member is removed?

M) The word cartel has strong negative associations (i.e. drug cartels), and even at a more basic level the use of this word in the manuscript is fundamentally inaccurate, since the economic definition of a cartel is based on an agreement to control prices, instead of an optimization principle. I am not sure why the authors want to go in this direction.

[Editors’ note: what now follows is the decision letter after the authors submitted for further consideration.]

Thank you for resubmitting your work entitled "Microbial consortia at steady supply" for further consideration at *eLife*. Your revised article has been favorably evaluated by Arup Chakraborty as the Senior Editor, Carl Bergstrom as the Reviewing Editor (Carl Bergstrom) and two reviewers.

The manuscript has been greatly improved and both referees enthusiastically recommend publication.

*Reviewer #1:*

In "Microbial consortia at steady supply", the authors consider a population model in which microorganisms exposed to a flux of metabolites follow different strategies concerning which metabolites to import and which to produce through conversion. They show that populations evolve to form cartels, which prevent invasion from other strategies and keep external resources fixed. Overall, I think the authors do a good and extensive job of demonstrating their results. I recommend this paper for publication.

*Reviewer #2:*

Taillefumier et al. present an elegant and compelling theory of microbial coexistence driven by diverse strategies for import and synthesis of essential metabolic building blocks. I found the revised manuscript to be improved over the original submission, making several changes that facilitated communication of the main results. In particular, the elaborated explanation of the dynamics of the simulation better clarifies that communities are converging toward optimal consortia (rather than being mistaken for oscillatory dynamics). Additionally, I found it reassuring that the new results using gradient-based optimization of communities arrived at similar results as the simulations that were continuously challenged with randomly sampled species types. This resolves one of my main concerns from the original manuscript, and shows that the authors' theory of optimal cartels captures essential aspects of the simulations despite extensive simplifications. Finally, the earlier definition of cartels in the Introduction aided understanding, and multiple revisions in the Discussion helped for putting the theory in context.

This is a high-quality work in theoretical ecology that makes significant progress toward the major problem of understanding microbial diversity, and I support its publication in *eLife* where it can reach a broad biological audience.

---

## [Author Response]

[Editors’ note: the author responses to the first round of peer review follow.]

*Reviewer #2:*

*The manuscript by Wingreen and co-workers examines the structure of bacterial communities that consist of different metabolic types, under a non-trivial model that involves feedbacks from external metabolite concentrations and internal metabolic conversions and fluxes, subject to specific enzyme availability, onto the growth rates. Overall, this is an important subject for which general theoretical results could be useful for guiding experiments as well as introducing new ideas for analysis of bacterial communities.*

*Unfortunately, the text does an extremely poor job of explaining the main conclusions. One can follow the main assumptions of the model, but once the notions of consortia and cartels are introduced, it requires a good deal of work to follow which features of the model have been preserved and which ones have been conveniently thrown out the window under some new assumption that enables analysis. Over the course of the manuscript, several such assumptions are made, which effectively change the underlying model, while the key notions of consortia and cartels are defined non-rigorously and in a model-dependent manner.*

*The section entitled Model is the clearest section of the paper. The authors make many modeling assumptions here, some of which are questionable (see below), but these are all explained clearly.*

*The Results section begins with Numerical Simulations, which motivates the rest of the manuscript. The authors give too few details on how extensive their simulations are. How reproducible are they? How long have they run them? How complex a space of strategies was explored?*

We regret the lack of clarity in our presentation. As we understand it, at the root of virtually all the reviewer’s objections is a concern that metabolic competition between different metabolic types may not lead to stable stationary states. This concern is the consequence of a miscommunication about the methodology and results of our simulations of competitive bacterial dynamics. This miscommunication led the reviewer to think that the population dynamics of metabolic types can exhibit “oscillations”. However, population oscillations never happen in our simulations: there are only new strategies sweeping existing ones. We hope to dispel the concern about the stability of the dynamics by offering detailed responses to the reviewer’s objections (A) and (B). We also have substantially revised our manuscript for greater clarity and to avoid any other unfortunate miscommunications. In particular, we added two new figures and revised the original figure at the root of the confusion.

*A) A key point at this stage of the manuscript (i.e. Results) and at the end of the Model Section is the statement that the authors are interested only in the structure of the equilibrium attained, not in the dynamics. They state: "Eventually, at long times, the surviving population will consist entirely of optimal cell types and will no longer change. It is this final population that concerns us; we only simulate metabolic competition to gain insight into the final optimized population, which is independent of the specific dynamics of the simulation."*

*This statement appears to be some kind of hope since it is neither (a) proven nor (b) demonstrated by extensive numerical simulations. The simulations assume that new types are only added once other types have gone extinct (subsection “Competitive population dynamics”, last paragraph), which is a pretty big assumption regarding "specific dynamics of the simulation" and which, in fact, guarantees that once a community consists of types that are not going extinct it won't have to worry about new types coming in. Indeed, the structures that are later called "cartels" in this manuscript are automatically stable under the given dynamics, due to the fact that avoidance of extinction necessarily excludes competition by new types; such dynamics are intrinsically 'cartel-friendly'. Under different dynamics, for example where new types appear at random times, or are formed by mutation from existing types, there would be no such guarantee, and it is therefore not at all clear that the final population structure (if such a stable structure even exists; see B) would not depend on the specific dynamics.*

*In fact, Figure 2 seems to contradict the basic assertion that a surviving population consists only of optimal types and no longer changes. In Figure 2, blue and yellow curves seem to exchange at regular intervals, and the population composition is dynamic in time. In this case, the caption states that the "external building-block concentrations fluctuate due to the invasion by and extinction of metabolic variants." Which leads to my next major point.*

Although we did not include an extensive account of our simulations in the original submission, we did perform exhaustive simulations, and indeed had thoroughly tested all these important points raised by the reviewer. In the revised manuscript, we show that the simulations of metabolic competition between microbes consistently lead to stationary states populated by the same metabolic types. In Figure 1 of the revised manuscript, we present 3 independent simulations of metabolic competition for supply condition s_1_ = 11, s_2_ = 9 > s_3_ = 0, starting from 24 different initial conditions for each run (i.e. different metabolic strategies). The volume of the colony Ω is chosen so that the carrying capacity is ≈ V/v = 100, 000 cells. Each simulation is run for 10^8^ time steps, which represents on the order of 10^5^ generations. In each case, the population of bacteria quickly reaches carrying capacity, at which point the availability of building blocks plumets to low values, and metabolic competition begins in earnest. At each time step, we simulate 24 different strategies. The simulation of Figure 1 together with the vastly expanded “Numerical results” section, clarify four points which we believe are the source of reviewer’s concerns:

1) The external building-block concentrations converge toward cartel-specific values.

In Figure 1, the bottom plots of all panels show that for all runs, the external building-block concentrations tend to the same stationary values with the two externally provided building blocks stabilizing at numerically identical values and the other building block stabilizing at lower concentration. Importantly, we observe that when a new strategy replaces another, i.e. when two curves cross each other in the top plots of each panel, the building-block concentrations are only marginally changed, gradually approximating the theoretically optimal cartel-specific values. The visible building-block fluctuations appearing in the original submission were largely due to the adoption of a low carrying capacity and, unfortunately, to poor editing in Adobe Illustrator. We apologize for the confusion.

2) Competition between metabolic types leads to stable selection of metabolic types.

After metabolic competition begins in earnest, building-block concentrations approach their theoretically optimal values and virtually all newly introduced variants quickly be- come extinct. Only metabolic strategies that improve on the already present, nearly optimal strategies can persist. When a newly introduced type does successfully invade the population, it always belongs to an optimal class and replaces the existing population of that same class. For instance, in the top panel of Figure 1, the most abundant type (purple and pink curves) always has a pure-importer strategy. Importantly, the color of a curve is not attached to a specific strategy but rather to its class: purple and pink indicate the two pure-importer strategies out of the 24 strategies. In particular, the population of a strategy never “oscillates”: when a curve goes down (e.g. the pink curve), the associated strategy goes extinct due to invasion by a new strategy belonging to the same class (purple), which in turn can be invaded by yet another strategy of the same class (pink again). We observe that the overall population of an optimal class, e.g. the sum of the pink and purple curves in the top plot of each panel, is nearly constant, a clear indication of a tendency toward a stationary population of each optimal type. We apologize for the lack of a more detailed explanation of the original simulation figures.

3) Existing types are constantly competing with newly introduced variants.

While three types may coexist for extended periods according to the competitive exclusion principle, the 21 other types have populations nσ that decay exponentially until extinction, i.e. until *n_σ_* < 0.9999. Upon extinction, a new type is introduced at *n_σ_* = 1. The overwhelming majority of these new types do not invade the already present types and quickly become extinct, leading to the introduction of new variants. As a result of this scheme, new metabolic strategies are constantly introduced, representing a total of more than 10^9^ different introduced strategies during each run.

4) The space of metabolic strategies is extensively sampled.

In a first set of simulations we performed, new strategies were uniformly sampled from the simplex Ʃ α + Ʃ κ = *E*. This sampling scheme revealed that competition consistently selected for types for which certain enzymes were negligible, approximating the theoretically predicted metabolic types (belonging to “metabolic classes” for which the abundance of certain enzymes is zero). Therefore, in subsequent simulations, including the ones presented here, new strategies were sampled according to a new scheme: four strategies were still uniformly sampled from the simplex Ʃ α + Ʃ κ = *E*, but two strategies were sampled from each of the 10 possible optimal metabolic classes. Whenever a strategy goes extinct, we replace it with a strategy sampled from the same metabolic class. We use this scheme because drawing a type from an optimal class has probability zero when one uses the naive uniform sampling of the simplex Ʃ α + Ʃ κ = *E*. The observed persistence of the expected optimal types confirmed our theoretical analysis.

We hope that the above points will dispel the reviewer’s concerns about our methodology used to simulate competitive bacterial dynamics.

*B) The entire analytical treatment of the manuscript is predicated on the assumption that the external concentrations Ciext are fixed. From –the second paragraph of the subsection “Optimal metabolic classes” to the end of the first paragraph of the subsection “Structure of consortia”, the authors develop the basic framework in which external concentrations are self-consistently derived such that they obtain an optimal metabolic strategy under the assumption that Ciext do not fluctuate. However, they have already shown in Figure 2 that even in the simplest cases the dynamics may converge to a fluctuating solution under even their cartel-friendly dynamics (see A above).*

*Thus, at this point in the text, the authors have subtly shifted our attention away from the dynamics they initially presented, to some new scenario, in which they search for a certain type of optimum (i.e. one with fixed Ciext). They carry out a careful analysis of such optima (with some additional caveats) throughout the rest of the text. However, they provide no information as to whether such optima are dynamically stable at all. But if these optima are not dynamically stable, then what use is all of their analysis?*

*Overall, it appears that the authors have performed a nice characterization of a well-defined graph-theoretic optimization problem (stated –in the aforementioned subsections). They motivated this problem in the context of a dynamical system, but they have failed to demonstrate that the dynamics converge to their optimum; or more precisely, to give a characterization of conditions (i.e. the choice of constant supply rates, initial conditions, and the exact simulation dynamics) under which their optimum is achieved. Therefore, I recommend that they submit this paper elsewhere, e.g. to a graph theory or combinatorial optimization journal, where their more rigorous results will be properly assessed by mathematicians. Biologists who read eLife will not understand the range of applicability, or lack thereof, of these results. I certainly do not.*

We addressed the reviewer’s concern about the stability of the population dynamics in response to point (A). We believe that the reviewer’s concern was due to a miscommunication and apologize for the confusion. In all of our extensive simulations, competitive dynamics converge to a final state where both the resource external concentrations and the cell-type populations are constant. To further address the reviewer’s concern, we numerically confirmed that the final cell types obtained via competitive dynamics achieve the optimal cell growth given resource availabilities. As explained in the expanded section “Numerical results”, we designed an iterative optimization algorithm that yields steady-state optimal cell types with high accuracy. The thus-obtained cell types are guaranteed to achieve optimal growth for the given external resource concentrations by design of the algorithm and are virtually identical to the final cell types obtained via competitive population dynamics. We illustrate the convergence of surviving cell types toward the optimal types in Figure 2 of the revised manuscript. Specifically, we define the ratio of the growth rate of surviving cell types to the growth rate of the optimal cell types as the “relative fitness” of a cell type. Tracking the relative fitness of surviving cells shows that each successful invasion and each ensuing displacement event lead to an increase of the relative fitness of surviving cells, and this quantity converges to one on large timescales. These additional studies confirm that the competitive dynamics eventually converges to a non-invadable steady state, motivating our further analytical study of the final state of the community.

*Other concerns:*

*C) The assumption to study the "minimum model" in the last paragraph of the subsection 2Structure of consortia” is quite strong. For example, the growth rate no longer depends on the b_i_ variables that were introduced initially. This is another example of the authors presenting a much more general model than the one they ultimately analyze. For the poor reader who has made it through the Model section, but is not technically proficient to follow the derivations where the extra assumptions are buried (such as in the last paragraph of the subsection 2Structure of consortia”), this can end up being misleading.*

The main conclusions of our paper concern the generic emergence of cartels and the benefit that division of labor provides to cartels’ members. This conclusion is independent of the growth model considered or the values of the *b_i_*. While we presented the simple case of *b_i_*= 1 in the main text, the optimization has been carried out for arbitrary *b_i_* in the Supporting Information. Changing the values of *b_i_* shifts the hypersurfaces of steady-state external concentrations, but cartels always emerge and are always made of the optimal types identified in our analysis. We show in Supporting Information that the results derived for the minimum model rigorously hold for any growth functions with reasonable properties, i.e. functions that are increasing with internal resources and that are limited by the relatively least abundant resource. Thus, our main conclusions are not dependent on the details of the model.

*D) On the technical level, the question of the b_i_ is important, since different bacteria may have different composition, at least with regard to a subset of key building blocks, and one would like to know if this impacts the metabolic specialization across the community to any extent. Another place where this seems to be relevant is in [Disp-formula equ3], where the term -gb_i_ at the end of the equation corresponds to the assumption, stated earlier, that building blocks are consumed at a rate proportional to their concentration in biomass, i.e. b_i_. This is necessary for the steady-state growth, and will be achieved automatically at the level of the bulk population, but at the level of enzymatic activity inside cells, one expects something like -gc_i_. This point should be discussed, and the relevant extra assumptions that are being made here need to be pointed out.*

As the reviewer suggests, we have considered dilution terms *−gc_i_
*in our model. In Supporting Information, we justify why these terms are negligible compared with the production/consumption fluxes within the cells. Moreover, we check that neglecting such dilution terms is compatible with overall building-block conservation.

*E) In the Discussion, the authors state "In particular, cartels maintain fixed external resource concentrations by adjusting their populations to compensate for changes in supply." This is a question of dynamics, again, but dynamics have not been addressed in the analysis. Critically, the statement here is that if the external resource changes (over some unspecified timescale), the subpopulation sizes that contribute to the cartel will adjust such that the external resource concentrations will be maintained. Since the change of subpopulation size is what appears to be controlling the resource concentrations in such cases, the authors appear to be violating their own "separation of timescales" assumption which they made: "since the lifetime of a cell is much larger than the timescales associated with metabolic processes… separation of timescales justifies steady-state approximation for the fast variables:*
(ddt)Ciσ=0∧(ddt)ciext=0

In microbial communities, metabolic processes generally have a much faster kinetic rate than the rate of cellular growth (≍δ). As a result, metabolic fluxes quickly equilibrate, which means that internal and external building-block concentrations are stationary on the slow timescale 1/δ, i.e. (ddt)ciσ and(ddt)ciext=0. This separation of timescales remains valid in the presence of external perturbations, e.g. a change in the rates of supply, as long as these perturbations occur on a slow timescale (≥ 1/δ). Moreover, if the supply rates change slowly compared with the rate of cellular growth (» 1/δ), there is another separation of timescales and cell-type populations nσ quickly reach their steady state, i.e. (ddt)nσ=0 over the timescale of changes in the supply rates. Thus, for different supply rates, the competitive steady state of the microbial community is obtained by solving (ddt)ciσ=0, (ddt)ciext=0 ∧(ddt)nσ=0, while requiring the optimality of the present cell types. It turns out that, as long as the supply rates belong to a cartel supply sector, the competitive steady state is established by cartel cell types at exactly the same external building-block concentrations, but for different cell type populations. For the sake of clarity, we specify that compensation via population dynamics occurs for slow changes of the rate of supply in the revised manuscript. We also discuss explicitly the effect of change in the supply rates occurring on different timescales in the last paragraph of the Discussion.

*F) Consortia and Cartels are never rigorously defined before they are used. The terms are typically used first and defined later. For example, Cartels are actually defined only in Figure 4 caption, and later in the Discussion. The term consortia is used in the first sentence of the subsection “Structure of consortia” in a loose sense, i.e. interchangeably with "community composition", but that is inconsistent with the use of the term in the definition of cartels as "consortia with at least p distinct metabolic classes" (Caption, Figure 4), since it lacks the important notion of optimal types, which appears in the definition in the Discussion (first paragraph).*

*Even after reading these sections multiple times, it seems that the authors are missing something in their verbal definitions. There is no mention of "co-optimality" in their final definition: "Cartels are defined as consortia of at least as many distinct optimal cell types – each with a fixed metabolic strategy – as there are shared resources". The definition really should include the fact that all of these types contribute to the same optimum.*

*It is striking that the authors neglected to give a rigorous mathematical definition of either Cartel or Consortia, for example in terms of a big set C = {σ: condition is true}.*

We apologize for the lack of clarity in the definition of “consortia” and “cartels”. We have added a paragraph at the end of the Model section where “consortia” and “cartels” are both clearly defined and where the co-optimality of the final cell types is mentioned.

*Additional points:*

*G) The derivation of the growth rate function, [Disp-formula equ1], assumes an assembly line for biomass in which reactions do not occur in parallel, at least for the most part. This might be fine for building up a protein in a sequential manner from amino-acids, but many other reactions that build biomass involve building blocks that can themselves be formed from other constituents along different parallel pathways.*

We agree that other biomass constituents such as RNA can be formed along different parallel pathways. However, proteins represents ∼ 60 − 70% of the dry biomass of prokaryotes (Simon and Azam, 1989) and ∼ 50% of the dry biomass of eukaryotes (Feij´o Delgado et al., 2013). Moreover, metabolic pathways are universally mediated by protein enzymes. Accordingly, our model primarily equates the rate of biomass production with the rate of protein biosynthesis.

*H) Interconversion reactions, represented by Κ_ij_, could in reality involve multiple reactants, for example any metabolic reaction of the form A + B <-> C, and these would introduce non-linearities that are not captured by the equations.*

We agree that considering reactions involving multiple reactants is an important question. In particular, a central issue to address is whether the nonlinearities of the corresponding reaction fluxes can impact the stability of the population dynamics. Unfortunately, studying the general effect of such nonlinearities is a very difficult problem which is beyond the scope of the present paper.

*I) Taking the death rate to be independent of the phenotype and the external conditions seems to be a very strong assumption [Disp-formula equ6].*

In our study, we assume the death rate δ to be a “failure rate” that is independent of the phenotype or the external conditions. Such an assumption would of course be very accurate in a chemostat environment, where cells are lost at a fixed dilution rate. However, we argue more generally that taking δ as an independent constant is not necessarily a strong assumption (it is actually a standard assumption within the field). Changing the values of δ for each cell type only shifts the hypersurface of steady-state concentrations Gm (cp,ext,…, cpext)=δ While the locus of their intersection may change, it is generically true that in p-dimension, p such hypesurfaces intersect. Therefore, cartels still emerge for distinct species-dependent death rates. Moreover, we model the influence of external conditions on cell population dynamics via a concentration-dependent biomass production rate gσ(c1ext,…, cpext), so that the net growth rate gσ(c1ext,…, cpext)−δ depends on resource availabilities. Thus, considering concentration-dependent death rates δ(cpext,...,cpext) does not represent a conceptual shift from our current model.

*J) One could not tell in the last paragraph of the subsection “Competitive population dynamics” whether the simulation captures finite population size effects or not, i.e. demographic fluctuations of size 1 / N.*

Our simulations do not capture finite population size effects. Our analysis is primarily concerned with the emergence of division of labor in metabolically competing species. We are considering resource supplies for which relatively large population of cells emerge.

*K) The numerical simulations do not involve mutation, in that new types are introduced randomly rather than as mutated version of existing types. One expects that mutational dynamics could have different stability criteria. Also, if a random type is introduced, presumably it arrives from outside the community, and therefore why should it be introduced in small numbers? Is that a reasonable assumption for bacteria? It doesn't seem intuitive.*

Considering mutated versions rather than random variants does not change the results of our analysis as long as the set of possible cell types remains the same. Extensive simulations (as well as analytical considerations of the case of 2 and 3 building blocks) show that mutational population dynamics are also unconditionally stable.

Introducing random variants or mutated versions in larger numbers does not change the results of our analysis either. When a new strategy is introduced in larger numbers, the change in consumption and leakage by the population may temporarily perturb the external concentrations of building blocks. However, the ensuing population dynamics ultimately converges to the same population steady state as when a new strategy is introduced in small numbers.

*L) What happens to a cartel if one member is removed? Does it matter which member is removed?*

The complete removal of a member of a cartel clearly modifies the steady state of the population dynamics. Suppose we consider a cartel emerging for three building blocks with well-ordered supply rates. Removing the pure-converter strategy causes the external concentrations of the two least abundant building blocks to decrease, yielding a decrease of both remaining cell-type populations, and potentially their extinction. Removing the mixed strategy causes the most abundant external concentrations to increase but the least abundant building block to decrease, yielding an increase of the pure-converter population and a decrease of the pure-importer population, and potentially its extinction. Removing the pure-importer strategy causes the external concentrations of the two least abundant building blocks to increase, yielding an increase of both remaining strategies. However, in all cases, the overall cell population will decrease.

Although interesting, we believe that the study of such removal effects is beyond the scope of our analysis. Moreover, it is hard to justify the meaning of such removals from a biological standpoint, since even after a catastrophic event that eliminated a species (e.g. phage infection), new species would likely quickly establish in the available niche.

*M) The word cartel has strong negative associations (i.e. drug cartels), and even at a more basic level the use of this word in the manuscript is fundamentally inaccurate, since the economic definition of a cartel is based on an agreement to control prices, instead of an optimization principle. I am not sure why the authors want to go in this direction.*

We note the reviewer’s comment and agree that the concept of microbial cartel is not a literal transposition of the economics concept. In fact, we only use the word “cartel” as a helpful analogy because we believe that this concept captures the idea that the benefit of diversity stems from the ability to collectively control resource availabilities. We followed the reviewer’s recommendation and now clarify that we use the word “cartel” only by analogy and specify to which extent this analogy applies. We also changed the title of the manuscript to “Microbial consortia at steady supply”. Finally, we made sure that the concept of cartel is not negatively connoted in the main text, stressing that microbial cartels achieve a collective optimum in term of biomass yield.

[Editors’ note: the author responses to the re-review follow.]

*The manuscript has been greatly improved and both referees enthusiastically recommend publication.*

*Reviewer #1:*

*In "Microbial consortia at steady supply", the authors consider a population model in which microorganisms exposed to a flux of metabolites follow different strategies concerning which metabolites to import and which to produce through conversion. They show that populations evolve to form cartels, which prevent invasion from other strategies and keep external resources fixed. Overall, I think the authors do a good and extensive job of demonstrating their results. I recommend this paper for publication.*

*Reviewer #2:*

*Taillefumier et al. present an elegant and compelling theory of microbial coexistence driven by diverse strategies for import and synthesis of essential metabolic building blocks. I found the revised manuscript to be improved over the original submission, making several changes that facilitated communication of the main results. In particular, the elaborated explanation of the dynamics of the simulation better clarifies that communities are converging toward optimal consortia (rather than being mistaken for oscillatory dynamics). Additionally, I found it reassuring that the new results using gradient-based optimization of communities arrived at similar results as the simulations that were continuously challenged with randomly sampled species types. This resolves one of my main concerns from the original manuscript, and shows that the authors' theory of optimal cartels captures essential aspects of the simulations despite extensive simplifications. Finally, the earlier definition of cartels in the Introduction aided understanding, and multiple revisions in the Discussion helped for putting the theory in context.*

*This is a high-quality work in theoretical ecology that makes significant progress toward the major problem of understanding microbial diversity, and I support its publication in eLife where it can reach a broad biological audience.*

We made two minor modifications to the manuscript in order to address the reviewers’ minor points: i) We added a sentence in the Model section stating that our growth model is inspired from protein elongation whereby building blocks (amino acids) are sequentially incorporated into biomass (protein). ii) We corrected the erroneously labeled time axes in panels B and C of Figure 2.